# Mixture of Mini Experts: Overcoming the Linear Layer Bottleneck in Multiple Instance Learning

**Daniel Shao**[1,2]    **Joel Runevic**[2]    **Richard J. Chen**[2]    **Drew F.K. Williamson**[3]
**Ahrong Kim**[2]    **Andrew H. Song**[2,4,†,*]    **Faisal Mahmood**[2,*]
[1]Massachusetts Institute of Technology, Cambridge MA    [2]Harvard University, Cambridge MA
[3]Emory University, Atlanta GA    [4] MD Anderson Cancer Center, Houston TX
dshao@mit.edu, asong2@mdanderson.org, FaisalMahmood@bwh.harvard.edu [*]

## Abstract

Multiple Instance Learning (MIL) is the predominant framework for classifying gigapixel whole-slide images in computational pathology. MIL follows a sequence of 1) extracting patch features, 2) applying a linear layer to obtain task-specific patch features, and 3) aggregating the patches into a slide feature for classification. While substantial efforts have been devoted to optimizing patch feature extraction and aggregation, none have yet addressed the second point, the critical layer which transforms general-purpose features into task-specific features. We hypothesize that this layer constitutes an overlooked performance bottleneck and that stronger representations can be achieved with a low-rank transformation tailored to each patch's phenotype, yielding synergistic effects with any of the existing MIL approaches. To this end, we introduce Mammoth, a parameter-efficient, multi-head mixture of experts module designed to improve the performance of any MIL model with minimal alterations to the total number of parameters. Across eight MIL methods and 19 different classification tasks, we find that such task-specific transformation has a larger effect on performance than the choice of aggregation method. For instance, when equipped with Mammoth, even simple methods such as max or mean pooling attain higher average performance than any method with the standard linear layer. Overall, Mammoth improves performance in 130 of the 152 examined configurations, with an average $+3.8\%$ change in performance.

## 1 Introduction

The technical advancements in computational pathology (CPath) have significantly transformed analysis of whole-slide images (WSIs), enabling machine learning models to achieve pathologist-level precision in diverse clinical tasks (Song et al., 2023; Bejnordi et al., 2017; Campanella et al., 2019; Bulten et al., 2022). However, unique challenges arise when analyzing gigapixel WSIs due to their immense size and morphological heterogeneity that spans diverse tissue structures, cellular formations, and spatially distributed pathological characteristics (Saltz et al., 2018; Abdul Jabbar et al., 2020; Marusyk & Polyak, 2010). In this context, multiple instance learning (MIL) frameworks have emerged as the cornerstone approach to distill gigapixel images into condensed slide-level representations for accurate downstream performance (Chen et al., 2024b; Lu et al., 2021; Shao et al., 2021; Wagner et al., 2023; Li et al., 2021). The MIL framework consists of three stages: 1) Dividing a WSI into a set of smaller image patches, which are encoded into general-purpose features with a patch feature encoder, 2) transforming the *general-purpose* features into *task-specific* features with a linear layer, and 3) aggregating the feature set into a slide-level representation. The first and last stages have been studied substantially, through histopathology foundation models that produce features encompassing diverse histomorphological concepts (Wang et al., 2022; Xu et al., 2024; Chen et al.,

---

[*]Equal supervision, † Work done primarily while at Harvard University, Code available at https://github.com/mahmoodlab/mammoth,

2024a; Wang et al., 2024; Lu et al., 2024) and aggregation architectures that yield task-optimized slide representations (Ilse et al., 2018; Lu et al., 2021; Shao et al., 2021; Campanella et al., 2024).

However, the critical intermediate step of encoding task-specific patch features remains unexplored. Most MIL models obtain task-specific representation by applying the same linear layer to all patch embeddings, regardless of their morphological content. We hypothesize that applying a single transformation to all patches limits the model's ability to capture diverse morphological features, ultimately reducing the quality of slide-level predictions. In breast cancer lesion subtyping, for example, diverse concepts such as epithelial cell morphology, spatial arrangement, and stromal layer architectures are collectively important factors for diagnosis (Brancati et al., 2021). This diversity suggests that the task-specific transformation would ideally separate patch embeddings into clusters corresponding to distinct morphological concepts; while in practice, the output of the linear layer forms a relatively continuous embedding space (**Fig. 1A**). As a result, MIL aggregation may struggle to distinguish between the array of morphological concepts necessary for a comprehensive slide-level representation.

These insights warrant a more flexible architecture that can adapt its transformations based on the morphological content of each patch. Mixture of experts (Jacobs et al., 1991; Jordan & Jacobs, 1994; Eigen et al., 2013) (MoE) presents a promising solution by maintaining a collection of specialized linear layers, known as *experts*, each optimized to process a different morphological pattern. A dynamic routing mechanism directs each patch to the most appropriate expert, enabling more nuanced feature transformations than those of a single linear layer (Eigen et al., 2013; Shazeer et al., 2017; Cai et al., 2024). However, a critical challenge of MoE is training instability: the hard assignments of experts to inputs lead to poor gradient flow, leading to imbalanced expert utilization, with certain experts receiving most inputs (Cai et al., 2024). Learning an effective hard assignment is particularly challenging in CPath due to the massive number of patch features ($\approx 10,000$ per sample) and the small number of training samples ($< 1,000$ patients) compared to traditional MoE tasks. MIL

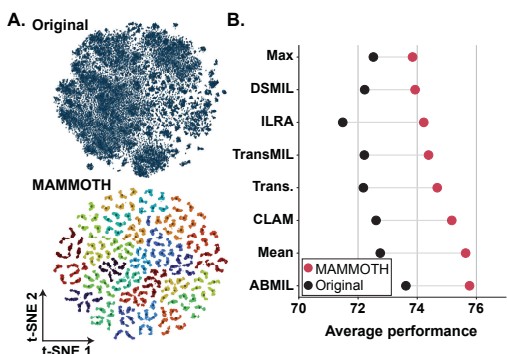

Figure 1: **The plug-and-play MoE module for MIL.** MAMMOTH replaces the task-specific linear layer in MIL models with a mixture of experts. (a) MAMMOTH leads to a structured embedding space (each expert corresponds to a different color) in contrast to the original linear layer and (b) results in improved slide-level classification performance, regardless of MIL model.

models also frequently suffer from poor generalizability: adding more experts can exacerbate these problems, increasing the risk of overfitting due to the expanded parameter count (Shao et al., 2025).

To address these challenges, we propose MAMMOTH, a MoE module that replaces the task-specific linear layer for learning specialized patch feature transformations. MAMMOTH is a plug-and-play module that can be integrated into any MIL model to improve downstream performance (**Fig. 1B.**), operating with the same parameter budget as the linear layer. Instead of hard expert assignments that lead to training instability, MAMMOTH leverages soft expert assignment where each expert processes a different linear combination of all patch embeddings, improving gradient flow and expert utilization (Puigcerver et al., 2024; Liu et al., 2024). Building on this foundation, MAMMOTH introduces several model designs uniquely suited to addressing the challenges of CPath slide classification. First, we partition each patch embedding into multiple embedding heads, with each smaller embedding processed in parallel by different MoE heads. This multihead approach not only provides fine-grained control over the patch embedding subspace but also handles larger patch embedding size ($>1,024$) compared to that of typical input token in natural images (196 or 256). Next, we employ low-rank decomposition in expert layers and weight sharing for parameter efficiency, enabling MAMMOTH to replace the original linear layer without altering the model size. Finally, MAMMOTH produces a compact set of output embeddings from the large input patch embedding set ($> 25\times$ reduction). This distills the large, noisy input set to a compact set of representative morphological aggregates, akin to prototype-based aggregation (Vu et al., 2023; Song et al., 2024a;b).

Our work demonstrates that applying multiple small, specialized transformations to each patch embedding via MAMMOTH substantially outperforms the conventional approach of using a single, larger transformation for all patch embeddings (**Fig. 1B**). Our key contributions are as follows:

- We propose MAMMOTH, a general-purpose MoE layer designed for gigapixel WSI classification that can be easily integrated into any MIL framework.
- We identify the task-specific linear layer as a critical performance bottleneck, showing that MAMMOTH improves performance in 130 out of the 152 examined configurations and allows simple MIL methods to outperform sophisticated MIL methods at baseline.
- Interpretability analyses confirm that MAMMOTH experts learn to specialize in distinct morphological concepts.
- Extensive ablations reveal that MAMMOTH surpasses other MoE adaptations in CPath.

## 2 RELATED WORKS

**Mixture of Experts (MoE):** MoE processes the input with *experts*, each tailored to different input spaces, producing generalizable embeddings. While Sparse MoE, which performs hard assignment of inputs to experts (Cai et al., 2024; Shazeer et al., 2017), is popular due to favorable model size scaling and handling of token heterogeneity (Cai et al., 2024), it often suffers from representation collapse (Chi et al., 2022) and under-utilization of experts (Shazeer et al., 2017; Lepikhin et al., 2020). Among efforts to balance expert utilization (Fedus et al., 2022; Du et al., 2022; Riquelme et al., 2021), Soft MoE stands out by providing a differentiable gating mechanism that routes weighted combinations of inputs across multiple experts (Puigcerver et al., 2024). Consequently, each input receives contributions from several experts, leading to stable training dynamics (Liu et al., 2024; Puigcerver et al., 2024). Alternatively, sparse multihead MoE (Wu et al., 2024a) allows more granular expert specialization by distributing partitioned inputs to multi-head experts.

Despite its success in improving classification performance for small images ($256 \times 256$ pixels), the suitability of MoE for the challenging tasks of classifying gigapixel WSIs in CPath remains unclear. To this end, MAMMOTH builds on the foundations of Soft and multihead MoE to achieve morphological specialization for slide-level classification tasks.

**Parameter-efficient MoE:** Increasing the number of experts or heads for MoE can lead to substantial growth in model size, and ultimately model overfitting (Cai et al., 2024). Recent works have explored lightweight experts by leveraging low-rank adaptors (Zadouri, 2024; Wu et al., 2024b), smaller experts (He, 2024), or matrix factorization (Oldfield et al., 2024; Gao et al., 2022) to reduce parameter count while preserving representational quality. Specifically, matrix factorization decomposes the expert layer weights into a series of low-rank matrices, enabling models to scale the number of experts without substantially increasing the parameters (Wu et al., 2024b). Weight sharing across experts also offers efficiency by reusing weight matrices between experts (Tan et al., 2023; Wu et al., 2024b; Jawahar et al., 2024). MAMMOTH combines these ideas to enable a larger number of experts within the same parameter budget as the linear layer it replaces.

**MoE for computational pathology:** Despite the popularity of MoE in machine learning literature, it remains relatively unexplored for computational pathology. Existing works either use a mixture of attention-based MIL experts to perform multitask mutation prediction (Li et al., 2024) with each expert corresponding to a single task, or train separate CNNs to detect tissue artifacts and weigh each model's prediction through the MoE formulation (Kanwal et al., 2024). However, these are highly tailored to specific tasks, and are not readily extensible to a large suite of MIL models. Recently, a pathology-aware sparse routing mechanism (PaMOE) was proposed to use pre-extracted patch prototypes to encourage experts to specialize in different pathologic contents, replacing the feedforward layers in the transformer encoder block with a standard sparse MoE (Wu et al., 2025). In contrast, MAMMOTH is a highly flexible plug-and-play MoE module built to replace the initial linear layer that universally exists in MIL frameworks (Ilse et al., 2018; Campanella et al., 2024).

**Pre-aggregation modules:** Recent works have explored two primary avenues for processing patch-level features prior to the aggregation layer. The first approach samples a subset of patch features for subsequent aggregation (Neidlinger et al., 2025; Zhu et al., 2025), as a means of regularization or inference-time efficiency. The second approach employs a module to re-embed these patch features in a spatially-aware manner, either with a regional Transformer that is trained along with

the aggregation module to produce task-optimized features (Tang et al., 2024), or by performing local self-attention among collections of neighboring patches (Guo et al., 2025). In contrast, MAM-MOTH fuses global information based on feature similarity rather than spatial proximity, while also performing MoE-based processing without additional parameter burden.

## 3 METHODS

We present MAMMOTH, a **MA**trix-factorized **M**ixture **M**odule of **T**ransformation **H**eads for learning task-specific WSI patch representations in CPath. MAMMOTH can easily replace the standard linear layer of any MIL architecture with a mixture of small, specialized multi-head experts, leading to improved downstream performance with the same parameter count (**Fig. 2**).

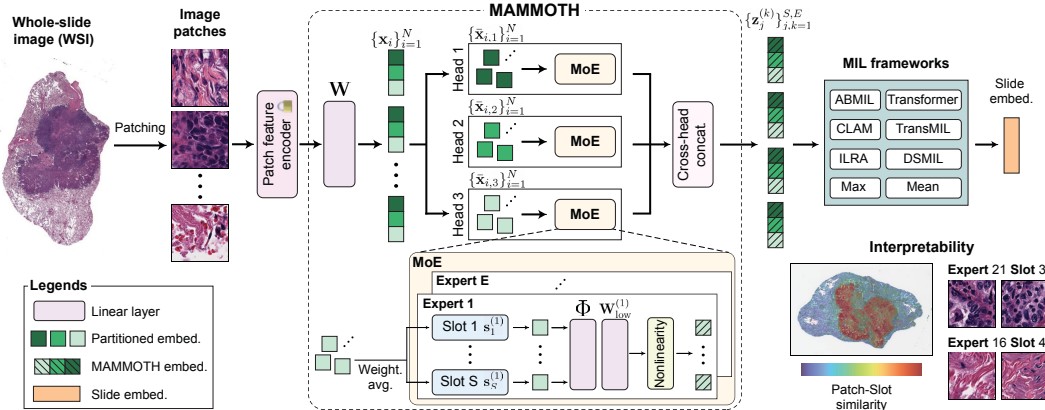

Figure 2: **MAMMOTH architecture** MAMMOTH replaces the initial linear layer of MIL models, transforming generic patch features into task-optimized features with a multiheaded soft MoE. Patch features are routed to different combinations of slots and experts for task- and morphology-specific processing. The MoE outputs are concatenated and fed into the MIL model.

To obtain a set of embeddings for MIL, each WSI is divided into $256 \times 256$ pixel patches, each of which is encoded into an embedding ($\approx$1,024 dim) by a pretrained histopathology patch feature encoder (Campanella et al., 2024). This results in a set of patch embeddings $\mathbf{X} = \{\mathbf{x}_i\}_{i=1}^N, \mathbf{x}_i \in \mathbb{R}^D$ for $N$ patches of a given WSI ($N \approx 10,000$). A standard MIL framework, $f_{\mathrm{MIL}}(\cdot)$, which converts $\mathbf{X}$ into the slide-level embedding $\mathbf{x}_{\mathrm{WSI}} \in \mathbb{R}^{D'}$, can be decomposed into the aggregator $f_{\mathrm{MIL}}^{\mathrm{agg.}}$ and the linear layer $f_{\mathrm{MIL}}^{\mathrm{linear}}$,

$$\mathbf{x}_{\mathrm{WSI}} = f_{\mathrm{MIL}}\left(\{\mathbf{x}_i\}_{i=1}^N\right) = f_{\mathrm{MIL}}^{\mathrm{agg.}}\left(\{f_{\mathrm{MIL}}^{\mathrm{linear}}(\mathbf{x}_i)\}_{i=1}^N\right). \tag{1}$$

MAMMOTH replaces $f_{\mathrm{MIL}}^{\mathrm{linear}}(\cdot)$ with following operations: (1) input partitioning into multiple segments (**Section 3.1**), (2) a slot-based pooling module based on a set of patch prototypes (**Section 3.2**), (3) a low-rank projection with matrix factorization (**Section 3.3**), and (4) concatenation of processed partitions to form output (**Section 3.4**).

### 3.1 MULTI-HEAD PROCESSING OF INPUT EMBEDDINGS

To enhance the expressivity of the input patch embeddings, we employ multi-head processing, where each head accounts for a different partition of the embedding. Specifically, each head consists of a MoE architecture comprised of $E$ experts, each with $S$ slots. After applying linear layer $\mathbf{W} \in \mathbb{R}^{(P \cdot H) \times D}$ to reduce the size of the embedding, it is divided into $H$ non-overlapping partitions, with the $h^{\mathrm{th}}$ head processing the $h^{\mathrm{th}}$ partition. The $h^{\mathrm{th}}$ partition $\bar{\mathbf{x}}_{i,h}$ is given as

$$\bar{\mathbf{x}}_{i,h} = (\mathbf{W}\mathbf{x}_i)[(h-1)P + 1 : hP] \in \mathbb{R}^P. \tag{2}$$

Each set of partitioned embeddings $\{\bar{\mathbf{x}}_{i,h}\}_{i=1}^N$ is independently processed by a distinct MoE, prior to the head-level concatenation at the last stage. For notational decluttering, we drop the subscript $h$ for **Sections 3.2** and **3.3**, noting that the same operations are performed on all heads. This is different

from Multihead MoE (Wu et al., 2024a), which flattens the partitioned embeddings into a larger set of $N \cdot H$ embeddings and processes them with a shared pool of experts.

## 3.2 SLOT-BASED POOLING

We apply slot-based pooling to obtain linear combinations of $\{\bar{\mathbf{x}}_i\}_{i=1}^N$, with each slot representing a unique morphological concept. For a given expert $k$, we pool the embeddings $\{\bar{\mathbf{x}}_i\}_{i=1}^N$ to $S$ slots via weighted averaging, based on the similarity of each input embedding to slot-specific trainable and randomly initialized prototypes $\{\mathbf{s}_j^{(k)}\}_{j=1}^S$ with $\mathbf{s}_j^{(k)} \in \mathbb{R}^P$. The similarity score of an input embedding with each prototype is computed with the inner product, normalized with a softmax operation across $N$ embeddings. The score $\alpha_{j,i}^{(k)}$ represents the similarity of the $i^{\text{th}}$ embedding to slot $j$ of expert $k$, and is used to compute the slot embedding $\mathbf{u}_j^{(k)} \in \mathbb{R}^P$,

$$\alpha_{j,i}^{(k)} = \frac{\exp(\langle \bar{\mathbf{x}}_i, \mathbf{s}_j^{(k)} \rangle)}{\sum_{i'=1}^N \exp(\langle \bar{\mathbf{x}}_{i'}, \mathbf{s}_j^{(k)} \rangle)}, \quad \mathbf{u}_j^{(k)} = \sum_{i=1}^N \alpha_{j,i}^{(k)} \cdot \bar{\mathbf{x}}_i, \tag{3}$$

where $\langle \cdot, \cdot \rangle$ denotes the inner product and $\mathbf{u}_j^{(k)}$ is computed as the average of input embeddings weighted by the similarity scores. The non-zero score $\alpha_{j,i}^{(k)}$ forms the basis of soft expert assignment, by allowing all patch embeddings to contribute to every slot and consequently to every expert. In this context, each weighted average can be interpreted as a summary of a distinct histomorphological feature in the WSI, as demonstrated in **Figures 3** and **A3- A7**.

## 3.3 LOW-RANK EXPERTS

With each slot aggregating a distinct morphological concept, we introduce experts to perform feature transformations tailored to each slot. For each expert, MoE typically uses an MLP to process the slot embedding, $\mathbf{z}_j^{(k)} = \text{LayerNorm}(\text{ReLU}(\mathbf{W}_{\text{full}}^{(k)} \mathbf{u}_j^{(k)}))$, where $\mathbf{W}_{\text{full}}^{(k)} \in \mathbb{R}^{(D'/H) \times P}$ represent the linear transformations and the ReLU and layer normalization represent additional nonlinearity.

The dense matrix $\mathbf{W}_{\text{full}}^{(k)}$, however, presents a scaling challenge as the parameter count increases proportionally with the number of experts. To alleviate this undesirable scaling property, we approximate $\mathbf{W}_{\text{full}}^{(k)}$ as a composition of light-weight expert-specific $\mathbf{W}_{\text{low}}^{(k)} \in \mathbb{R}^{(D'/H) \times Q}$ and shared $\Phi \in \mathbb{R}^{Q \times P}$ weight matrices. The low-rank expert output, $\mathbf{z}_j^{(k)} \in \mathbb{R}^{D'/H}$, is given as

$$\mathbf{z}_j^{(k)} = \text{LayerNorm}(\text{ReLU}(\mathbf{W}_{\text{low}}^{(k)} \cdot \Phi \mathbf{u}_j^{(k)})). \tag{4}$$

Such low-rank decomposition (Hu et al., 2021; Handschutter et al., 2020), $\mathbf{W}_{\text{full}}^{(k)} \simeq \mathbf{W}_{\text{low}}^{(k)} \cdot \Phi$, allows us to scale the number of experts while maintaining a fixed parameter budget.

## 3.4 MAMMOTH OUTPUT FOR DOWNSTREAM TASKS

The low-rank expert output $\mathbf{z}_{j,h}^{(k)}$, corresponding to head $h$, is concatenated across all heads to form the final MAMMOTH output, $\mathbf{z}_j^{(k)} = \text{Concat}([\mathbf{z}_{j,1}^{(k)}, \ldots, \mathbf{z}_{j,H}^{(k)}]) \in \mathbb{R}^{D'}$. Consequently, the output set $\{\mathbf{z}_j^{(k)}\}_{j,k=1}^{S \cdot E}$, instead of the original embedding set $\{\mathbf{x}_i\}_{i=1}^N$, is processed by $f_{\text{MIL}}^{\text{agg.}}$. This differs from Soft MoE (Puigcerver et al., 2024) which returns the updated patch embeddings $\{\hat{\mathbf{x}}_i\}_{i=1}^N$ of the same set size as the input, computed as a linear combination of $\{\mathbf{z}_j^{(k)}\}_{j,k=1}^{S \cdot E}$. In contrast, MAMMOTH condenses morphological information into a smaller set of $S \cdot E \ll N$ task-specific embeddings. This reduced number of input embeddings for $f_{\text{MIL}}^{\text{agg.}}$ facilitates stable model training by simplifying the aggregation step, similar to prototype-based approaches (Vu et al., 2023; Song et al., 2024a;b).

# 4 EXPERIMENTS

## 4.1 DATASETS

**Morphological Tasks**: We evaluate on six morphological classification tasks: EBRAINS fine-grained (EBRAINS-F, $C = 30$ classes) and coarse-grained subtyping (EBRAINS-C, $C = 12$) for

rare brain cancer ($n = 2,319$ slides) (Roetzer-Pejrimovsky et al., 2022); Non-Small Cell Lung Carcinoma (NSCLC, $C = 2$) subtyping with 5-fold cross-validation (CV) on TCGA ($n = 1,041$), with external validation on the CPTAC ($n = 1,091$) and NLST ($n = 1,008$) (Campbell et al., 2016); ISUP grading based on the PANDA prostate cancer challenge ($C = 6$, $n = 10,616$) (Bulten et al., 2022); BRACS breast carcinoma subtyping with coarse (BRACS-C, $C = 3$) and fine (BRACS-F, $C = 7$) granularity ($n = 547$) (Brancati et al., 2021).

**Molecular biomarker prediction**: We evaluate MAMMOTH on 13 molecular biomarker status prediction tasks: glioma IDH1 mutation prediction (GBMLGG-C, $C = 2$) and histomolecular subtyping (GBMLGG-F, $C = 5$) on TCGA GBMLGG ($n = 1,123$) with external evaluation on EBRAINS cases with IDH1 status ($n = 849$) (Roetzer-Pejrimovsky et al., 2022), 5-fold CV on TCGA lung mutation status for TP53, KRAS, STK11, and EGFR ($C = 2$, $n = 524$), TCGA breast cancer mutation status for HER2, ER, PIK3CA, and PR ($C = 2$, $n = 1,034$), and 10-fold CV on breast core needle biopsy (BCNB) (Xu et al., 2021) for ER, PR, and HER2 ($C = 2$, $n = 1,058$).

We use AUC for binary tasks and balanced accuracy for multiclass tasks, with weighted $\kappa$ for the grading task. We use official dataset splits or splits presented in UNI (Chen et al., 2024a) otherwise. For tasks with external cohorts, we report the macro-averaged performance between each cohort.

**Survival prediction**: We evaluate on four survival prediction tasks, using overall survival as the outcome: 5-fold site-stratified CV on TCGA breast cancer (BRCA, $n = 1,041$), Colorectal cancer (Surgen, $n = 427$), TCGA lung adenocarcinoma (LUAD, $n = 456$), and TCGA lung squamous cell carcinoma (LUSC, $n = 471$). For LUAD and LUSC, we also perform external validation with CPTAC (LUAD, $n = 185$; LUSC, $n = 98$) and NLST (LUAD, $n = 244$; LUSC, $n = 118$). We use concordance index to assess the concordance between the rankings of true and predicted risks.

## 4.2 EVALUATION

**Baselines:** We evaluate MAMMOTH by replacing the initial linear layer for ABMIL (Ilse et al., 2018), CLAM (Lu et al., 2021), TransMIL (Shao et al., 2021), Transformer (Wagner et al., 2023; Vaswani, 2017), ILRA (Xiang & Zhang, 2023), DSMIL (Li et al., 2021), MeanMIL, and MaxMIL. We use the published hyperparameter values for all models. Additional details are in **Section A1**.

**Implementation:** WSIs at $20\times$ magnification ($0.5$ $\mu$m/pixel) were tessellated into $256{\times}256$ patches. We extracted features using UNI (Chen et al., 2024a), a ViT-L/16 DINOv2-based model (Oquab et al., 2024) pretrained on $10^5$ internal histology slides. We use $E = 30$ experts, $H = 16$ heads, and $S = 9$ slots per expert. We set $P = 256/H$, and $Q = \lfloor \frac{DD' - DPH}{HP + ED'} \rfloor$ to keep the number of trainable parameters close to that of the original linear layer. Additional details are in **Section A2**.

## 5 RESULTS

### 5.1 DOWNSTREAM CLINICAL TASK PERFORMANCE

**Morphological Classification:** Across all six classification tasks, eight testing cohorts, and eight MIL methods, MAMMOTH yields an average percent change of $+7.36\%$ (**Table 1**). Overall, 46 out of the 48 evaluated configurations showed a performance increase. We find that both cases of decrease occur in NSCLC subtyping, a relatively simple binary task with high average performance, which may not benefit as extensively from the morphological specialization by MAMMOTH.

**Molecular biomarker prediction:** Average performance across biomarkers within each dataset is shown in **Table 2**. At the dataset-level, we find that MAMMOTH improves the average performance in every configuration. At the individual biomarker level (**Table A2**), MAMMOTH improves performance in 84 out of the 104 total configurations, with an average percent change of $+2.1\%$. For challenging tasks with lower baseline AUC performance (e.g., BRCA PIK3CA and Lung KRAS), improvements with MAMMOTH were variable compared to tasks with overall higher AUC. Unlike tissue subtyping that relies on H&E morphology, the ground truth for biomarker status is determined by molecular tests or supplemental stains. Consequently, the biomarkers with low baseline performance may lack adequate signal to be reliably identified from morphology alone (Kather et al., 2020; Fu et al., 2020), and not benefit as consistently from MoE. Despite this, average performance increased across all tasks, underscoring MAMMOTH's adaptability to diverse tasks and organs.

Table 1: **Tissue subtyping.** MIL performance with and without MAMMOTH. The number of classes ($C$) is indicated below each task, with its evaluation metric in parentheses. Standard deviation across 1,000 bootstrap trials is reported in parentheses. Trans., Transformer.

| Task | Status | ABMIL | CLAM | TransMIL | Trans. | ILRA | Mean | Max | DSMIL | Avg. |
|---|---|---|---|---|---|---|---|---|---|---|
| BRACS-C | Base | $67.10_{(1.2)}$ | $56.16_{(1.0)}$ | $66.80_{(2.7)}$ | $63.40_{(2.8)}$ | $63.27_{(1.8)}$ | $65.13_{(1.7)}$ | $64.54_{(2.4)}$ | $62.64_{(2.4)}$ | 63.63 |
| $C = 3$ | +Ours | $72.70_{(1.4)}$ | $73.41_{(0.2)}$ | $70.52_{(3.1)}$ | $71.11_{(3.6)}$ | $74.05_{(2.9)}$ | $72.37_{(1.4)}$ | $67.21_{(1.6)}$ | $68.48_{(2.8)}$ | 71.23 |
| (Bal. acc.) | $\Delta$ | +5.60 | +17.25 | +3.72 | +7.71 | +10.78 | +7.25 | +2.67 | +5.84 | +7.60 |
| BRACS-F | Base | $42.84_{(2.5)}$ | $32.26_{(2.5)}$ | $32.10_{(2.7)}$ | $35.70_{(1.9)}$ | $32.65_{(2.4)}$ | $33.68_{(1.4)}$ | $33.90_{(2.4)}$ | $36.48_{(4.2)}$ | 34.95 |
| $C = 7$ | +Ours | $46.12_{(2.4)}$ | $46.82_{(0.6)}$ | $38.32_{(1.0)}$ | $38.95_{(2.0)}$ | $42.50_{(1.9)}$ | $43.55_{(2.9)}$ | $35.52_{(0.5)}$ | $39.72_{(0.5)}$ | 41.44 |
| (Bal. acc.) | $\Delta$ | +3.28 | +14.56 | +6.22 | +3.25 | +9.85 | +9.87 | +1.62 | +3.24 | +6.49 |
| EBRAINS-C | Base | $86.10_{(1.1)}$ | $87.85_{(1.0)}$ | $87.86_{(1.1)}$ | $86.94_{(0.6)}$ | $83.41_{(1.7)}$ | $86.70_{(0.7)}$ | $84.55_{(1.2)}$ | $86.37_{(2.0)}$ | 86.22 |
| $C = 12$ | +Ours | $89.98_{(0.7)}$ | $91.32_{(0.2)}$ | $88.23_{(1.2)}$ | $90.45_{(0.9)}$ | $91.68_{(0.8)}$ | $89.42_{(1.1)}$ | $85.14_{(0.1)}$ | $89.17_{(0.3)}$ | 89.43 |
| (Bal. acc.) | $\Delta$ | +3.88 | +3.47 | +0.37 | +3.51 | +8.27 | +2.72 | +0.59 | +2.81 | +3.20 |
| EBRAINS-F | Base | $67.20_{(1.0)}$ | $69.77_{(0.6)}$ | $65.20_{(0.5)}$ | $69.07_{(1.7)}$ | $64.64_{(1.2)}$ | $70.30_{(1.4)}$ | $64.94_{(1.0)}$ | $63.87_{(1.7)}$ | 66.87 |
| $C = 30$ | +Ours | $72.40_{(1.2)}$ | $72.51_{(0.6)}$ | $74.22_{(0.2)}$ | $69.73_{(0.1)}$ | $70.23_{(0.3)}$ | $72.89_{(0.2)}$ | $68.22_{(0.3)}$ | $69.40_{(0.4)}$ | 71.20 |
| (Bal. acc.) | $\Delta$ | +5.20 | +2.74 | +9.02 | +0.66 | +5.59 | +2.59 | +3.28 | +5.53 | +4.33 |
| NSCLC | Base | $94.68_{(0.1)}$ | $91.73_{(0.0)}$ | $93.90_{(0.1)}$ | $94.69_{(0.1)}$ | $93.25_{(0.1)}$ | $91.44_{(0.1)}$ | $94.86_{(0.1)}$ | $94.08_{(0.1)}$ | 93.58 |
| $C = 2$ | +Ours | $94.68_{(0.1)}$ | $93.72_{(0.0)}$ | $93.99_{(0.1)}$ | $94.04_{(0.1)}$ | $93.87_{(0.1)}$ | $93.91_{(0.1)}$ | $94.44_{(0.1)}$ | $94.43_{(0.1)}$ | 94.14 |
| (AUROC) | $\Delta$ | +0.00 | +1.99 | +0.10 | -0.65 | +0.62 | +2.47 | +0.42 | +0.35 | +0.56 |
| PANDA | Base | $93.12_{(0.2)}$ | $92.60_{(0.1)}$ | $90.75_{(0.7)}$ | $91.39_{(0.5)}$ | $91.89_{(0.4)}$ | $92.67_{(0.3)}$ | $88.79_{(0.3)}$ | $92.78_{(0.2)}$ | 91.75 |
| $C = 6$ | +Ours | $94.28_{(0.2)}$ | $93.26_{(0.1)}$ | $93.68_{(0.3)}$ | $91.90_{(0.8)}$ | $94.07_{(0.5)}$ | $93.52_{(0.2)}$ | $92.34_{(0.2)}$ | $92.96_{(0.1)}$ | 93.25 |
| (Weighted $\kappa$) | $\Delta$ | +1.15 | +0.65 | +2.93 | +0.51 | +2.18 | +0.85 | +3.55 | +0.19 | +1.50 |

Table 2: **Molecular Biomarker Prediction Averages** MIL model performance with the standard linear layer (Base) and MAMMOTH (Ours). Each biomarker is a separate task, and results are averaged across tasks within each dataset. Balanced accuracy is reported for GBMLGG-F, and AUROC is reported otherwise. Propagated standard error specified in parentheses.

| Dataset | Status | ABMIL | CLAM | TransMIL | Trans. | ILRA | Mean | Max | DSMIL | Avg. |
|---|---|---|---|---|---|---|---|---|---|---|
| | Base | $81.97_{(0.2)}$ | $83.10_{(0.3)}$ | $80.76_{(0.4)}$ | $80.36_{(0.3)}$ | $81.03_{(0.3)}$ | $82.69_{(0.1)}$ | $82.91_{(0.3)}$ | $81.30_{(0.3)}$ | 81.76 |
| BCNB | +Ours | $84.26_{(0.2)}$ | $84.98_{(0.1)}$ | $82.97_{(0.2)}$ | $83.74_{(0.1)}$ | $83.27_{(0.2)}$ | $84.46_{(0.1)}$ | $84.00_{(0.2)}$ | $82.73_{(0.1)}$ | 83.80 |
| (3 tasks) | $\Delta$ | +2.29 | +1.89 | +2.21 | +3.38 | +2.24 | +1.78 | +1.09 | +1.44 | +2.04 |
| | Base | $71.97_{(0.3)}$ | $71.93_{(0.3)}$ | $71.38_{(0.7)}$ | $71.35_{(0.4)}$ | $70.40_{(0.5)}$ | $71.34_{(0.4)}$ | $72.47_{(0.7)}$ | $71.92_{(0.3)}$ | 71.59 |
| BRCA | +Ours | $73.65_{(0.3)}$ | $72.27_{(0.2)}$ | $73.41_{(0.3)}$ | $73.20_{(0.2)}$ | $71.68_{(0.4)}$ | $73.60_{(0.3)}$ | $72.87_{(0.2)}$ | $73.18_{(0.2)}$ | 72.98 |
| (4 tasks) | $\Delta$ | +1.68 | +0.34 | +2.04 | +1.85 | +1.28 | +2.26 | +0.40 | +1.26 | +1.39 |
| | Base | $67.04_{(0.5)}$ | $66.36_{(0.4)}$ | $65.25_{(0.7)}$ | $64.94_{(0.7)}$ | $65.27_{(0.9)}$ | $66.24_{(0.4)}$ | $65.50_{(1.2)}$ | $65.02_{(0.6)}$ | 65.70 |
| Lung | +Ours | $68.41_{(0.6)}$ | $66.89_{(0.3)}$ | $65.95_{(0.6)}$ | $67.46_{(0.4)}$ | $65.32_{(0.4)}$ | $69.17_{(0.4)}$ | $67.35_{(0.4)}$ | $66.03_{(0.4)}$ | 67.07 |
| (4 tasks) | $\Delta$ | +1.37 | +0.53 | +0.70 | +2.53 | +0.05 | +2.94 | +1.85 | +1.00 | +1.37 |
| | Base | $71.85_{(0.7)}$ | $72.08_{(0.7)}$ | $73.32_{(0.8)}$ | $72.00_{(1.3)}$ | $71.64_{(0.5)}$ | $72.01_{(0.5)}$ | $72.83_{(0.6)}$ | $72.21_{(1.2)}$ | 72.24 |
| GBMLGG | +Ours | $74.20_{(0.9)}$ | $72.78_{(0.2)}$ | $73.98_{(0.7)}$ | $74.40_{(0.6)}$ | $73.00_{(0.4)}$ | $73.48_{(0.7)}$ | $73.63_{(0.3)}$ | $72.74_{(0.4)}$ | 73.53 |
| (2 tasks) | $\Delta$ | +2.35 | +0.70 | +0.65 | +2.41 | +1.36 | +1.47 | +0.80 | +0.53 | +1.28 |

The average performance of MAMMOTH across all morphological and molecular tasks is shown in **Fig. 1B.**. We observe that MAMMOTH-based models consistently outperform MIL approaches, with even the lowest-performing model (MaxMIL, 73.9%) with MAMMOTH exceeding the strongest baseline (ABMIL, 73.6%). Interestingly, MAMMOTH allows simple non-parametric approaches, mean pooling and max pooling, to surpass the strong ABMIL baseline by 2.0% and 0.3%, respectively. These results suggest that the linear layer is a bottleneck for performance, with the inclusion of MAMMOTH leading to larger improvements, compared to changing MIL architectures.

**Survival prediction:** We observe that MAMMOTH also improves the prognostication performance over the baselines in 30/32 configurations (**Table 3**), yielding an average improvement of +2.78 percentage points on the C-index. These results, taken together with the previous classification results, underscore the versatility of MAMMOTH for both diagnostic and prognostic tasks.

Table 3: **Survival prediction.** MIL model performance with the standard linear layer (Base) and MAMMOTH (Ours) using overall survival as the clincal outcome. For LUAD and LUSC, the results are averaged across TCGA, CPTAC, and NLST cohorts. Concordance index is reported. Standard deviation is reported in parentheses.

| Task | State | ABMIL | CLAM | TransMIL | Trans. | ILRA | Mean | Max | DSMIL | Avg. |
|---|---|---|---|---|---|---|---|---|---|---|
| | Base | $58.56_{(4.5)}$ | $61.91_{(4.1)}$ | $58.86_{(3.9)}$ | $56.90_{(4.1)}$ | $57.26_{(5.2)}$ | $58.27_{(8.5)}$ | $56.69_{(5.1)}$ | $59.31_{(4.3)}$ | 58.68 |
| BRCA | +Ours | $63.98_{(4.9)}$ | $64.36_{(3.3)}$ | $65.02_{(5.4)}$ | $65.23_{(4.2)}$ | $63.94_{(4.1)}$ | $63.18_{(3.9)}$ | $62.70_{(5.3)}$ | $60.43_{(4.4)}$ | 63.48 |
| | $\Delta$ | +5.42 | +2.45 | +6.16 | +8.33 | +6.68 | +4.91 | +6.01 | +1.12 | +4.80 |
| | Base | $63.67_{(4.7)}$ | $63.93_{(4.1)}$ | $57.94_{(3.2)}$ | $60.59_{(4.8)}$ | $62.29_{(4.5)}$ | $64.03_{(5.2)}$ | $56.53_{(4.4)}$ | $60.17_{(5.4)}$ | 61.14 |
| SURGEN | +Ours | $65.64_{(4.5)}$ | $64.99_{(5.3)}$ | $63.10_{(3.9)}$ | $63.91_{(4.9)}$ | $64.80_{(5.2)}$ | $64.97_{(5.2)}$ | $59.31_{(4.6)}$ | $65.11_{(4.7)}$ | 63.98 |
| | $\Delta$ | +1.97 | +1.06 | +5.16 | +3.32 | +2.51 | +0.94 | +2.78 | +4.94 | +2.84 |
| | Base | $58.70_{(3.6)}$ | $58.97_{(3.4)}$ | $56.76_{(4.1)}$ | $58.31_{(5.1)}$ | $57.24_{(4.1)}$ | $59.11_{(3.6)}$ | $55.95_{(4.7)}$ | $58.14_{(4.6)}$ | 57.90 |
| LUAD | +Ours | $60.12_{(3.7)}$ | $61.89_{(2.8)}$ | $60.97_{(2.3)}$ | $60.49_{(3.4)}$ | $58.10_{(4.5)}$ | $60.56_{(3.1)}$ | $57.18_{(4.4)}$ | $57.99_{(3.3)}$ | 59.66 |
| | $\Delta$ | +1.42 | +2.92 | +4.21 | +2.18 | +0.86 | +1.46 | +1.23 | -0.15 | +1.60 |
| | Base | $56.62_{(4.0)}$ | $55.63_{(4.7)}$ | $52.53_{(4.5)}$ | $55.11_{(5.1)}$ | $54.44_{(5.8)}$ | $56.04_{(5.0)}$ | $49.39_{(2.8)}$ | $51.26_{(3.7)}$ | 53.88 |
| LUSC | +Ours | $59.32_{(4.8)}$ | $58.91_{2.1}$ | $53.48_{(2.8)}$ | $55.68_{(5.3)}$ | $54.48_{(3.9)}$ | $58.30_{(5.1)}$ | $50.12_{(3.3)}$ | $55.69_{(2.6)}$ | 55.75 |
| | $\Delta$ | +2.70 | +3.28 | +0.94 | -0.44 | +0.03 | +2.26 | +0.73 | +4.43 | +1.87 |

## 5.2 INTERPRETABILITY

The primary motivation for using MoE with WSIs is to process distinct morphologic phenotypes with specialized experts. To assess whether the routing mechanism led to expert specialization of distinct morphological concepts, two board-certified pathologists examined the routing scores between each slot and patch embedding (**Fig. 3A** and **Section A3**), finding that the model consolidates morphologically similar patches into the same slot. For instance, the patches with high weights routed to slot 5 of expert 21 (**Fig. 3B**) overlap heavily with the tumor region of both LUAD and LUSC slides. The routing scheme consistently routed different morphologies into distinct slots, such as stroma and alveoli to Expert 16, and lymphocytes and red blood cells to Expert 9. These results suggest that the slot aggregation enables expert specialization by grouping the similar patches across a variety of concepts. Additional examples are in **Figs. A3- A7**. In addition, we quantify expert specialization by associating each patch with a morphological concept according to its similarity to a set of histologic terms, using the pathology vision-language model MUSK (Xiang et al., 2025). We confirm that experts differentially prioritize concepts such as tumor cells, alveoli, stroma, and lymphocytes (**Fig. A8**).

**Mechanism of specialization:** We additionally investigated the origins of specialization by tracking the training dynamics of slots identified as specialized experts at model convergence. We observed that these slots exhibit higher routing weights for their target concepts even without training, followed by a sharp increase and stabilization within the first epoch (**Fig. A9**). As foundation model embeddings already cluster similar morphological concepts together (Chen et al., 2024a; Lu et al., 2024; Xiang et al., 2025), we hypothesize that the slot embeddings are able to leverage this implicit grouping from the start (**Figs A10- A11**).

**Instance-Gradient Interference:** Lastly, we find that heterogeneous instances yield conflicting gradient updates in standard linear layers (**Fig. A12A-B**), which we term Instance Gradient Interference (IGI). We observe that MAMMOTH mitigates IGI by routing heterogeneous instances to distinct experts, enabling decoupled updates and increased gradient similarity **(Fig. A12C)**, providing insight into the rapid specialization observed at the start of training.

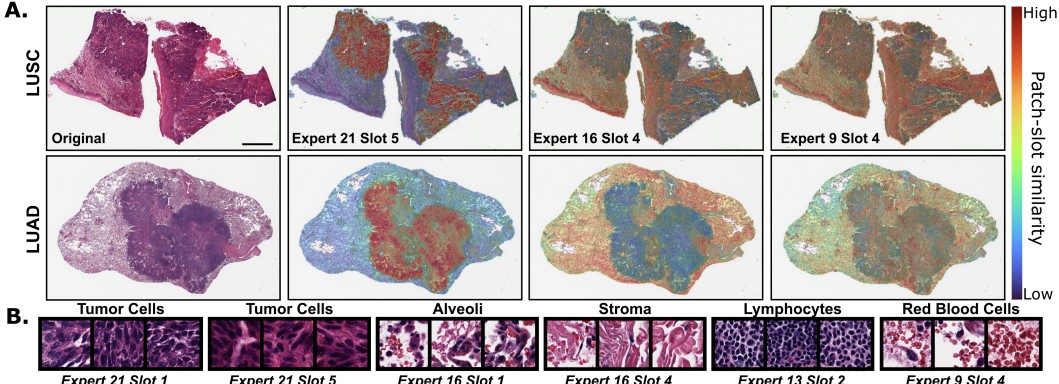

Figure 3: **Visualization of patch routing A.** WSI images of LUSC and LUAD for NSCLC subtyping task, with heatmap of routing weights from patches to three different slots. **B.** Highest similarity patches for each slot among patches from LUSC slide and LUAD slide. Morphological clusters are annotated by two board-certified pathologists, indicating that morphologically similar patches are collected within a single slot. Scale bars: **A.** 500 $\mu$m, **B.** 20 $\mu$m.

## 5.3 ABLATION STUDIES

**Model design ablations**: We first investigate how different components of MAMMOTH affect downstream performance by removing each design component. Performance is measured with ABMIL averaged across six tasks: BRACS C/F, EBRAINS C/F, and GBMLGG C/F. The key ablations are as follows. **(1) MoE method**: We replace MAMMOTH with various related methods: Soft MoE (Puigcerver et al., 2024) and sparse Multiheaded MoE (Wu et al., 2024a), two popular sparse MoE methods (softmax-based MoE (Shazeer et al., 2017) and sinkhorn-based MoE (Tay et al.,

2020)), the pathology-specific routing method, PaMoE, and the original linear layer. **(2) Num. heads**: We investigate the effect of removing the multihead component of MAMMOTH by setting $H = 1$. **(3) Slot transformation**: We use an expert-specific dense transformation $\mathbf{W}_{\text{full}}^{(k)}$, instead of its low-rank approximation, $\mathbf{W}_{\text{low}}^{(k)}\Phi$. **(4) Shared $\Phi$**: We replace the shared low-rank projection, $\Phi$, with an expert-specific projection to assess the effect of weight-sharing. **(5) Initial projection with W**: We replace the initial projection $\mathbf{W}$ with an identity matrix. This results in higher-dimensional slot representations and increased model size. **(6) MAMMOTH output**: Following Soft MoE, we update the patch embeddings $\{\bar{\mathbf{x}}_i\}_{i=1}^N$ as a linear combination of slot outputs $\{\mathbf{z}_j^{(k)}\}_{j,k=1}^{S \cdot E}$ and feed these updated patch embeddings into the MIL module. **(7) Pre-aggregation module:** We replace MAMMOTH with other approaches that modify the set of patch features prior to MIL aggregation such as patch feature re-embedding (Tang et al., 2024), local self-attention (Guo et al., 2025), or sampling (Zhu et al., 2025). Further details are provided in **Section A4**. **(8) MoE Target:** We evaluate a pathology-specific MoE method which replaces the full MIL aggregation layer with a mixture of ABMIL-based experts (Li et al., 2024), instead of only the linear layer.

Table 4: **Ablation studies over design components.** (a) Ablations for model design components. (b) Inference efficiency comparison. Metrics are measured on random inputs of shape $10,000$ (batch size) $\times 1,024$ (feature dim.) averaged over 1,000 forward passes. Best performance among MoE methods shown in **bold**, second best underlined. (c) Performance with GigaPath, Musk, and Virchow, averaged across ABMIL, TransMIL, MaxMIL, and CLAM. Lin., linear; Sp., Sparse; MH, multihead; soft., softmax; sink., sinkhorn.

### (a) Model design ablations

| Ablation | Model | | | Avg. |
|---|---|---|---|---|
| **Full model** | | Ours | | **71.6** |
| MoE method | Ours | $\Rightarrow$ | Lin. layer | 68.1 (−4.9%) |
| | | | Soft MoE | 66.9 (−6.6%) |
| | | | Sp. MH | 69.1 (−3.5%) |
| | | | Sp. soft. | 67.8 (−5.3%) |
| | | | Sp. sink. | 67.6 (−5.6%) |
| | | | PaMoE | 69.2 (−3.4%) |
| Num. heads | 16 | $\Rightarrow$ | 1 | 67.7 (−5.4%) |
| Slot transform | $\mathbf{W}_{\text{low}}^{(k)}\Phi$ | $\Rightarrow$ | $\mathbf{W}_{\text{full}}^{(k)}$ | 69.0 (−3.6%) |
| $\Phi$ | Shared | $\Rightarrow$ | Per-expert | 70.6 (−1.4%) |
| $\mathbf{W}$ | Learned | $\Rightarrow$ | Identity | 68.2 (−4.7%) |
| Output | Slots | $\Rightarrow$ | Patches | 68.2 (−4.7%) |
| Pre-Aggregation Module | MoE | $\Rightarrow$ | RRT | 69.5 (−2.9%) |
| | | | MIL Drop | 69.6 (−2.8%) |
| | | | Querent | 68.9 (−3.8%) |
| MoE Target | Lin. layer | $\Rightarrow$ | Aggregator (M4) | 67.4 (−5.9%) |

### (b) Inference efficiency for MoEs

| Architecture | Latency (MS) | GPU (MB) | GFLOPs |
|---|---|---|---|
| Linear | 0.6 | 74.0 | 5.3 |
| Sp. Soft | 19.2 | 140.9 | 10.8 |
| Sp. Sink. | 27.5 | 141.2 | 10.8 |
| Sp. MH | 194.0 | 2169.6 | 20.3 |
| Soft | **4.8** | 119.6 | **0.8** |
| PaMoE | 24.8 | 610.8 | 125.9 |
| **Ours** | 19.5 | **89.2** | 2.8 |

### (c) Ablation for feature encoders

| Task | State | GigaPath | Musk | Virchow |
|---|---|---|---|---|
| EBRAINS-C | Base | 86.03 | 81.90 | 84.03 |
| | +Ours | 87.13 | 84.50 | 86.30 |
| | $\Delta$ | +1.10 | +2.60 | +2.27 |
| EBRAINS-F | Base | 66.78 | 65.28 | 66.88 |
| | +Ours | 79.02 | 76.02 | 77.78 |
| | $\Delta$ | +12.24 | +10.74 | +10.90 |
| BRACS-C | Base | 61.28 | 58.90 | 60.60 |
| | +Ours | 66.75 | 66.82 | 69.55 |
| | $\Delta$ | +5.47 | +7.92 | +8.95 |
| BRACS-F | Base | 35.05 | 35.15 | 35.20 |
| | +Ours | 41.50 | 43.28 | 41.50 |
| | $\Delta$ | +6.45 | +8.13 | +6.30 |

The results in **Table 4a** show that each design element contributes to MAMMOTH's efficacy. Using alternative single-head MoE methods leads to an average −5.4% change in performance. For both sparse MoE and MAMMOTH, adding a multihead component improves performance, though the benefits of using multiple heads is particularly pronounced in MAMMOTH, in which removing the multihead component leads to a −5.4% change (Num. heads: 16 ⇒ 1), while changing the architecture from sparse multihead to sparse MoE leads to a −2.4% change (Sp. soft ⇒ Sp. MH), emphasizing the confluent benefits of using multiple heads with soft assignments. Replacing MAMMOTH with the pathology-specific PaMoE leads to a −3.4% change in performance. This degradation in performance is present across all 8 MIL methods, with PaMoE exhibiting an average −4.0 decrease in absolute performance compared to MAMMOTH (**Table A5**). Performance within each dataset and parameter count for each MoE method are indicated in **Tables A3** and **A4**.

Using dense expert-specific transformation $\mathbf{W}_{\text{full}}^{(k)}$, removing weight sharing $\Phi$, and removing initial dimensionality reduction layer $\mathbf{W}$ all lead to performance decrease, highlighting the importance of our parameter-efficient design. The Soft MoE approach of using $N$ updated patch representations rather than our proposed $S\cdot$ slot-level outputs leads to a 4.7% performance increase, indicating the benefits of consolidating similar patches for downstream aggregation. While other pre-aggregation

feature modification approaches indeed increase the performance over the original linear layer, they still lag behind MAMMOTH, with even the best performing feature re-embedding approach (RRT) 2.9% worse than MAMMOTH. We report the complete comparison across all classification tasks and MIL methods in **Tables A6, A7, and A8**. Similarly, the MoE-based MIL method, M4, exhibits a sharp drop of -5.9% performance, highlighting the value of targeting the initial task-specific transformation. Performance of M4 across all tasks is shown in **Table A9**.

**Inference-time efficiency:** We evaluate inference-time efficiency for various task-specific transformation layers according to peak GPU memory, per-sample latency, and per-sample GFLOPS in **Table 4b**. The per-sample metrics are averaged over 1,000 forward passes of random samples shaped $10,000$ (batch) $\times 1,024$ (feature). As expected, the linear layer achieves the lowest latency and GPU usage. However, MAMMOTH is both *faster* and more *lightweight* than all Sparse MoE methods. Considering that MAMMOTH also outperformed Soft MoE and the linear layer in downstream tasks, we conclude that MAMMOTH effectively balances performance and efficiency.

**Patch Encoder:** With new CPath feature encoders continuously emerging, we evaluate performance using GigaPath (Xu et al., 2024), Musk (Xiang et al., 2025), and Virchow (Vorontsov et al., 2024) as patch encoders on EBRAINS C/F and BRACS C/F. Across the four MIL methods investigated (ABMIL, CLAM, TransMIL, MaxMIL), MAMMOTH leads to an average improvement in balanced accuracy of +3.52% (GigaPath), +5.36% (MUSK), and +5.24% (Virchow) (**Tables 4c and A1**), indicating that MAMMOTH is robust to feature encoder choice.

**Data efficiency:** A core design principle of MAMMOTH is to facilitate stable training in the data-scarce regimes common in CPath. We test this hypothesis by training ABMIL and TransMIL on different fractions of the training dataset (**Fig. 4**), on EBRAINS-C/F, BRACS-C/F, and GBMLGG-C. This is repeated over three independently sampled training data subsets. MAMMOTH attains the highest overall performance across all fractions compared to other MoE methods. Notably, other MoE methods consistently underperform compared to the linear layer (base) at lower data fractions, highlighting the limitations of traditional MoE approaches for CPath.

**Key hyperparameters**: We perform ablations over the key hyperparameters, $H$, $E$, and $S$. First, we assess the effect of varying $H$ and $E$ for EBRAINS-F, LUNG TP53, and BRCA HER2 tasks with ABMIL and TransMIL. We find that with a low number of heads ($H \in \{2, 4\}$), performance depends on the number of experts selected, with high expert counts ($E \in \{72, 96\}$) showing low overall performance (**Fig. A1**). Meanwhile, increasing the number of heads ($H \in \{8, 16, 32\}$) stabilizes performance, with high expert counts ($E \in \{72, 96\}$) converging with lower expert counts ($E \in \{4, 8\}$). Additionally, we observe that 8-48 experts with $H \in \{16, 32\}$ achieve the highest overall performance. We hypothesize that, because increasing the number of experts leads to a lower rank for $Q$, this intermediate expert count is a "sweet spot" that balances representation capacity with morphological specialization. We conduct a similar experiment varying the total

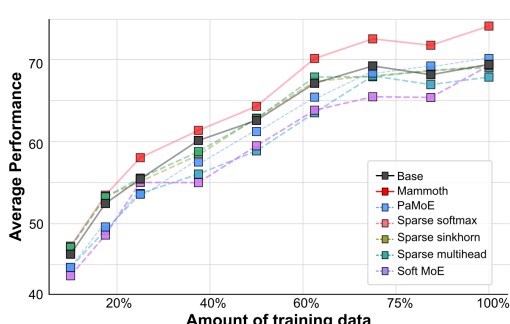

Figure 4: **Data efficiency of MAMMOTH**. MoE performance with varying training samples averaged across tasks EBRAINS-C/F, BRACS-C/F, GBMLGG-C, models ABMIL and TransMIL, and 3 randomly sampled subsets of the training data.

slots, finding that low expert ($E \in \{2, 4\}$) counts reach perform best with low total slots ($50 - 100$), while higher expert counts (24-96 experts) tend to perform better with 200-400 total slots (**Fig. A2**).

## 6 CONCLUSION AND LIMITATIONS

We introduced MAMMOTH, a multihead soft MoE module designed to enhance slide-level performance in computational pathology. MAMMOTH consistently improves classification performance by leveraging a large set of specialized, low-rank feedforward layers, without substantially altering the total parameter count. Limitations include the use of a fixed configuration of experts, slots, and heads for each task. Future works could investigate dynamically selecting the hyperparameters, initializing the slot embeddings with morphological prototypes, and extending to multimodal inputs.

ETHICS STATEMENT

This work utilizes datasets derived from publicly available images of tissues collected from anonymized human subjects. No personally identifiable information was accessible to the authors at any stage of this study. The analysis did not examine model performance across patient demographic subgroups; we acknowledge that further research is needed to ensure algorithmic fairness, particularly with respect to underrepresented populations.

REPRODUCIBILITY STATEMENT

To promote reproducibility, we have submitted the codebase to initialize MAMMOTH, as well as examples for how to equip two popular MIL models, ABMIL and TransMIL, with MAMMOTH. We have described the training details for MAMMOTH in **Sections A1** and **A2** and key ablations in **Sections 5.3, A3.1, A4**. Details for interpretability experiments are described in **Section A3**. All datasets used were publicly available and described in **Sections 4** and **A4.5**.

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

# A APPENDIX

## A1 MULTIPLE INSTANCE LEARNING IMPLEMENTATION

All Multiple instance learning (MIL) models are adapted according to their official implementation, using the default hyperparameters provided by their official codebases. For **MeanMIL**, we obtain a slide-level prediction by feeding the average of task-specific embeddings through a classification head. For **MaxMIL**, we feed each task-specific embedding through a classification head, and select the patch with the single highest logit as the final slide-level prediction. For the baseline of every model, we apply the following linear layer to the pretrained features, $f(x) = \mathrm{ReLU}(\mathbf{W}\mathbf{x})$, where $W \in \mathbb{R}^{D' \times D}$ and input features $\mathbf{x} \in \mathbb{R}^D$. We note that **ILRA** does not natively include an initial task-specific linear layer. Following the architecture of all other MIL examined, which apply a linear layer to the frozen patch embeddings, we introduce this linear layer prior to the ILRA aggregation step.

## A2 TRAINING DETAILS

We train all models with the AdamW optimizer with a learning rate of $1 \times 10^{-4}$, a cosine decay scheduler, and mixed precision according to PyTorch's native implementation. For datasets with a validation set, we train with a maximum of 20 epochs with an early stopping patience of 5 epochs for a minimum of 10 epochs. For datasets without a validation set, we train for 10 epochs. We use cross-entropy loss with random class-weighted sampling and a batch size of 1. For regularization, we use a weight decay of $1 \times 10^{-5}$, a dropout of 0.25 at every feedforward layer, and a dropout of 0.1 on the features from the pretrained encoder. Experiments were performed on one NVIDIA RTX A4000.

## A3 INTERPRETABILITY

We generate interpretable heatmaps by examining the normalized routing scores obtained in **Eq.** 3.2. We average the routing scores across all heads to obtain the slot-patch routing scores shown in **Figures 3** and **A3**. Assessment of the heatmaps and routing scores from two board-certified pathologists reveals that MAMMOTH learned to direct patches with similar morphology to the same slots. We note that the ability for slots to collect patches with similar morphologies is a necessary condition for allowing MAMMOTH experts to specialize in specific morphologic phenotypes. For instance, we show in **Fig. 3** that Expert 9 has likely specialized in processing patches with cells of low diagnostic importance, as one of its slots specializes in patches with lymphocytes, and another one of its patches specializes in red blood cells. Similarly, Expert 21 has three slots which specialize in aggregating both LUSC and LUAD tumor cells. We observe a similar pattern in our BRACS subtyping model, in which patches with ductal hyperplasia are closest in embedding space to slot 2 of expert 4, while patches with ductal carcinoma are strongly routed to slot 5 of expert 4. In this example, expert 4 has likely specialized in processing patches of high diagnostic relevance.

Lastly, the routing scores within different heads of a single slot are shown in **Fig. A7**. Interestingly, we observe that, while the highest-scoring patches routed to each head primarily reside in the tumor region, the distribution of routing scores are highly variable between heads of a single slot. These results, combined with the empirical improvement in performance when the number of heads is set to be $\geq 16$, suggest that our use of multiple heads allows MAMMOTH fine-grained control by partitioning the slot representations into a large number of embedding subspaces.

### A3.1 ABLATIONS

For all ablation experiments, we train for a maximum of 10 epochs with an early stopping patience of 5 epochs, and a minimum of 5 epochs. For our experiments evaluating different configurations of experts and heads in **Section 5.3**, we roughly fix the number of *total* slots across varying numbers of experts by setting the number of slots per expert, $S$, to

$$S = \max(\lfloor (\frac{T}{E}) \rfloor, 1) \tag{}$$

where $T$ is the target number of total slots, and $E$ is the number of experts.

## A4    MIXTURE OF EXPERTS IMPLEMENTATION DETAILS

For comparison with MAMMOTH, we implemented sparsely-gated MoE with softmax and sinkhorn routing (Shazeer et al., 2017; Clark et al., 2022), sparsely-gated multihead MoE (Wu et al., 2024a), soft MoE (Puigcerver et al., 2024), and pathology-aware MoE (Wu et al., 2025) using 5 experts rather than 30 experts for all benchmark MoE methods in order to prevent model capacity from overly expanding.

### A4.1    SPARSE MOE

We implement Softmax MoE according to a PyTorch transcription of the official Tensorflow implementation from GSHard (Lepikhin et al., 2020)(`https://github.com/lucidrains/mixture-of-experts`), using top 2 gating for each patch, alongside an expert capacity factor of 1.25 for training and 2.0 for inference to balance expert utilization. For Sinkhorn MoE, we replace the softmax-based routing mechanism with the Sinkhorn-Knopp algorithm, as described in (Clark et al., 2022). We use 5 experts, with each expert consisting of a $D \times D'$-dimensional linear layer with ReLU activation.

### A4.2    SPARSE MULTIHEAD MOE

We implement sparse multihead MoE from the official implementation (`https://github.com/yushuiwx/MH-MoE`), using 16 heads for all experiments. Following the paper's architecture, we let each expert consist of a 2-layer feedforward network with ReLU activation and set the expert capacity to equal that of the $D \times D'$-dimensional sparse MoE expert layer described above, resulting in a hidden dimension of $\frac{HDD'}{D+D'}$.

### A4.3    SOFT MOE

We use the Soft MoE implementation from `https://github.com/lucidrains/soft-moe-pytorch`. Mirroring the hyperpamaters used in MAMMOTH, we use 200 total slots for morphological classification and 400 total slots for molecular classification tasks. We use 5 experts, with each expert consisting of a $D \times D'$-dimensional linear layer with ReLU activation.

### A4.4    PAMOE

We use the official PaMoE implementation from `https://github.com/wjx-error/PAMoE`. Following the paper's suggested configuration, we use 6 total experts, with 2 free experts and 4 experts initialized according to the matching organ of the evaluation task. For instance, we use the TCGA GBMLGG initialization to evaluate on EBRAINS and GBMLGG. Similarly, we use the TCGA BRCA initialization to evaluate on BRCA and BRACS tasks.

### A4.5    DATASETS

We briefly describe the datasets that were used to evaluate MAMMOTH.

#### A4.5.1    MORPHOLOGICAL SUBTYPING

**EBRAINS** (Roetzer-Pejrimovsky et al., 2022): We perform coarse-grained (12 classes) and fine-grained (30 classes) classification of brain tumor subtypes. The dataset consisted 2,319 Hematoxylin and Eosin (H&E) Formalin-fixed and paraffin-embedded (FFPE) Whole Slide Images (WSIs). We use label-stratified train/val/test splits (50% / 25% / 25%) provided by UNI (Chen et al., 2024a). We evaluate performance using balanced accuracy.

**NSCLC**: The non-small cell lung carcinoma (NSCLC) subtyping task was a binary classification problem for distinguishing lung adenocarcinoma (LUAD) and lung squamous cell carcinoma (LUSC). The training data consisted of publicly available H&E WSIs from TCGA ($n = 1,041$

slides). We used 5-fold site-stratified cross validation on the TCGA dataset for training and internal validation, and evaluated the trained model on two external datasets: the Clinical Proteomic Tumor Analysis Consortium (CPTAC, $n = 1,091$ slides) and the National Lung Screening Trial (NLST, $n = 1,008$ slides) (Campbell et al., 2016; Satpathy et al., 2021; Gillette et al., 2020). We report average AUROC across the five folds for performance on this binary classification task. We report the performance averaged across the TCGA, NLST, and CPTAC datasets in **Table 1**.

**PANDA** (Bulten et al., 2022; 2020): We used prostate cancer core needle biopsies ($n = 10,616$) from the Prostate Cancer Grade Assessment (PANDA) challenge to perform 6-class classification according to the prostate cancer grade. We use the same train/val/test folds (80% / 10 % / 10%) as UNI, and evaluate using Cohen's quadratic weighted Kappa $\kappa$ metric.

**BRACS** (Brancati et al., 2021): The BRACS subtyping task consisted of a 3-class coarse-grained classification task to distinguish benign, malignant, and atypical breast carcinoma H&E slides, as well as a fine-grained 7-class classification task that classifies benign tumors into three subtypes, atypical tumors into two subtypes, and malignant tumors as two subtypes. We use the official train/val/test folds (72% / 12% / 16%), with the same folds for both coarse- and fine-grained tasks. We evaluate performance using balanced accuracy.

### A4.5.2 BIOMARKER PREDICTION

**Lung cancer biomarkers**: We conduct 5-fold cross-validation on H&E-stained WSIs for the binary classification task of predicting mutation status of TP53, KRAS, STK11, and EGFR in TCGA lung cancer cases ($n = 524$ slides) (Cancer Genome Atlas Research Network et al., 2015), with each task site- and label-stratified into an approximate train/val/test splits (60% / 20% / 20%). We evaluate performance using AUROC.

**Breast cancer biomarkers**: We conduct 5-fold cross-validation for the binary classification tasks of predicting mutation status of ER, PR, HER2, and PIK3CA on H&E-stained WSIs from TCGA breast cancer (BRCA) cases ($n = 1,034$), each site-stratified and label-stratified in an approximate train/val/test splits (60% / 20% / 20%). Additionally, we perform 10-fold cross-validation on breast cancer core needle biopsies (BCNB, $n = 1,058$) (Xu et al., 2021) for ER, PR, and HER2. We evaluate performance using AUROC.

**GBMLGG mutational subtyping** (Brennan et al., 2013; Roetzer-Pejrimovsky et al., 2022): These tasks include binary coarse-grained mutation prediction of IDH1 status using the TCGA GBMLGG dataset (1,123 slides), and 5-class fine-grained histomolecular subtyping. The 5-class histomolecular subtyping task was separated into the categories of Astrocytoma, IDH1-mutant, Glioblastoma, IDH1-mutant, Oligodendroglioma, IDH1-mutant and 1p/19q codeleted, Astrocytoma, IDH1-wildtype, and Glioblastoma, IDH1-wildtype. For training and evaluation of both tasks, we use the UNI splits, which label-stratified TCGA-GBMLGG into a train/val/test fold with a 47:22:31 ratio. Additionally, we perform external validation on the held-out EBRAINS cohort ($n = 873$ slides) for the cases with known IHD1 status. We evaluate GBMLGG-C with AUROC, and GBMLGG-F with balanced accuracy.

### A4.6 SOFT MoE PATCH OUTPUT FORMULATION

Here, we describe the process for returning updated patch representations according to Soft MoE (Puigcerver et al., 2024) for the model design ablation **MAMMOTH output**. Let $\{\bar{\mathbf{x}}_i\}_{i=1}^{N}$ be the set of patch embeddings and $\{\mathbf{z}_j^{(k)}\}_{j,k=1}^{S \cdot E}$ be the slot outputs for MAMMOTH across $H$ heads, $E$ experts, and $S$ slots per expert. The linear weights are normalized weighted combination over the routing scores of each slot, where for any head, the weight between patch $i$ and the output of expert $k$, slot $j$ is given by:

$$\alpha_{j,i}^{(k)} = \frac{\exp(\langle \bar{x}_i, s_j^{(k)} \rangle)}{\sum_{k=1}^{E} \sum_{j=1}^{S} \exp(\langle \bar{x}_i, s_j^{(k)} \rangle)}, \tag{}$$

and the updated representation $\hat{x}_i$ is the weighted combination

$$\hat{x}_i = \sum_{j,k=1}^{S,E} \alpha_{j,i}^{(k)} \mathbf{z}_j^{(k)}. \tag{}$$

## A5 INTERPRETABILITY AND VISUALIZATION PROTOCOLS

### A5.1 DETERMINISTIC EXPERT-SLOT SELECTION PROTOCOL

To ensure rigorous interpretability of MAMMOTH, we implemented a fully deterministic protocol to identify and visualize the semantic specialization of expert slots. This process aligns histological concepts with model routing behaviors through the following steps:

1. **Key Term Generation:** A set of $N = 30$ organ-specific histologic terms was generated using the Gemini 2.5 Pro large language model.
2. **Vision-Language Embedding:** Text embeddings for each histologic term, $x_{\text{text}}$, were generated using the MUSK vision-language pathology foundation model Xiang et al. (2025). Similarly, patch-level image embeddings, $X = \{x_i\}_{i=1}^N$, were extracted for all Whole Slide Images (WSIs) in the evaluation dataset using the MUSK image encoder.
3. **Semantic Relatedness Scoring:** For every patch in the dataset, we computed a "relatedness" score to each histologic term via the cosine similarity between the patch embedding and the term embedding:
$$S(p, t) = \cos(x_i, x_{\text{text}}^{(t)}) \qquad ()$$
4. **Routing-Weighted Attribution:** To determine the specialization of specific expert slots, we calculated a weighted attribution score. For each expert-slot pair and histologic term, we computed the sum of the semantic relatedness scores ($S$), weighted by the router's probability assignment (routing score) of that patch to the specific slot.
5. **Selection and Visualization:** For each histologic term, the expert-slot pair yielding the highest accumulated weighted score was selected for visualization.

### A5.2 INSTANCE GRADIENT INTERFERENCE PROTOCOL

To empirically motivate the use of specialized experts, we analyze the alignment of gradient updates across different instance types. We randomly sample $M = 100$ Whole Slide Images (WSIs) and partition their constituent patch embeddings $\{x_i\}_{i=1}^N$ into $K$ clusters using $k$-means clustering. From each cluster, we select 100 representative instances per slide to form a balanced evaluation set.

Let $\mathcal{L}$ be the task-specific loss and $\theta$ represent the weights of the layer under evaluation. We compute the instance-level gradient $\mathbf{g}_i = \nabla_\theta \mathcal{L}(\mathbf{x}_i)$ for each sampled instance. To ensure a controlled comparison across architectures, all gradients are captured at the first training epoch without updating model parameters. This ensures that $\theta$ remains identical for all $\mathbf{g}_i$ within a trial, making the updates directly comparable. The cosine similarity $S$ between two instances $i$ and $j$ is defined as:

$$S(i, j) = \frac{\mathbf{g}_i \cdot \mathbf{g}_j}{\|\mathbf{g}_i\|_2 \|\mathbf{g}_j\|_2}, \qquad ()$$

where $\cdot$ refers to the inner product between two gradients.

**Baseline Linear Layer and Single-Expert Variant** For the standard MIL framework, we evaluate $\theta$ with respect to the linear layer $f_{\text{MIL}}^{\text{linear}}$. For the single-expert variant of MAMMOTH, we evaluate the gradients with respect to the expert-specific projection weights $\mathbf{W}_{\text{low}}^{(e)}$ where $E = 1$. This variant serves as a baseline to isolate the effect of routing from the underlying architecture change. We report the mean similarity for intra-cluster pairs (where instances $i, j$ belong to the same $k$-means cluster, with $k = 8$) versus inter-cluster pairs (where $i, j$ belong to different clusters):

$$\bar{S}_{\text{type}} = \mathbb{E}_{i,j\sim\text{type}}[S(i, j)], \quad \text{where type} \in \{\text{intra, inter}\} \qquad ()$$

**Multi-Expert Routing** Finally, for MAMMOTH with $E = 30$ experts, we examine the gradients of the specific expert layers $\mathbf{W}_{\text{low}}^{(e)}$. We report the average gradient similarity between all instances assigned to the same expert $e$ via the routing mechanism. By comparing this to the single-expert baseline, we demonstrate how instance-level routing promotes gradient homogeneity within each specialized module, facilitating more effective learning of phenotype-specific features.

## A5.3 ADDITIONAL VISUALIZATIONS

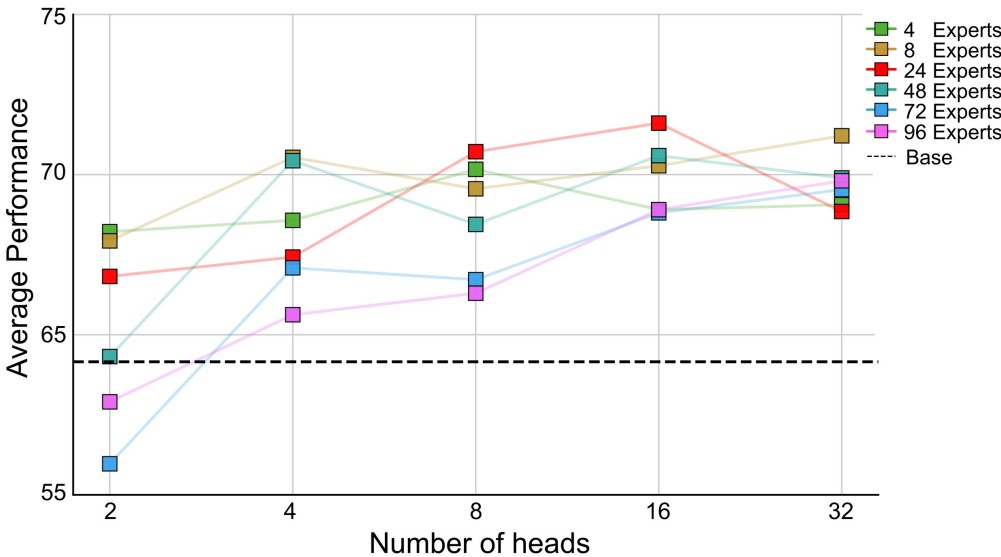

Figure A1: **Performance with varying heads and experts**. ABMIL and TransMIL performance with MAMMOTH and varying numbers of heads and experts, averaged across EBRAINS-C and 3-fold cross-validation of LUNG TP53 and BRCA HER2. Performance is most stable with intermediate number of experts and high number of heads.

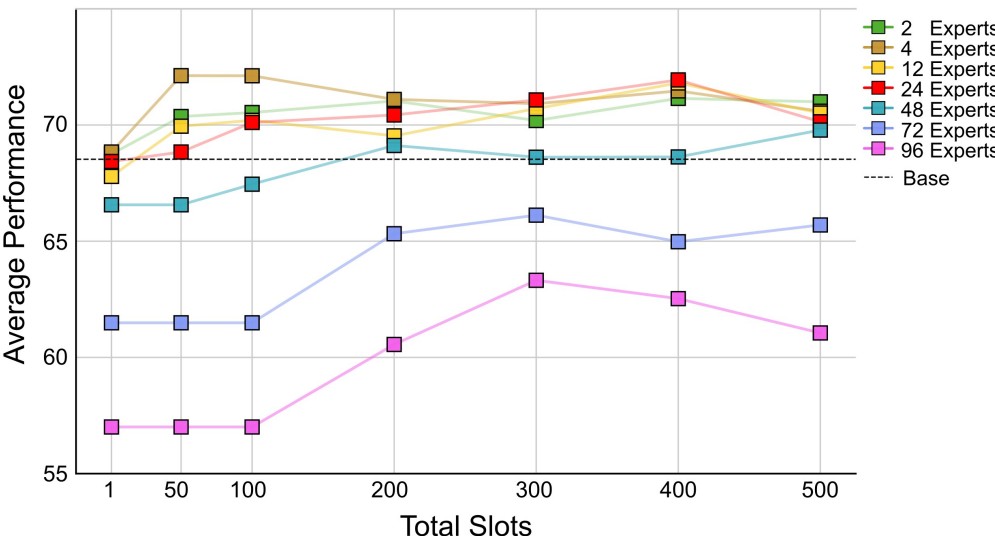

Figure A2: **Performance with varying total slots**. ABMIL trained on EBRAINS-F with varying experts and total slots. Results are averaged across head counts $H \in \{2, 4, 8, 16, 32\}$. Slots per expert are set as $S = \lfloor \frac{\text{Total Slots}}{E} \rfloor$. Low expert counts ($E \in \{2, 4\}$) reach highest performance with low total slots, while high expert counts ($E \in \{2, 4\}$) reach highest performance with 200-400 total slots.

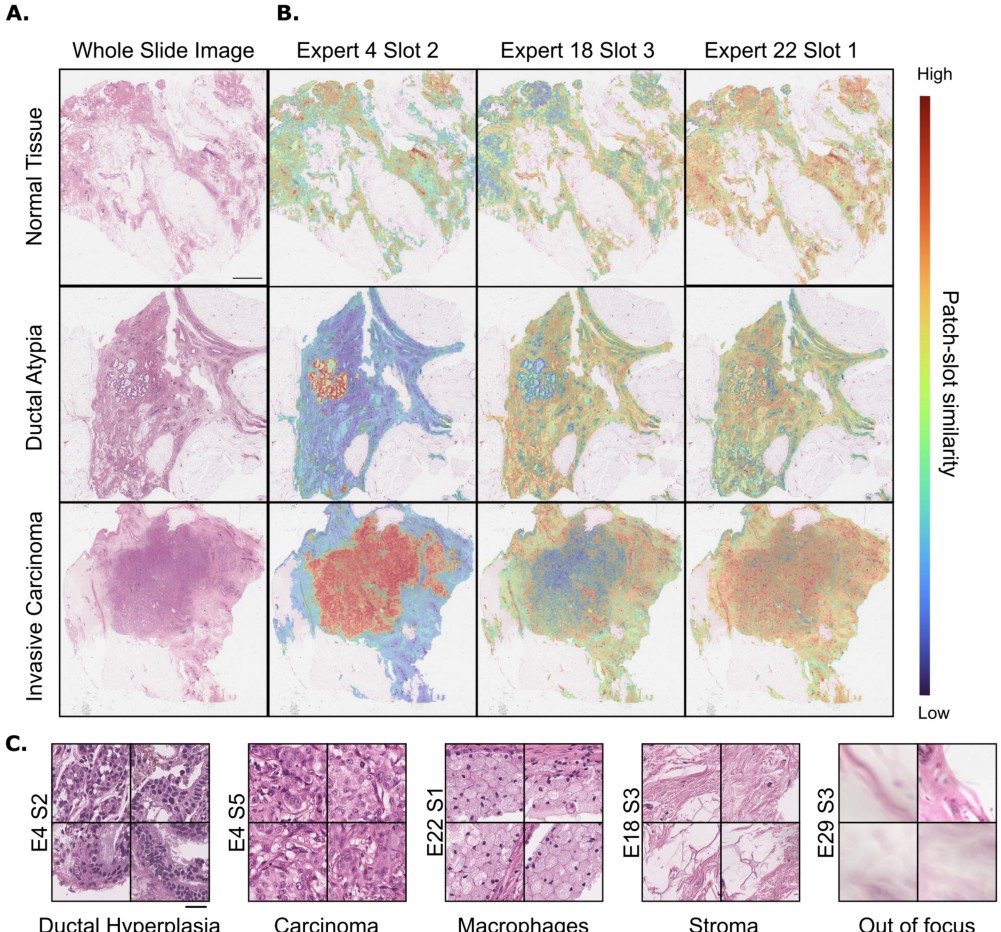

Figure A3: **Slot routing scores on BRACS subtyping**. Slot routing scores for an ABMIL model trained on BRACS coarse-grained subtyping. **(A)** Whole slide image for normal tissue, ductal atypia, and invasive carcinoma. **(B)** Softmax-normalized routing scores between each patch and slots of different experts. Expert 4 Slot 2 (E4 S2) places high routing scores on the key diagnostic regions for normal tissue, ductal atypia, and invasive carcinoma. Stroma had high routing scores allocated to Expert 18 Slot 3. Expert 22 Slot 1 (E22 S1) had diffusely distributed routing scores throughout the tissue. **(C)** Patches from slides in **(A)** with the highest routing scores for select expert-slot pairs. Top patches routed to different slots have clear morphological phenotypes: the top patches for E4 S2 contain diagnostically relevant cells with ductal hyperplasia, and the top patches for E4 S5 contains invasive carcinoma, while the top patches for E18 S3 consist primarily of stroma, those of E22 S1 consist of macrophages, and those of E29 S3 consist of blurry tissue. Scale bars: **A-B.** 500 $\mu$m, **C.** 20 $\mu$m.

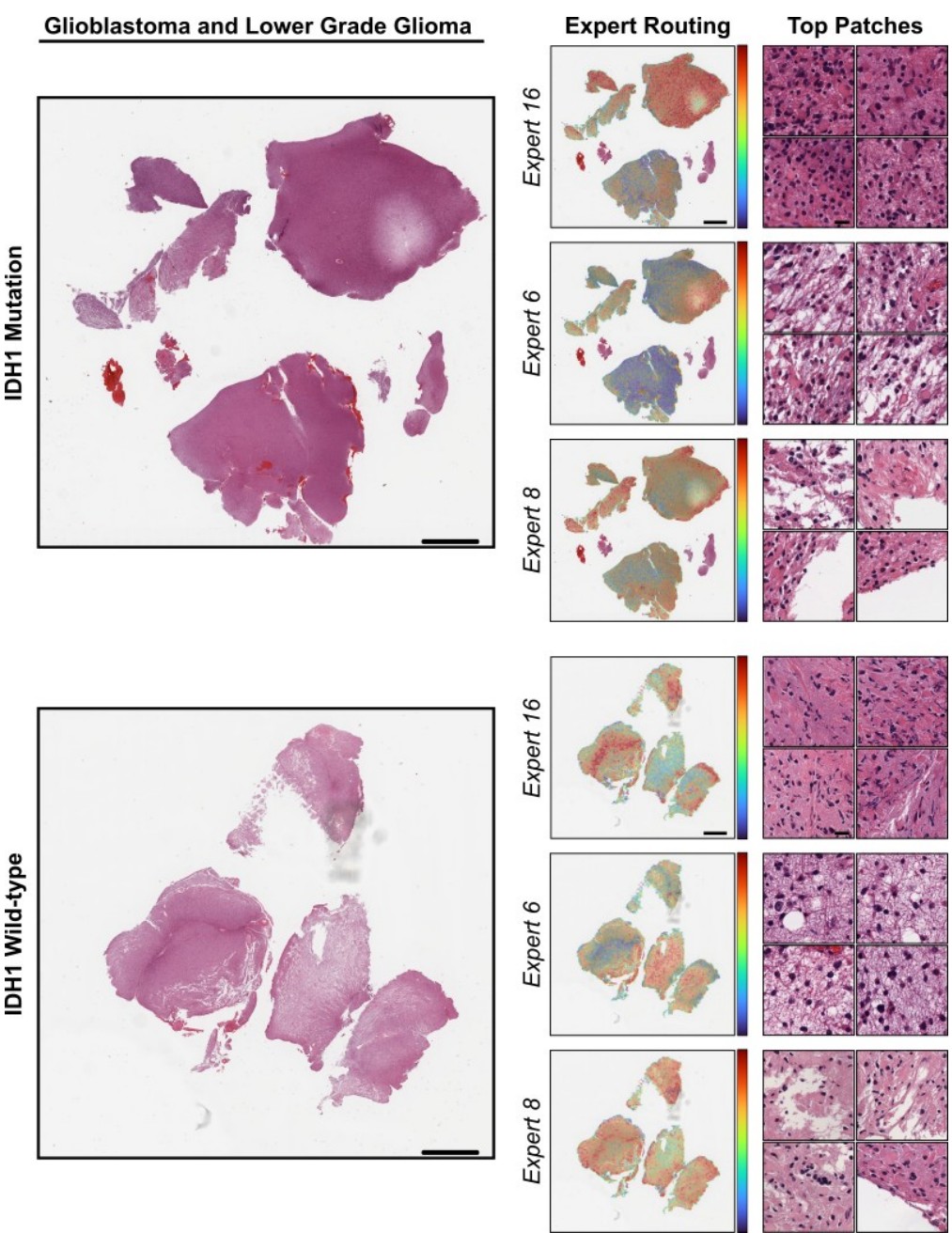

Figure A4: **Expert routing scores in GBMLGG C.** Total routing scores from each patch to each expert, averaged across the slots and heads of each expert. In both mutant and wild-type IDH1 WSIs, we find that Expert 16 specializes in dense tumor cells with tightly-packed neuropil. Expert 6 specializes in dense tumor cells with loose neuropil. Expert 8 specializes in diffuse tumor cells with loose neuropil. Scale bars: WSI; 500 $\mu$m, Expert Routing; 500 $\mu$m, Top Patches; 10 $\mu$m

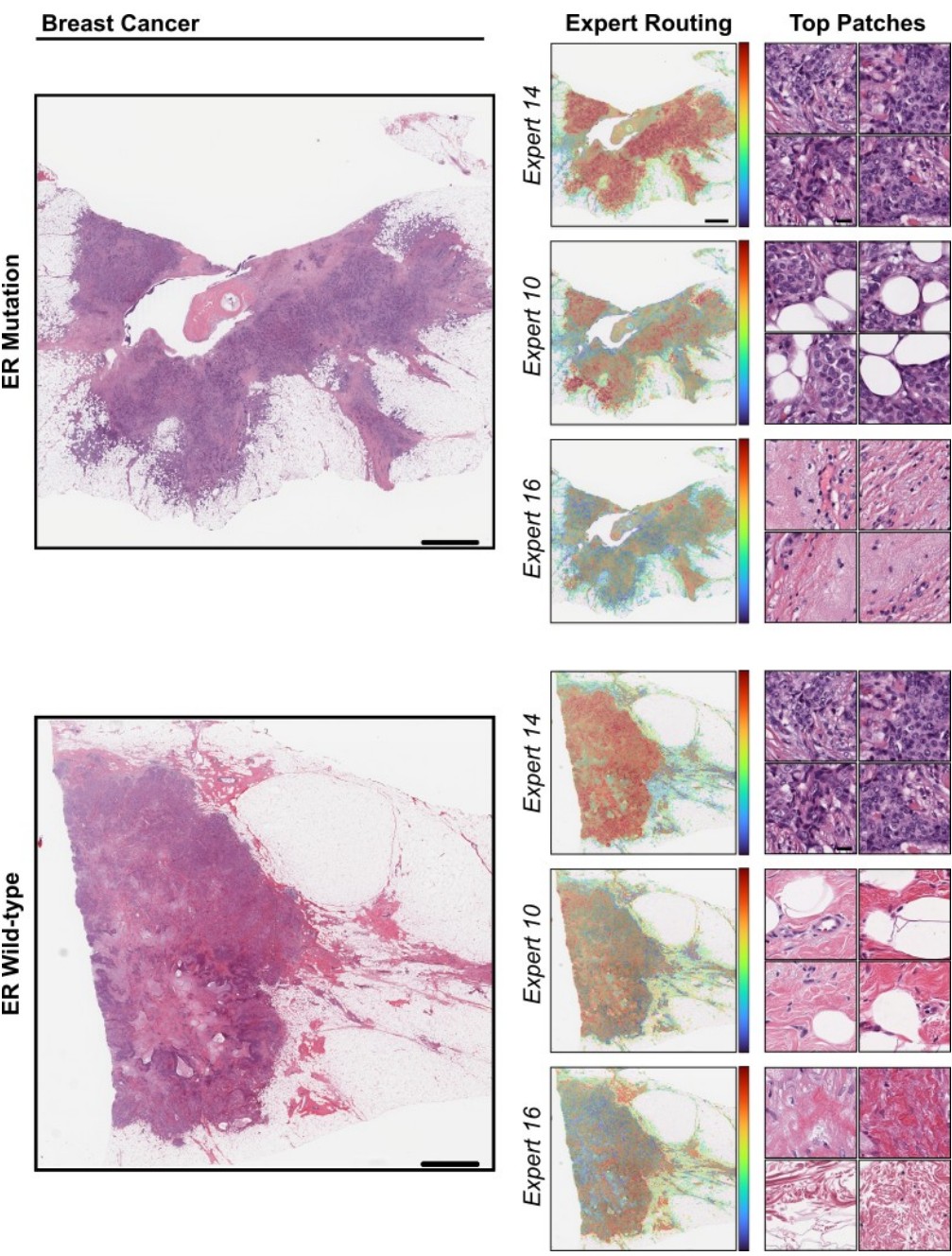

Figure A5: **Expert routing scores in BRCA ER.** Total routing scores from each patch to each expert, averaged across the slots and heads of each expert. In both mutant and wild-type IDH1 WSIs, we find that Expert 14 specializes in patches rich in tumor cells, Expert 10 specializes in adipocytes in conjunction with tumor cells, and Expert 16 specializes in connective tissue. Scale bars: WSI; 500 $\mu$m, Expert Routing; 500 $\mu$m, Top Patches; 10 $\mu$m

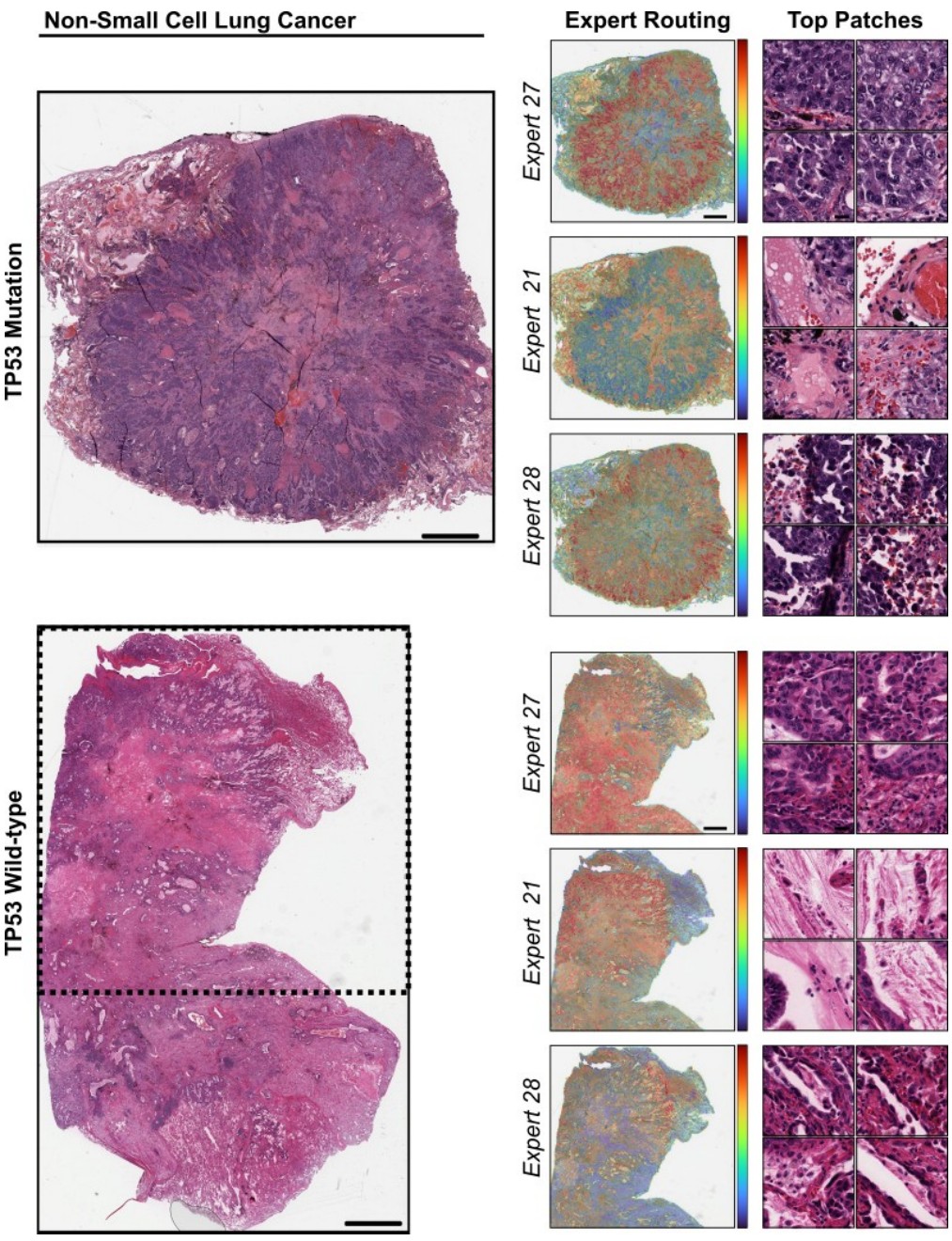

Figure A6: **Expert routing scores in LUNG TP53.** Total routing scores from each patch to each expert, averaged across the slots and heads of each expert. We find that Expert 27 specializes in processing patches rich in tumor cells. Expert 21 specializes in background structures such as blood vessels, lymphatics, and connective tissue. Expert 28 specializes in tumor cells around or forming spaces. Dashed box indicates ROI dispalyed in Expert Routing. Scale bars: WSI; 500 $\mu$m, Expert Routing; 500 $\mu$m, Top Patches; 10 $\mu$m

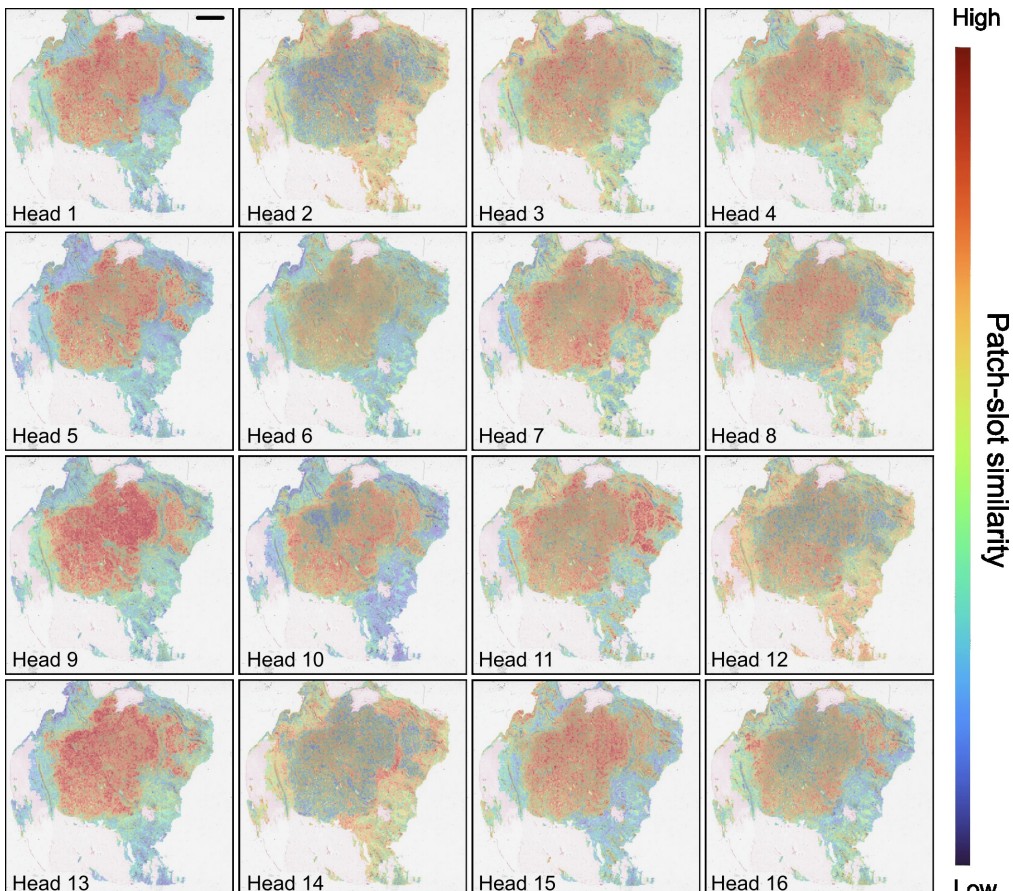

Figure A7: **Routing scores on BRACS subtyping across heads of one slot**. Routing scores for a single slot (Expert 4 Slot 2) across each head of a 16-head MAMMOTH ABMIL model trained on BRACS coarse-grained subtyping. Each image corresponds to the routing scores within one head. The image shown corresponds to invasive carcinoma. We observe that while the attention scores are routed to the same general tumor area between different heads, the distribution of attenton scores varies between heads, suggesting that different heads may attend to different details of the tumor regions. Scale bars: 500 $\mu$m.

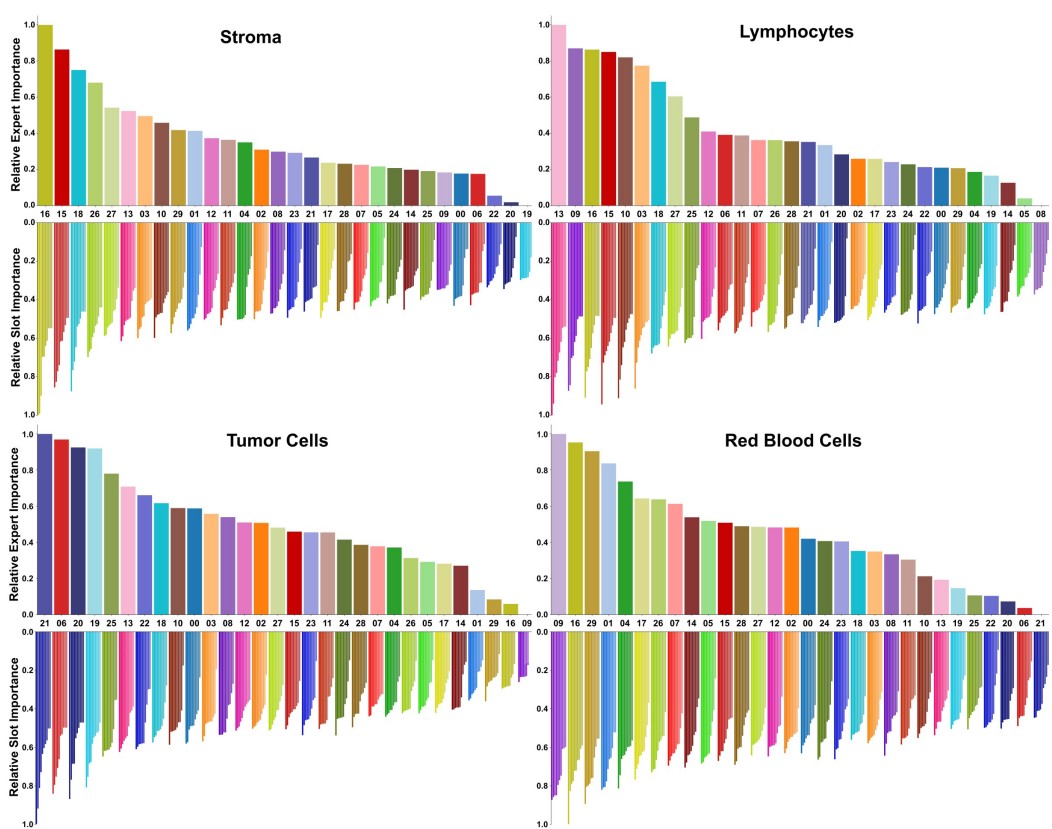

Figure A8: **Quantifying expert specialization via vision–language scoring.** To decode expert roles, we annotated all NSCLC patches with **term-relevance scores**, based on the cosine similarity between image embeddings and text concepts such as *stroma* or *tumor cells* using the MUSK pathology foundation model Xiang et al. (2025). Higher scores indicate more accurate textual description of the corresponding morphology. We then calculated the **Relative Slot Importance** for each concept by weighting these MUSK relevance scores with the slot's own routing weights across all patches. Finally, **Relative Expert Importance** represents the aggregated importance of all slots assigned to a given expert. The distribution highlights distinct specialization: Experts 16, 13, 21, and 9 demonstrate high attention to stroma, lymphocytes, tumor cells, and red blood cells, respectively. Furthermore, slots within the same expert display highly correlated routing patterns, indicating strong intra-expert synchrony. Representative patches with the highest routing scores were validated by pathologists to confirm alignment with these histological concepts. The slot with the highest relative slot importance is selected for downstream visualization with attention heatmaps.

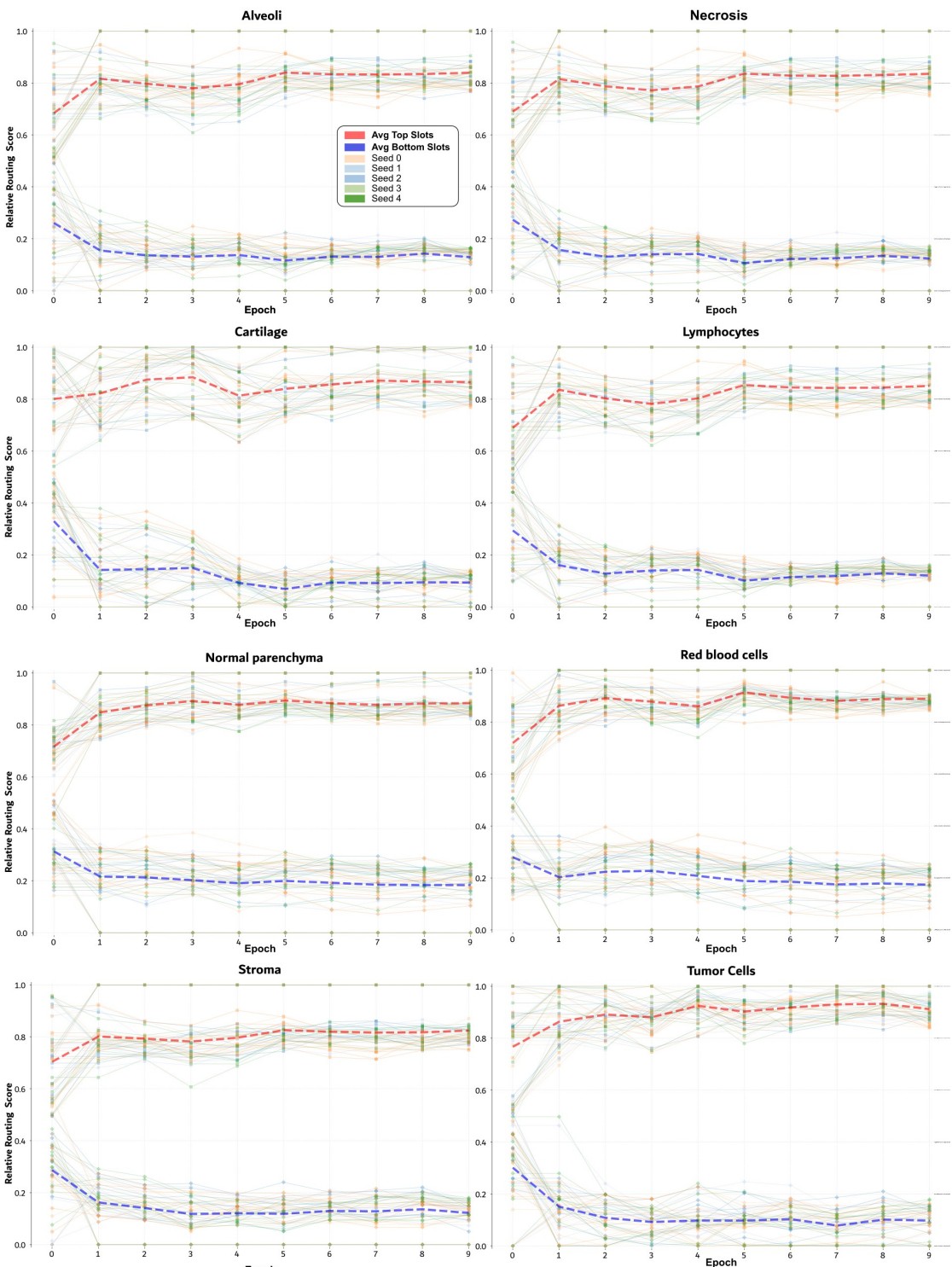

Figure A9: **Emergent specialization during training**. We tracked the **Relative Routing Scores**, derived from MUSK term-relevance scores weighted by routing probability, across NSCLC sub-typing training. Higher relative routing score implies concept-aligned slots. The plots display the trajectories of the top ten most relevant slots (red) and least relevant slots (blue), identified at model convergence and traced back to initialization. Data is averaged over five random seeds with normal-ization across epochs and seeds. Dashed lines indicate group averages. We observe that **concept-aligned slots exhibit higher relative importance even at initialization** (epoch 0), followed by a sharp increase and stabilization within the first epoch. This suggests that MAMMOTH's specializa-tion is partly driven by differential routing at initialization, which is rapidly reinforced during early training.

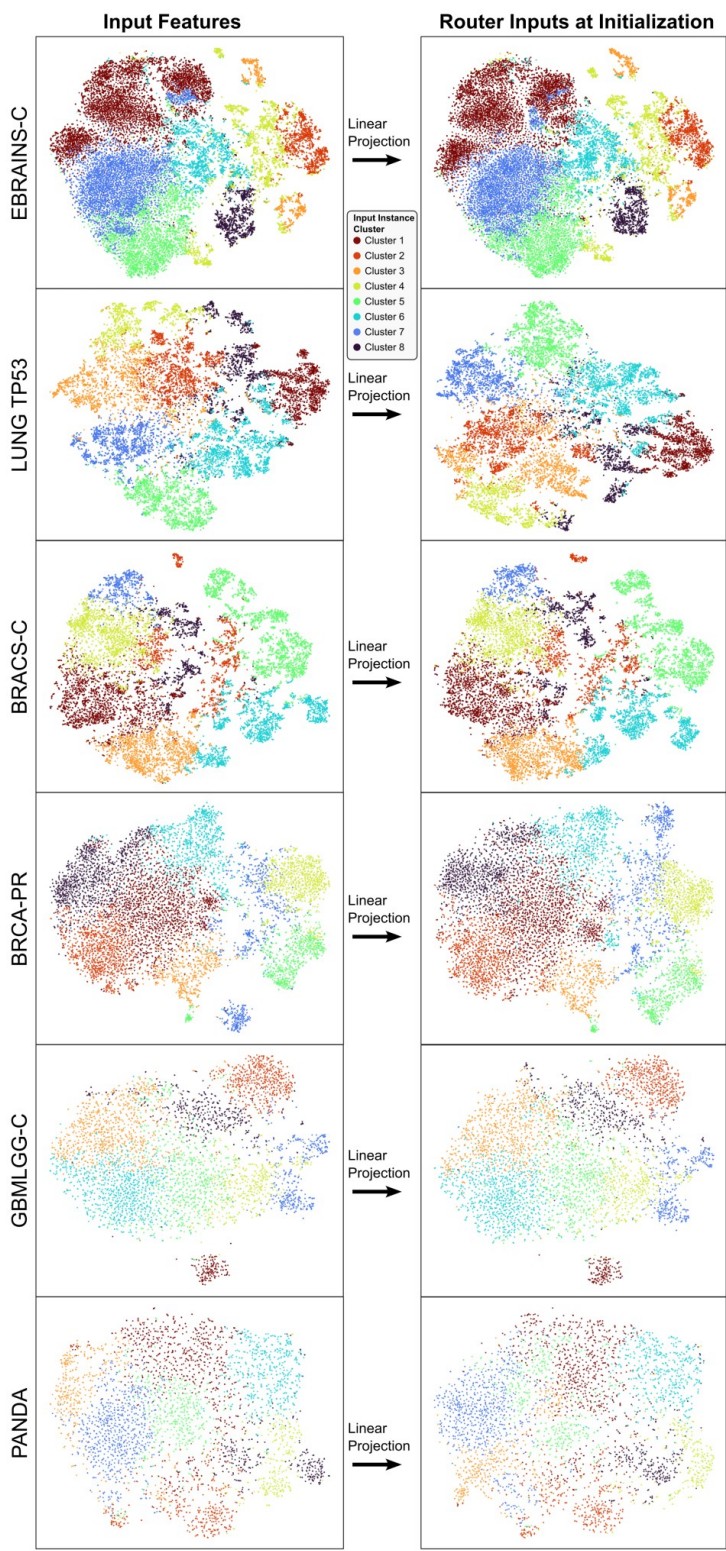

Figure A10: **Preservation of clusters after linear projection**. t-SNE of the instance features before and after MAMMOTH linear projection via $\mathbf{W} \in \mathbb{R}^{(P \cdot H) \times D}$ at the start and end of training. Instances are colored according to K-means clustering of the *input features* with $K = 8$. The linear projection preserves instance clusters encoded by the pathology foundation model, both at random initialization and at model convergence.

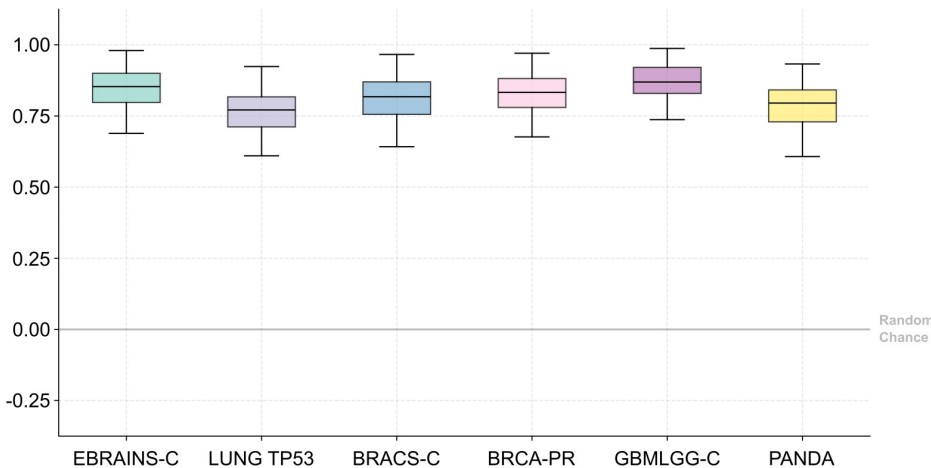

Figure A11: **Quantitative assessment of cluster preservation via Adjusted Rand Index (ARI)**. We quantify instance-level similarity after randomly-initialized linear projection, utilizing K-means cluster assignments ($K = 8$) of the input features as reference labels. An ARI score of 0.0 indicates random assignment. Across all 6 evaluated tasks, our method consistently achieves ARI scores exceeding 0.75, demonstrating that the randomly initialized linear projection reliably preserves semantic structure encoded by the pathology foundation model.

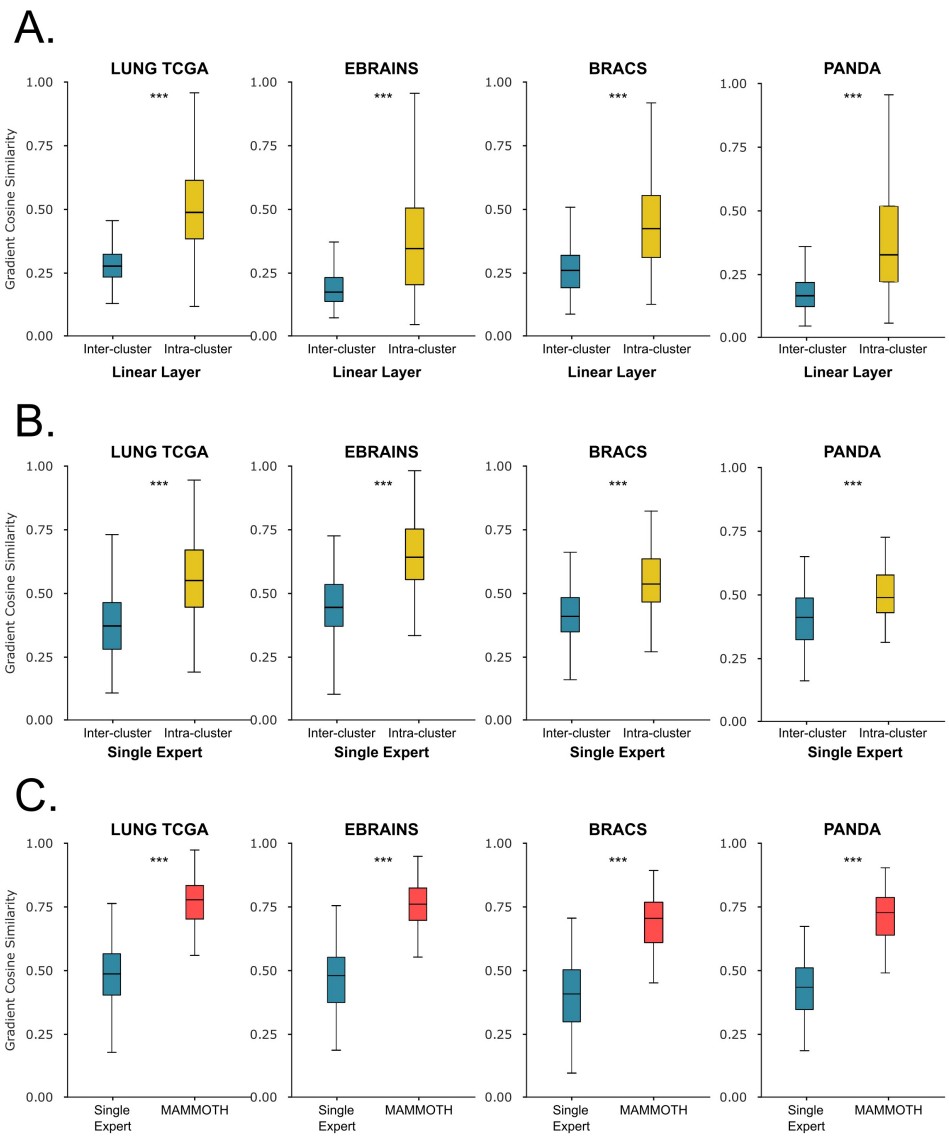

Figure A12: **Cosine similarity of gradient updates across instance clusters and architectures**. **A.** Standard linear layers and **B.** MAMMOTH (single-expert) exhibit significantly higher gradient similarity within clusters than across clusters, confirming that heterogeneous patch types yield conflicting gradient updates. **C.** Comparison of average gradient similarity between MAMMOTH with a single-expert baseline and with 30-expert instance-level routing. For the multi-expert model, similarity is calculated between instances assigned to the same expert and then averaged across all experts. **By routing similar instances to specialized experts, the internal gradient updates become more homogeneous compared to the single-expert counterpart**, leading to more stable and specialized feature learning. Clusters were defined per WSI by $k$-means on UNI features with $k = 8$. For each task, gradient similarity was computed per WSI by sampling 100 instances per cluster per WSI and averaging across 100 randomly sampled WSIs. More details can be found in Section A5.2. *Significance: *$p < 0.05$, **$p < 0.01$, **$p < 0.001$ (one-sided t-test).

Table A1: **Performance on different encoders**. Performance of MIL models on different encoders. Models were trained with 30 experts, 16 heads, and 6 slots per expert. Addition of MAMMOTH consistently leads to improved performance over the original MIL models across all three encoders. Balanced accuracy is reported.

| Task | State | ABMIL | | | CLAM | | | TransMIL | | | Max | | |
|---|---|---|---|---|---|---|---|---|---|---|---|---|---|
| | | GigaPath | Musk | Virchow | GigaPath | Musk | Virchow | GigaPath | Musk | Virchow | GigaPath | Musk | Virchow |
| EBRAINS-C | Base | 85.9 | 85.5 | 83.4 | 87.0 | 79.1 | 83.9 | 87.3 | 80.2 | 85.5 | 83.9 | 82.8 | 83.3 |
| $C = 12$ | +Ours | 83.6 | 84.2 | 87.9 | 89.9 | 88.8 | 87.1 | 87.8 | 82.3 | 86.6 | 87.2 | 82.7 | 83.6 |
| (Bal. Acc.) | Δ | -2.3 | -1.3 | +4.5 | +2.9 | +9.7 | +3.2 | +0.5 | +2.1 | +1.1 | +3.3 | -0.1 | +0.3 |
| EBRAINS-F | Base | 67.9 | 67.1 | 65.5 | 70.5 | 68.3 | 69.8 | 67.6 | 64.7 | 66.7 | 61.1 | 61.0 | 65.5 |
| $C = 30$ | +Ours | 69.7 | 69.5 | 70.9 | 71.6 | 72.1 | 72.3 | 71.8 | 62.9 | 65.5 | 70.6 | 65.7 | 68.0 |
| (Bal. Acc.) | Δ | +1.8 | +2.4 | +5.4 | +1.1 | +3.8 | +2.5 | +4.2 | -1.8 | -1.2 | +9.5 | +4.7 | +2.5 |
| BRACS-C | Base | 69.4 | 60.9 | 73.5 | 58.9 | 53.7 | 49.3 | 59.3 | 65.5 | 63.3 | 57.5 | 55.5 | 56.3 |
| $C = 3$ | +Ours | 69.2 | 70.7 | 71.8 | 66.9 | 73.0 | 72.3 | 70.5 | 66.6 | 66.9 | 60.4 | 57.0 | 67.2 |
| (Bal. Acc.) | Δ | -0.2 | +9.8 | -1.7 | +8.0 | +19.3 | +23.0 | +11.2 | +1.1 | +3.6 | +2.9 | +1.5 | +10.9 |
| BRACS-F | Base | 40.2 | 43.6 | 45.6 | 28.6 | 28.9 | 30.2 | 38.7 | 37.7 | 30.2 | 32.7 | 30.4 | 34.8 |
| $C = 7$ | +Ours | 43.6 | 43.6 | 45.7 | 43.7 | 43.5 | 43.1 | 42.6 | 41.7 | 37.5 | 36.1 | 44.3 | 39.7 |
| (Bal. Acc.) | Δ | +3.4 | +0.0 | +0.1 | +15.1 | +14.6 | +12.9 | +3.9 | +4.0 | +7.3 | +3.4 | +13.9 | +4.9 |

Table A2: **Molecular biomarker prediction** Change in performance between baseline MIL models and after the addition of MAMMOTH for 13 molecular biomarker prediction tasks. All tasks are binary prediction, with AUROC reported, with the exception of gbmlgg fine, which is a 7 class histomolecular classification task with balanced accuracy as the reported metric. Performance on GBMLGG is averaged between the internal TCGA cohort and external EBRAINS cohort. Standard deviation is reported according to 1,000 boostrapped trials.

| Task | Status | ABMIL | CLAM | TransMIL | Transf. | ILRA | MeanMIL | MaxMIL | DSMIL | Average |
|---|---|---|---|---|---|---|---|---|---|---|
| BCNB ER | Base | $90.38_{(0.2)}$ | $91.22_{(0.7)}$ | $91.08_{(0.4)}$ | $90.35_{(0.5)}$ | $89.80_{(0.4)}$ | $90.77_{(0.3)}$ | $90.07_{(0.6)}$ | $88.84_{(0.8)}$ | 90.31 |
| | +Ours | $92.25_{(0.1)}$ | $92.72_{(0.2)}$ | $92.15_{(0.1)}$ | $92.02_{(0.1)}$ | $91.58_{(0.2)}$ | $92.19_{(0.1)}$ | $91.33_{(0.0)}$ | $90.93_{(0.1)}$ | 91.90 |
| | Δ | +1.87 | +1.50 | +1.06 | +1.67 | +1.78 | +1.42 | +1.26 | +2.09 | +1.58 |
| BCNB HER2 | Base | $73.05_{(0.4)}$ | $73.91_{(0.1)}$ | $68.90_{(0.7)}$ | $69.24_{(0.3)}$ | $71.38_{(0.6)}$ | $73.46_{(0.2)}$ | $74.33_{(0.5)}$ | $72.09_{(0.5)}$ | 72.04 |
| | +Ours | $74.70_{(0.4)}$ | $76.64_{(0.1)}$ | $71.40_{(0.1)}$ | $75.27_{(0.2)}$ | $74.34_{(0.3)}$ | $76.35_{(0.3)}$ | $75.56_{(0.0)}$ | $73.39_{(0.1)}$ | 74.71 |
| | Δ | +1.65 | +2.73 | +2.50 | +6.02 | +2.97 | +2.89 | +1.23 | +1.30 | +2.66 |
| BCNB PR | Base | $82.48_{(0.4)}$ | $84.16_{(0.3)}$ | $82.30_{(0.8)}$ | $81.49_{(0.5)}$ | $81.90_{(0.6)}$ | $83.83_{(0.1)}$ | $84.34_{(0.5)}$ | $82.96_{(0.4)}$ | 82.93 |
| | +Ours | $85.84_{(0.5)}$ | $85.59_{(0.2)}$ | $85.37_{(0.5)}$ | $83.92_{(0.3)}$ | $83.88_{(0.5)}$ | $84.84_{(0.2)}$ | $85.12_{(0.4)}$ | $83.88_{(0.2)}$ | 84.80 |
| | Δ | +3.36 | +1.42 | +3.07 | +2.43 | +1.98 | +1.02 | +0.78 | +0.92 | +1.87 |
| BRCA ER | Base | $86.93_{(0.3)}$ | $86.46_{(0.4)}$ | $87.38_{(0.3)}$ | $85.61_{(0.9)}$ | $85.00_{(0.4)}$ | $86.18_{(0.3)}$ | $86.84_{(0.3)}$ | $87.46_{(0.3)}$ | 86.48 |
| | +Ours | $87.94_{(0.3)}$ | $90.06_{(0.3)}$ | $88.59_{(0.1)}$ | $88.26_{(0.7)}$ | $87.01_{(0.3)}$ | $88.27_{(0.3)}$ | $87.65_{(0.0)}$ | $86.75_{(0.5)}$ | 88.07 |
| | Δ | +1.01 | +3.60 | +1.20 | +2.65 | +2.01 | +2.10 | +0.81 | -0.72 | +1.58 |
| BRCA HER2 | Base | $64.35_{(1.1)}$ | $64.38_{(0.9)}$ | $61.31_{(1.5)}$ | $65.25_{(1.2)}$ | $61.80_{(1.4)}$ | $62.59_{(1.0)}$ | $63.58_{(2.6)}$ | $60.90_{(0.6)}$ | 63.02 |
| | +Ours | $68.35_{(0.8)}$ | $61.84_{(0.1)}$ | $64.71_{(0.8)}$ | $64.84_{(0.1)}$ | $63.40_{(1.0)}$ | $67.59_{(0.6)}$ | $65.42_{(0.4)}$ | $65.94_{(0.7)}$ | 65.26 |
| | Δ | +4.01 | -2.54 | +3.40 | -0.41 | +1.60 | +5.00 | +1.83 | +5.04 | +2.24 |
| BRCA PIK3CA | Base | $60.23_{(0.7)}$ | $59.15_{(0.7)}$ | $58.79_{(1.9)}$ | $57.43_{(0.9)}$ | $58.90_{(1.3)}$ | $60.23_{(1.2)}$ | $61.67_{(1.1)}$ | $61.30_{(0.7)}$ | 59.71 |
| | +Ours | $59.55_{(0.5)}$ | $58.38_{(0.2)}$ | $61.30_{(0.0)}$ | $60.27_{(0.5)}$ | $59.22_{(0.9)}$ | $58.99_{(0.8)}$ | $60.22_{(0.6)}$ | $60.96_{(0.2)}$ | 59.86 |
| | Δ | -0.68 | -0.77 | +2.50 | +2.84 | +0.32 | -1.24 | -1.45 | -0.35 | +0.15 |
| BRCA PR | Base | $76.37_{(0.3)}$ | $77.73_{(0.7)}$ | $78.02_{(1.2)}$ | $77.12_{(0.3)}$ | $75.91_{(0.8)}$ | $76.36_{(0.5)}$ | $77.79_{(0.2)}$ | $78.00_{(0.8)}$ | 77.16 |
| | +Ours | $78.77_{(0.5)}$ | $78.80_{(0.6)}$ | $79.07_{(0.8)}$ | $79.44_{(0.1)}$ | $77.10_{(0.7)}$ | $79.54_{(0.9)}$ | $78.21_{(0.2)}$ | $79.05_{(0.2)}$ | 78.75 |
| | Δ | +2.39 | +1.07 | +1.05 | +2.32 | +1.19 | +3.18 | +0.43 | +1.06 | +1.59 |
| GBMLGG-C | Base | $91.82_{(0.4)}$ | $94.38_{(0.5)}$ | $94.46_{(0.1)}$ | $93.41_{(1.0)}$ | $93.72_{(0.4)}$ | $94.34_{(0.2)}$ | $95.34_{(1.0)}$ | $94.88_{(0.4)}$ | 94.04 |
| | +Ours | $96.19_{(0.4)}$ | $94.53_{(0.1)}$ | $95.74_{(0.0)}$ | $95.68_{(0.2)}$ | $93.47_{(0.4)}$ | $95.34_{(0.7)}$ | $95.54_{(0.3)}$ | $94.80_{(0.4)}$ | 95.16 |
| | Δ | +4.37 | +0.15 | +1.28 | +2.27 | -0.25 | +1.00 | +0.20 | -0.08 | +1.12 |
| GBMLGG-F | Base | $51.89_{(1.3)}$ | $49.78_{(1.3)}$ | $52.19_{(1.6)}$ | $50.58_{(1.9)}$ | $49.57_{(1.4)}$ | $49.68_{(0.9)}$ | $50.31_{(0.8)}$ | $49.53_{(2.4)}$ | 50.44 |
| | +Ours | $52.22_{(1.7)}$ | $51.03_{(0.3)}$ | $50.28_{(1.4)}$ | $53.12_{(0.4)}$ | $52.53_{(0.8)}$ | $51.63_{(1.2)}$ | $51.71_{(0.6)}$ | $50.68_{(0.7)}$ | 51.65 |
| | Δ | +0.33 | +1.25 | -1.91 | +2.55 | +2.97 | +1.95 | +1.39 | +1.15 | +1.21 |
| LUNG EGFR | Base | $61.27_{(1.2)}$ | $65.85_{(0.6)}$ | $63.66_{(1.2)}$ | $60.20_{(1.8)}$ | $62.00_{(2.7)}$ | $64.42_{(0.7)}$ | $65.43_{(4.1)}$ | $63.93_{(1.6)}$ | 63.35 |
| | +Ours | $63.68_{(1.2)}$ | $65.98_{(0.8)}$ | $65.30_{(2.3)}$ | $67.55_{(1.2)}$ | $62.57_{(1.2)}$ | $66.17_{(1.1)}$ | $64.12_{(1.3)}$ | $65.51_{(1.3)}$ | 65.11 |
| | Δ | +2.42 | +0.13 | +1.64 | +7.35 | +0.57 | +1.74 | -1.31 | +1.59 | +1.77 |
| LUNG KRAS | Base | $58.06_{(0.7)}$ | $60.81_{(0.7)}$ | $60.31_{(1.1)}$ | $58.22_{(1.5)}$ | $60.10_{(0.9)}$ | $60.88_{(1.2)}$ | $56.93_{(0.6)}$ | $59.21_{(0.4)}$ | 59.31 |
| | +Ours | $59.40_{(1.5)}$ | $59.42_{(0.8)}$ | $61.20_{(0.1)}$ | $59.45_{(0.3)}$ | $61.31_{(0.2)}$ | $61.22_{(0.5)}$ | $61.35_{(0.4)}$ | $58.10_{(0.7)}$ | 60.18 |
| | Δ | +1.34 | -1.39 | +0.89 | +1.23 | +1.21 | +0.35 | +4.43 | -1.10 | +0.87 |
| LUNG STK11 | Base | $76.41_{(1.1)}$ | $65.75_{(0.9)}$ | $68.95_{(2.2)}$ | $71.14_{(0.3)}$ | $69.10_{(1.8)}$ | $67.35_{(1.0)}$ | $70.06_{(2.6)}$ | $65.65_{(1.4)}$ | 69.30 |
| | +Ours | $74.36_{(1.5)}$ | $70.57_{(0.5)}$ | $66.44_{(1.1)}$ | $69.39_{(0.6)}$ | $68.10_{(0.7)}$ | $74.31_{(0.7)}$ | $73.48_{(0.3)}$ | $68.61_{(0.4)}$ | 70.66 |
| | Δ | -2.05 | +4.81 | -2.51 | -1.75 | -1.00 | +6.96 | +3.41 | +2.96 | +1.36 |
| LUNG TP53 | Base | $72.43_{(1.1)}$ | $73.04_{(0.3)}$ | $68.07_{(0.7)}$ | $70.19_{(1.1)}$ | $69.89_{(0.8)}$ | $72.29_{(0.3)}$ | $69.57_{(0.9)}$ | $71.31_{(0.8)}$ | 70.85 |
| | +Ours | $76.20_{(0.7)}$ | $71.60_{(0.1)}$ | $70.86_{(0.5)}$ | $73.46_{(0.9)}$ | $69.29_{(0.6)}$ | $75.00_{(0.5)}$ | $70.44_{(0.4)}$ | $71.88_{(0.7)}$ | 72.34 |
| | Δ | +3.77 | -1.44 | +2.79 | +3.28 | -0.60 | +2.71 | +0.87 | +0.56 | +1.49 |

Table A3: **Ablations for model design.** Performance of ABMIL across individual tasks as a single MAMMOTH component is modified. Lin., linear; Sp., Sparse; MH, multihead; sink., sinkhorn.

| Ablation | Model | | | EBRAINS | | GBMLGG | | BRACS | |
|---|---|---|---|---|---|---|---|---|---|
| | | | | C | F | C | F | C | F |
| Full model | | Ours | | **90.0** | **72.9** | **96.2** | **52.2** | **72.4** | **46.1** |
| MoE method | MAMMOTH | ⇒ | Lin. layer | 86.1 (−4.3%) | 67.2 (−7.8%) | 91.8 (−4.6%) | 51.9 (−0.6%) | 67.1 (−7.3%) | 42.8 (−7.2%) |
| | | | Soft | 88.7 (−1.4%) | 70.3 (−3.6%) | 93.7 (−2.6%) | 43.8 (−16.1%) | 64.9 (−10.4%) | 46.0 (−0.2%) |
| | | | Sp. MH | 88.1 (−2.1%) | 67.5 (−7.4%) | 93.6 (−2.7%) | 53.7 (+2.9%) | 66.0 (−8.8%) | 44.6 (−3.3%) |
| | | | Sp. soft. | 87.2 (−3.1%) | 69.6 (−4.5%) | 93.9 (−2.4%) | 56.0 (+7.3%) | 65.9 (−9.0%) | 28.6 (−38.0%) |
| | | | Sp. sink. | 87.8 (−2.4%) | 70.9 (−2.7%) | 93.7 (−2.6%) | 43.6 (−16.5%) | 64.9 (−10.4%) | 46.0 (−0.2%) |
| Num. heads | 16 | ⇒ | 1 | 87.8 (−2.4%) | 64.1 (−12.1%) | 92.1 (−4.3%) | 51.0 (−2.3%) | 67.2 (−7.2%) | 44.3 (−3.9%) |
| Slot transform | $\mathbf{W}_{low}^{(k)}\Phi$ | ⇒ | $\mathbf{W}_{full}^{(k)}$ | 89.2 (−0.9%) | 72.7 (−0.3%) | 95.2 (−1.0%) | 51.7 (−1.0%) | 69.2 (−4.4%) | 36.2 (−21.5%) |
| $\Phi$ | Shared | ⇒ | Per-expert | 89.9 (−0.1%) | 70.7 (−3.0%) | 95.6 (−0.6%) | 49.6 (−5.0%) | 76.9 (+6.2%) | 41.4 (−10.2%) |
| $\mathbf{W}$ | Learned | ⇒ | Identity | 86.8 (−3.6%) | 73.8 (+1.2%) | 93.2 (−3.1%) | 51.2 (−1.9%) | 63.0 (−13.1%) | 41.1 (−10.9%) |
| Output | Slots | ⇒ | Patches | 88.5 (−1.7%) | 69.6 (−4.5%) | 95.3 (−0.9%) | 53.8 (+3.1%) | 75.3 (+4.0%) | 43.9 (−4.8%) |

Table A4: **Parameter count with varying number of experts** Number of parameters across different expert counts in the task-specific layer as a single MAMMOTH component is modified, where $D = 1024, D' = 512, P = 256$. Linear layer indicates the baseline parameter count without experts. Entries with more parameters than the linear layer are **shown in bold**. Lin., Linear; Sp., Sparse; MH., Multihead.

| Ablation | Model | | | Parameter Count (Millions) | | | |
|---|---|---|---|---|---|---|---|
| | | | | 5 Experts | 10 Experts | 20 Experts | 30 Experts |
| **Full model** | | **Ours** | | 0.5 | 0.5 | 0.5 | 0.5 |
| MoE method | MAMMOTH | ⇒ | Lin. layer | 0.5 | 0.5 | 0.5 | 0.5 |
| | | | Soft | **2.6** | **5.2** | **10.4** | **15.7** |
| | | | Sp. MH | **2.6** | **5.2** | **10.4** | **15.7** |
| | | | Softmax | **2.6** | **5.2** | **10.4** | **15.7** |
| | | | Sinkhorn | **2.6** | **5.2** | **10.4** | **15.7** |
| | | | PaMoE | **2.6** | **5.2** | **10.4** | **15.7** |
| Num. heads | 16 | ⇒ | 1 | 0.5 | 0.5 | 0.5 | 0.5 |
| Slot transform | $\mathbf{W}_{low}^{(k)}\Phi$ | ⇒ | $\mathbf{W}_{full}^{(k)}$ | **0.92** | **1.6** | **2.9** | **4.2** |
| $\Phi$ | Shared | ⇒ | Per-expert | 0.5 | 0.5 | 0.5 | 0.5 |
| $\mathbf{W}$ | Learned | ⇒ | Identity | 0.5 | 0.5 | 0.5 | 0.5 |
| Output | Slots | ⇒ | Patches | 0.5 | 0.5 | 0.5 | 0.5 |

Table A5: **PaMoE comparison** Results of MIL methods with PaMoE and with MAMMOTH. The number of classes is specified below each task. The evaluation metrics for each task are specified in parentheses. All models use UNI features as patch embeddings (Chen et al., 2024a). Performance on NSCLC subtyping is averaged across the internal TCGA cohort and the external NLST and CPTAC cohorts. Trans., Transformer. Standard deviation is reported according to 1,000 bootstrapped trials.

| Task | Status | ABMIL | CLAM | TransMIL | Trans. | ILRA | Mean | Max | DSMIL | Average |
|---|---|---|---|---|---|---|---|---|---|---|
| BRACS-C | PaMoE | 70.52 (1.6) | 54.3 (2.3) | 73.01 (3.3) | 63.40 (2.8) | 63.27 (1.8) | 64.72 (1.3) | 56.79 (1.8) | 61.59 (2.6) | 63.45 (5.9) |
| C=3 | +Ours | 72.70 (1.4) | 73.41 (2.1) | 70.52 (3.1) | 71.11 (3.6) | 74.05 (2.5) | 72.37 (1.4) | 67.21 (1.6) | 68.48 (2.8) | 71.23 (2.2) |
| (Bal. acc.) | Δ | +2.18 | +19.11 | −2.49 | +7.71 | +10.78 | +7.65 | +10.42 | +6.89 | +7.78 |
| BRACS-F | PaMoE | 43.29 (2.0) | 34.43 (1.4) | 43.82 (0.8) | 35.70 (1.9) | 32.65 (2.1) | 34.88 (2.4) | 28.34 (0.5) | 28.59 (0.5) | 35.21 (5.5) |
| C=7 | +Ours | 46.12 (2.4) | 46.82 (1.3) | 38.32 (1.0) | 38.95 (2.0) | 42.50 (1.4) | 43.55 (2.9) | 35.52 (0.5) | 39.72 (0.5) | 41.44 (3.7) |
| (Bal. acc.) | Δ | +2.83 | +12.39 | −5.5 | +3.25 | +9.85 | +8.67 | +7.18 | +11.13 | +6.22 |
| EBRAINS-C | PaMoE | 89.95 (0.7) | 87.82 (0.8) | 86.71 (1.3) | 86.94 (0.6) | 83.41 (1.0) | 86.93 (0.9) | 83.61 (0.1) | 85.45 (0.2) | 86.35 (2.0) |
| C=12 | +Ours | 89.98 (0.7) | 91.32 (0.7) | 88.23 (1.2) | 90.45 (0.9) | 91.68 (0.6) | 89.42 (1.1) | 85.14 (0.1) | 89.17 (0.3) | 89.42 (1.9) |
| (Bal. acc.) | Δ | +0.03 | +3.5 | +1.52 | +3.51 | +8.27 | +2.49 | +1.53 | +3.72 | +3.07 |
| EBRAINS-F | PaMoE | 66.68 (1.1) | 65.83 (0.4) | 67.0 (0.2) | 69.07 (1.7) | 64.64 (1.0) | 64.87 (0.2) | 54.65 (0.3) | 52.13 (0.5) | 63.11 (5.8) |
| C=30 | +Ours | 72.40 (1.2) | 72.51 (0.4) | 74.22 (0.2) | 69.73 (0.1) | 70.23 (0.4) | 72.89 (0.2) | 68.22 (0.3) | 69.40 (0.4) | 71.2 (1.9) |
| (Bal. acc.) | Δ | +5.72 | +6.68 | +7.22 | +0.66 | +5.59 | +8.02 | +13.57 | +17.27 | +8.09 |
| NSCLC | PaMoE | 94.68 (0.1) | 91.73 (0.1) | 93.90 (0.1) | 94.69 (0.1) | 93.25 (0.1) | 91.44 (0.1) | 94.86 (0.1) | 94.08 (0.1) | 93.58 (1.3) |
| C=2 | +Ours | 94.68 (0.1) | 93.72 (0.1) | 93.99 (0.1) | 94.04 (0.1) | 93.87 (0.1) | 93.91 (0.1) | 94.44 (0.1) | 94.43 (0.1) | 94.14 (0.3) |
| (AUROC) | Δ | +0.0 | +1.99 | +0.09 | −0.65 | +0.62 | +2.47 | −0.42 | +0.35 | +0.56 |
| BCNB ER | PaMoE | 93.04 (0.1) | 90.99 (0.1) | 89.77 (0.1) | 90.35 (0.1) | 89.80 (0.4) | 92.61 (0.1) | 88.61 (0.1) | 90.52 (0.1) | 90.71 (1.4) |
| C=2 | +Ours | 92.25 (0.1) | 92.72 (0.1) | 92.15 (0.1) | 92.02 (0.1) | 91.58 (0.1) | 92.19 (0.1) | 91.33 (0.1) | 90.93 (0.1) | 91.9 (0.5) |
| (AUROC) | Δ | −0.79 | +1.73 | +2.38 | +1.67 | +1.78 | −0.42 | +2.72 | +0.41 | +1.19 |
| BCNB HER2 | PaMoE | 72.28 (0.5) | 73.76 (0.2) | 69.96 (0.1) | 69.24 (0.3) | 71.38 (0.4) | 70.98 (0.2) | 67.23 (0.3) | 70.21 (0.1) | 70.63 (1.8) |
| C=2 | +Ours | 74.70 (0.4) | 76.64 (0.3) | 71.40 (0.1) | 75.27 (0.2) | 74.34 (0.2) | 76.35 (0.3) | 75.56 (0.2) | 73.39 (0.1) | 74.71 (1.6) |
| (AUROC) | Δ | +2.42 | +2.88 | +1.44 | +6.03 | +2.96 | +5.37 | +8.33 | +3.18 | +4.08 |
| BCNB PR | PaMoE | 83.81 (0.5) | 83.69 (0.3) | 82.0 (0.4) | 81.49 (0.5) | 81.90 (0.4) | 83.62 (0.2) | 82.89 (0.4) | 83.13 (0.2) | 82.82 (0.8) |
| C=2 | +Ours | 85.84 (0.5) | 85.59 (0.4) | 85.37 (0.5) | 83.92 (0.4) | 83.88 (0.4) | 84.84 (0.2) | 85.12 (0.4) | 83.88 (0.2) | 84.8 (0.8) |
| (AUROC) | Δ | +2.03 | +1.9 | +3.37 | +2.43 | +1.98 | +1.22 | +2.23 | +0.75 | +1.99 |
| BRCA ER | PaMoE | 87.71 (0.3) | 87.15 (0.4) | 87.52 (0.1) | 85.61 (0.9) | 85.00 (0.4) | 84.24 (0.3) | 81.16 (0.4) | 85.53 (0.5) | 85.49 (2.0) |
| C=2 | +Ours | 87.94 (0.3) | 90.06 (0.3) | 88.59 (0.1) | 88.26 (0.7) | 87.01 (0.4) | 88.27 (0.3) | 87.65 (0.3) | 86.75 (0.5) | 88.07 (1.0) |
| (AUROC) | Δ | +0.23 | +2.91 | +1.07 | +2.65 | +2.01 | +4.03 | +6.49 | +1.22 | +2.58 |
| BRCA HER2 | PaMoE | 61.11 (0.9) | 66.94 (0.5) | 61.88 (0.9) | 65.25 (1.2) | 61.80 (1.5) | 63.5 (0.7) | 57.93 (0.4) | 60.38 (0.6) | 62.35 (2.7) |
| C=2 | +Ours | 68.35 (0.8) | 61.84 (0.6) | 64.71 (0.8) | 64.84 (0.1) | 63.40 (0.5) | 67.59 (0.6) | 65.42 (0.4) | 65.94 (0.7) | 65.26 (2.0) |
| (AUROC) | Δ | +7.24 | −5.1 | +2.83 | −0.41 | +1.6 | +4.09 | +7.49 | +5.56 | +2.91 |
| BRCA PIK3CA | PaMoE | 59.55 (0.5) | 59.11 (0.5) | 58.48 (0.5) | 57.43 (0.9) | 58.90 (1.0) | 58.52 (1.0) | 54.07 (0.6) | 53.64 (0.2) | 57.46 (2.2) |
| C=2 | +Ours | 59.55 (0.5) | 58.38 (0.5) | 61.30 (0.5) | 60.27 (0.5) | 59.22 (0.5) | 58.99 (0.6) | 60.22 (0.6) | 60.96 (0.2) | 59.86 (0.9) |
| (AUROC) | Δ | +0.0 | −0.73 | +2.82 | +2.84 | +0.32 | +0.47 | +6.15 | +7.32 | +2.4 |
| BRCA PR | PaMoE | 76.53 (0.4) | 76.5 (0.4) | 75.53 (0.7) | 77.12 (0.3) | 75.91 (0.5) | 75.05 (0.5) | 72.87 (0.2) | 76.26 (0.2) | 75.72 (1.2) |
| C=2 | +Ours | 78.77 (0.5) | 78.80 (0.4) | 79.07 (0.8) | 79.44 (0.1) | 77.10 (0.3) | 79.54 (0.6) | 78.21 (0.2) | 79.05 (0.2) | 78.75 (0.7) |
| (AUROC) | Δ | +2.24 | +2.3 | +3.54 | +2.32 | +1.19 | +4.49 | +5.34 | +2.79 | +3.03 |
| GBMLGG-C | PaMoE | 95.79 (0.5) | 93.83 (0.5) | 94.65 (0.2) | 93.41 (1.0) | 93.72 (0.5) | 93.59 (0.6) | 87.12 (0.5) | 89.13 (0.4) | 92.66 (2.8) |
| C=2 | +Ours | 96.19 (0.4) | 94.53 (0.5) | 95.74 (0.3) | 95.68 (0.4) | 93.47 (0.4) | 95.34 (0.7) | 95.54 (0.6) | 94.80 (0.5) | 95.16 (0.8) |
| (AUROC) | Δ | +0.4 | +0.7 | +1.09 | +2.27 | −0.25 | +1.75 | +8.42 | +5.67 | +2.51 |
| GBMLGG-F | PaMoE | 54.15 (1.8) | 52.5 (1.0) | 53.94 (1.2) | 50.58 (1.9) | 49.57 (1.2) | 53.02 (1.4) | 51.19 (0.6) | 51.31 (0.7) | 52.03 (1.5) |
| C=5 | +Ours | 52.22 (1.7) | 51.03 (1.1) | 50.28 (1.4) | 53.12 (0.4) | 52.53 (1.0) | 51.63 (1.2) | 51.71 (0.6) | 50.68 (0.7) | 51.65 (0.9) |
| (Bal. acc.) | Δ | −1.93 | −1.47 | −3.66 | +2.54 | +2.96 | −1.39 | +0.52 | −0.63 | −0.38 |
| LUNG EGFR | PaMoE | 67.25 (1.2) | 61.41 (1.2) | 62.68 (2.4) | 60.20 (1.8) | 62.00 (1.5) | 62.45 (0.9) | 65.59 (1.3) | 59.78 (1.5) | 62.67 (2.4) |
| C=2 | +Ours | 63.68 (1.2) | 65.98 (1.4) | 65.30 (2.3) | 67.55 (1.2) | 62.57 (1.7) | 66.17 (1.1) | 64.12 (1.3) | 65.51 (1.3) | 65.11 (1.5) |
| (AUROC) | Δ | −3.57 | +4.57 | +2.62 | +7.35 | +0.57 | +3.72 | −1.47 | +5.73 | +2.44 |
| LUNG KRAS | PaMoE | 59.52 (1.5) | 59.18 (0.8) | 60.41 (0.1) | 58.22 (1.5) | 60.10 (0.9) | 62.58 (0.6) | 54.73 (0.8) | 51.82 (0.8) | 58.32 (3.2) |
| C=2 | +Ours | 59.40 (1.5) | 59.42 (0.8) | 61.20 (0.1) | 59.45 (0.6) | 61.31 (0.8) | 61.22 (0.5) | 61.35 (0.7) | 58.10 (0.7) | 60.18 (1.2) |
| (AUROC) | Δ | −0.12 | +0.24 | +0.79 | +1.23 | +1.21 | −1.36 | +6.62 | +6.28 | +1.86 |
| LUNG STK11 | PaMoE | 75.41 (1.4) | 70.47 (1.0) | 69.28 (1.1) | 71.14 (0.3) | 69.10 (1.6) | 67.83 (0.8) | 68.02 (0.3) | 65.74 (0.4) | 69.62 (2.7) |
| C=2 | +Ours | 74.36 (1.5) | 70.57 (0.9) | 66.44 (1.1) | 69.39 (0.6) | 68.10 (0.8) | 74.31 (0.7) | 73.48 (0.3) | 68.61 (0.4) | 70.66 (2.9) |
| (AUROC) | Δ | −1.05 | +0.1 | −2.84 | −1.75 | −1.0 | +6.48 | +5.46 | +2.87 | +1.03 |
| LUNG TP53 | PaMoE | 70.22 (0.8) | 68.22 (0.9) | 69.47 (0.5) | 70.19 (1.1) | 69.89 (0.7) | 72.46 (0.5) | 70.09 (0.5) | 70.7 (0.7) | 70.16 (1.1) |
| C=2 | +Ours | 76.20 (0.7) | 71.60 (0.8) | 70.86 (0.5) | 73.46 (0.9) | 69.29 (0.7) | 75.00 (0.5) | 70.44 (0.6) | 71.88 (0.7) | 72.34 (2.2) |
| (AUROC) | Δ | +5.98 | +3.38 | +1.39 | +3.27 | −0.6 | +2.54 | +0.35 | +1.18 | +2.19 |

Table A6: **Pre-Aggregation comparisons with tissue subtyping.** Performance comparison across MIL methods for tissue subtyping with different plug-and-play methods. Best task-level performance is shown in **bold**, second best underlined. Ours (MAMMOTH) consistently yields the highest performance across all tasks and MIL methods.

| | Task | Base | +Ours | +PAMoE | +RRT | +MIL Dropout | +Querent |
|---|---|---|---|---|---|---|---|
| **ABMIL** | BRACS C | 67.10 (1.20) | **72.70** (1.40) | 70.52 (1.24) | 66.39 (1.29) | 71.60 (1.37) | 66.49 (1.22) |
| | BRACS F | 42.84 (2.50) | 46.12 (2.40) | 43.29 (2.76) | 44.13 (2.65) | 47.09 (2.42) | **47.71** (2.64) |
| | EBRAINS C | 86.10 (1.10) | **89.98** (0.70) | 89.95 (0.80) | 88.70 (0.96) | 88.51 (0.90) | 86.76 (0.94) |
| | EBRAINS F | 67.20 (1.00) | **72.40** (1.20) | 66.68 (1.31) | 69.20 (1.26) | 68.25 (1.03) | 67.94 (1.22) |
| | NSCLC | 94.68 (0.10) | 94.68 (0.10) | 94.68 (0.10) | 95.42 (0.10) | 94.33 (0.10) | **95.67** (0.10) |
| | PANDA | 93.12 (0.20) | **94.28** (0.20) | 93.36 (0.22) | 93.29 (0.22) | 90.35 (0.21) | 90.46 (0.21) |
| | **Average** | 75.17 | **78.36** | 76.41 | 76.19 | 77.69 | 75.84 |
| **CLAM** | BRACS C | 56.16 (2.32) | **73.41** (2.06) | 54.30 (2.03) | 60.33 (2.26) | 58.28 (2.12) | 65.04 (2.25) |
| | BRACS F | 32.26 (2.91) | **46.82** (1.26) | 34.43 (1.41) | 34.05 (1.96) | 33.05 (2.33) | 34.07 (1.67) |
| | EBRAINS C | 87.85 (0.91) | **91.32** (0.73) | 87.82 (0.68) | 90.54 (0.83) | 88.88 (0.90) | 86.45 (0.81) |
| | EBRAINS F | 69.77 (1.37) | **72.51** (0.43) | 65.83 (0.39) | 71.52 (1.02) | 69.83 (0.69) | 56.95 (0.97) |
| | NSCLC | 91.73 (0.12) | 93.72 (0.10) | 91.73 (0.12) | 91.50 (0.12) | 90.52 (0.10) | **96.01** (0.10) |
| | PANDA | 92.60 (0.40) | **93.26** (0.27) | 93.11 (0.29) | 92.59 (0.38) | 88.26 (0.35) | 92.14 (0.35) |
| | **Average** | 71.73 | **78.51** | 71.20 | 73.42 | 72.47 | 71.77 |
| **DSMIL** | BRACS C | 62.64 (2.40) | 68.48 (2.80) | 61.59 (2.86) | 57.65 (2.48) | 56.65 (2.78) | **68.84** (2.49) |
| | BRACS F | 36.48 (4.20) | 39.72 (0.50) | 28.59 (0.48) | **48.64** (2.59) | 47.64 (3.90) | 34.75 (1.13) |
| | EBRAINS C | 86.37 (2.00) | **89.17** (0.30) | 85.45 (0.32) | 84.77 (0.70) | 83.77 (1.31) | 87.01 (1.76) |
| | EBRAINS F | 63.87 (1.70) | 69.40 (0.40) | 52.13 (0.44) | 68.61 (0.76) | 62.91 (1.53) | **72.43** (0.77) |
| | NSCLC | 94.08 (0.11) | 94.43 (0.11) | 94.08 (0.11) | 93.53 (0.11) | 92.53 (0.11) | **94.56** (0.11) |
| | PANDA | 92.78 (0.20) | 92.96 (0.10) | 91.26 (0.11) | **93.56** (0.15) | 92.56 (0.16) | 92.20 (0.11) |
| | **Average** | 72.70 | **75.69** | 68.85 | 74.46 | 73.68 | 74.97 |
| **ILRA** | BRACS C | 63.27 (1.78) | **74.05** (2.54) | 63.27 (1.78) | 62.14 (2.32) | 66.57 (2.31) | 70.74 (2.23) |
| | BRACS F | 32.65 (2.10) | 42.50 (1.38) | 32.65 (2.10) | 36.21 (1.80) | 32.85 (1.51) | **46.06** (1.48) |
| | EBRAINS C | 83.41 (0.98) | **91.68** (0.64) | 83.41 (0.98) | 89.15 (0.67) | 85.27 (0.81) | 90.40 (0.74) |
| | EBRAINS F | 64.64 (0.98) | 70.23 (0.37) | 64.64 (0.98) | 69.13 (0.65) | 61.65 (0.67) | **71.14** (0.73) |
| | NSCLC | 93.25 (0.09) | 93.87 (0.12) | 93.25 (0.09) | **94.36** (0.11) | 93.42 (0.10) | 93.28 (0.10) |
| | PANDA | 91.89 (0.30) | **94.07** (0.25) | 91.89 (0.30) | 93.47 (0.29) | 91.65 (0.29) | 92.98 (0.29) |
| | **Average** | 71.52 | **77.73** | 71.52 | 74.07 | 72.90 | 77.43 |
| **meanMIL** | BRACS C | 65.13 (1.70) | **72.37** (1.40) | 64.72 (1.14) | 57.43 (1.18) | 56.43 (1.49) | 67.93 (1.37) |
| | BRACS F | 33.68 (1.40) | **43.55** (2.90) | 34.88 (2.40) | 32.31 (2.51) | 31.31 (2.61) | 29.51 (2.55) |
| | EBRAINS C | 86.70 (0.70) | **89.42** (1.10) | 86.93 (1.06) | 86.90 (0.71) | 85.90 (1.01) | 82.13 (0.77) |
| | EBRAINS F | 70.30 (1.40) | **72.89** (0.20) | 64.87 (0.17) | 72.28 (0.66) | 71.28 (0.84) | 60.03 (1.19) |
| | NSCLC | 91.44 (0.11) | 93.91 (0.11) | 91.44 (0.11) | 92.27 (0.11) | 91.27 (0.11) | **94.96** (0.11) |
| | PANDA | 92.67 (0.30) | **93.52** (0.20) | 90.64 (0.21) | 92.46 (0.21) | 91.46 (0.23) | 92.46 (0.25) |
| | **Average** | 73.32 | **77.61** | 72.25 | 72.27 | 72.27 | 71.17 |
| **TransMIL** | BRACS C | 66.80 (2.70) | 70.52 (3.10) | **73.01** (2.83) | 57.43 (3.01) | 62.90 (3.04) | 60.28 (3.03) |
| | BRACS F | 32.10 (2.70) | 38.32 (1.00) | 43.82 (1.03) | **44.21** (2.03) | 43.42 (1.63) | 40.13 (2.68) |
| | EBRAINS C | 87.86 (1.10) | 88.23 (1.20) | 86.71 (1.39) | **89.75** (1.34) | 85.35 (1.29) | 88.96 (1.19) |
| | EBRAINS F | 65.20 (0.50) | **74.22** (0.20) | 67.00 (0.21) | 67.67 (0.36) | 66.83 (0.35) | 68.48 (0.36) |
| | NSCLC | 93.90 (0.08) | 93.99 (0.10) | 93.90 (0.08) | **95.22** (0.09) | 93.97 (0.09) | 94.78 (0.09) |
| | PANDA | 90.75 (0.70) | **93.68** (0.30) | 89.74 (0.34) | 90.78 (0.34) | 91.29 (0.42) | 92.23 (0.31) |
| | **Average** | 72.77 | **76.49** | 75.70 | 74.18 | 74.96 | 74.14 |
| **Transformer** | BRACS C | 63.40 (2.80) | **71.11** (3.60) | 63.40 (2.80) | 68.70 (3.42) | 58.78 (3.22) | 60.51 (3.12) |
| | BRACS F | 35.70 (1.90) | 38.95 (2.00) | 35.70 (1.90) | **44.21** (1.92) | 40.45 (1.98) | 40.56 (1.92) |
| | EBRAINS C | 86.94 (0.60) | **90.45** (0.90) | 86.94 (0.60) | 85.63 (0.89) | 84.03 (0.83) | 83.47 (0.81) |
| | EBRAINS F | 69.07 (1.70) | 69.73 (0.10) | 69.07 (1.70) | 66.74 (0.98) | 67.10 (0.21) | **70.16** (0.79) |
| | NSCLC | 94.69 (0.08) | 94.04 (0.10) | 94.69 (0.08) | 93.72 (0.10) | 93.41 (0.09) | **95.35** (0.09) |
| | PANDA | 91.39 (0.50) | 91.90 (0.80) | 91.39 (0.50) | 89.97 (0.63) | 88.72 (0.51) | **92.46** (0.60) |
| | **Average** | 73.53 | **76.03** | 73.53 | 74.83 | 73.08 | 73.75 |
| **maxMIL** | BRACS C | 64.54 (2.40) | **67.21** (1.60) | 56.79 (1.28) | 57.88 (1.83) | 63.27 (2.02) | 60.91 (1.97) |
| | BRACS F | 33.90 (2.40) | 35.52 (0.50) | 28.34 (0.48) | 28.40 (2.13) | 33.61 (0.69) | **47.63** (0.60) |
| | EBRAINS C | 84.55 (1.20) | 85.14 (0.10) | 83.61 (0.09) | **87.60** (0.49) | 81.76 (0.94) | 77.98 (1.06) |
| | EBRAINS F | 64.94 (1.00) | **68.22** (0.30) | 54.65 (0.35) | 68.10 (0.78) | 63.26 (0.72) | 56.11 (0.95) |
| | NSCLC | 94.86 (0.10) | 94.44 (0.10) | 94.86 (0.10) | **95.59** (0.10) | 94.89 (0.10) | 94.33 (0.10) |
| | PANDA | 88.79 (0.30) | **92.34** (0.20) | 86.81 (0.20) | 88.82 (0.24) | 87.81 (0.29) | 90.75 (0.26) |
| | **Average** | 71.93 | **73.81** | 67.51 | 71.06 | 71.77 | 71.29 |

Table A7: **MIL performance with pre-aggregation methods**. Performance comparison across Transformer, TransMIL, ILRA, and CLAM MIL methods for molecular subtyping. Best task-level performance is shown in **bold**, second best underlined.

| | Task | Base | +Ours | +PAMoE | +RRT | +MIL Dropout | +Querent |
|---|---|---|---|---|---|---|---|
| **Transformer** | BCNB ER | 90.35 $_{(0.50)}$ | **92.02** $_{(0.09)}$ | 90.35 $_{(0.50)}$ | 88.25 $_{(0.32)}$ | 87.67 $_{(0.47)}$ | 90.87 $_{(0.24)}$ |
| | BCNB HER2 | 69.24 $_{(0.30)}$ | **75.27** $_{(0.20)}$ | 69.24 $_{(0.30)}$ | 68.58 $_{(0.30)}$ | 66.07 $_{(0.24)}$ | 74.36 $_{(0.29)}$ |
| | BCNB PR | 81.49 $_{(0.50)}$ | **83.92** $_{(0.33)}$ | 81.49 $_{(0.50)}$ | 78.78 $_{(0.42)}$ | 79.54 $_{(0.35)}$ | 83.65 $_{(0.41)}$ |
| | BRCA ER | 85.61 $_{(0.90)}$ | **88.26** $_{(0.70)}$ | 85.61 $_{(0.90)}$ | 84.35 $_{(0.88)}$ | 87.07 $_{(0.88)}$ | 86.34 $_{(0.88)}$ |
| | BRCA HER2 | **65.25** $_{(1.20)}$ | 64.84 $_{(0.10)}$ | **65.25** $_{(1.20)}$ | 57.00 $_{(0.69)}$ | 58.21 $_{(0.73)}$ | 63.43 $_{(0.14)}$ |
| | BRCA PIK3CA | 57.43 $_{(0.90)}$ | **60.27** $_{(0.50)}$ | 57.43 $_{(0.90)}$ | 53.60 $_{(0.87)}$ | 52.80 $_{(0.58)}$ | 57.58 $_{(0.55)}$ |
| | BRCA PR | 77.12 $_{(0.30)}$ | **79.44** $_{(0.10)}$ | 77.12 $_{(0.30)}$ | 75.76 $_{(0.21)}$ | 74.49 $_{(0.23)}$ | 75.50 $_{(0.26)}$ |
| | GBMLGG C | 93.41 $_{(1.00)}$ | **95.68** $_{(0.51)}$ | 93.41 $_{(1.00)}$ | 93.50 $_{(0.69)}$ | 93.21 $_{(0.99)}$ | 92.36 $_{(0.75)}$ |
| | GBMLGG F | 50.58 $_{(1.90)}$ | **53.12** $_{(0.40)}$ | 50.58 $_{(1.90)}$ | 48.04 $_{(1.45)}$ | 50.07 $_{(0.59)}$ | 51.17 $_{(1.80)}$ |
| | LUNG EGFR | 60.20 $_{(1.80)}$ | 67.55 $_{(1.20)}$ | 60.20 $_{(1.80)}$ | 59.06 $_{(1.39)}$ | 58.00 $_{(1.48)}$ | **68.23** $_{(1.56)}$ |
| | LUNG KRAS | 58.22 $_{(1.50)}$ | **59.45** $_{(0.78)}$ | 58.22 $_{(1.50)}$ | 53.69 $_{(1.26)}$ | 53.39 $_{(1.10)}$ | 57.02 $_{(1.27)}$ |
| | LUNG STK11 | **71.14** $_{(0.30)}$ | 69.39 $_{(0.60)}$ | **71.14** $_{(0.30)}$ | 64.45 $_{(0.47)}$ | 66.04 $_{(0.53)}$ | 69.57 $_{(0.36)}$ |
| | LUNG TP53 | 70.19 $_{(1.10)}$ | **73.46** $_{(0.90)}$ | 70.19 $_{(1.10)}$ | 68.98 $_{(1.00)}$ | 66.36 $_{(1.03)}$ | 69.04 $_{(0.92)}$ |
| | **Average** | 71.56 | **74.05** | 71.56 | 68.77 | 69.69 | 72.24 |
| **TransMIL** | BCNB ER | 91.08 $_{(0.40)}$ | **92.15** $_{(0.10)}$ | 89.77 $_{(0.11)}$ | 89.45 $_{(0.33)}$ | 88.79 $_{(0.25)}$ | 89.85 $_{(0.33)}$ |
| | BCNB HER2 | 68.90 $_{(0.70)}$ | 71.40 $_{(0.10)}$ | 69.96 $_{(0.10)}$ | 72.43 $_{(0.20)}$ | 71.86 $_{(0.13)}$ | **75.51** $_{(0.62)}$ |
| | BCNB PR | 82.30 $_{(0.80)}$ | **85.37** $_{(0.50)}$ | 82.00 $_{(0.56)}$ | 82.04 $_{(0.78)}$ | 80.75 $_{(0.52)}$ | 83.69 $_{(0.53)}$ |
| | BRCA ER | 87.38 $_{(0.30)}$ | **88.59** $_{(0.10)}$ | 87.52 $_{(0.09)}$ | 87.47 $_{(0.28)}$ | 86.51 $_{(0.13)}$ | 86.70 $_{(0.25)}$ |
| | BRCA HER2 | 61.31 $_{(1.50)}$ | **64.71** $_{(0.80)}$ | 61.88 $_{(0.82)}$ | 62.98 $_{(1.45)}$ | 61.74 $_{(0.98)}$ | 61.47 $_{(1.13)}$ |
| | BRCA PIK3CA | 58.79 $_{(1.90)}$ | **61.30** $_{(0.43)}$ | 58.48 $_{(0.50)}$ | 56.90 $_{(1.53)}$ | 55.52 $_{(1.59)}$ | 59.99 $_{(1.19)}$ |
| | BRCA PR | 78.02 $_{(1.20)}$ | **79.07** $_{(0.80)}$ | 75.53 $_{(0.76)}$ | 74.39 $_{(0.81)}$ | 72.75 $_{(0.95)}$ | 78.17 $_{(1.07)}$ |
| | GBMLGG C | 94.46 $_{(0.10)}$ | **95.74** $_{(0.30)}$ | 94.65 $_{(0.35)}$ | 94.01 $_{(0.21)}$ | 92.62 $_{(0.35)}$ | 91.56 $_{(0.22)}$ |
| | GBMLGG F | 52.19 $_{(1.60)}$ | 50.28 $_{(1.40)}$ | **53.94** $_{(1.65)}$ | 51.03 $_{(1.58)}$ | 53.62 $_{(1.54)}$ | 52.01 $_{(1.43)}$ |
| | LUNG EGFR | 63.66 $_{(1.20)}$ | **65.30** $_{(2.30)}$ | 62.68 $_{(2.35)}$ | 62.03 $_{(1.40)}$ | 61.74 $_{(1.31)}$ | 61.80 $_{(1.78)}$ |
| | LUNG KRAS | 60.31 $_{(1.10)}$ | **61.20** $_{(0.10)}$ | 60.41 $_{(0.09)}$ | 55.39 $_{(0.66)}$ | 57.25 $_{(0.87)}$ | 58.90 $_{(0.10)}$ |
| | LUNG STK11 | 68.95 $_{(2.20)}$ | 66.44 $_{(1.10)}$ | 69.28 $_{(1.31)}$ | 60.10 $_{(1.84)}$ | **69.48** $_{(1.38)}$ | 67.63 $_{(1.34)}$ |
| | LUNG TP53 | 68.07 $_{(0.70)}$ | 70.86 $_{(0.50)}$ | 69.47 $_{(0.54)}$ | 67.64 $_{(0.62)}$ | 68.31 $_{(0.67)}$ | **71.77** $_{(0.52)}$ |
| | **Average** | 71.96 | **73.26** | 71.97 | 70.45 | 71.84 | 72.23 |
| **ILRA** | BCNB ER | 89.80 $_{(0.46)}$ | **91.58** $_{(0.09)}$ | 89.80 $_{(0.46)}$ | 91.25 $_{(0.25)}$ | 91.04 $_{(0.13)}$ | 90.11 $_{(0.29)}$ |
| | BCNB HER2 | 71.38 $_{(0.47)}$ | 74.34 $_{(0.26)}$ | 71.38 $_{(0.47)}$ | 73.10 $_{(0.29)}$ | 73.73 $_{(0.42)}$ | **75.16** $_{(0.27)}$ |
| | BCNB PR | 81.90 $_{(0.51)}$ | 83.88 $_{(0.29)}$ | 81.90 $_{(0.51)}$ | **84.28** $_{(0.50)}$ | 81.56 $_{(0.34)}$ | 82.36 $_{(0.50)}$ |
| | BRCA ER | 85.00 $_{(0.47)}$ | 87.01 $_{(0.40)}$ | 85.00 $_{(0.47)}$ | 86.50 $_{(0.44)}$ | 85.80 $_{(0.43)}$ | **87.24** $_{(0.47)}$ |
| | BRCA HER2 | 61.80 $_{(1.21)}$ | 63.40 $_{(0.51)}$ | 61.80 $_{(1.21)}$ | **63.86** $_{(0.80)}$ | 62.34 $_{(0.59)}$ | 63.40 $_{(0.60)}$ |
| | BRCA PIK3CA | 58.90 $_{(1.04)}$ | 59.22 $_{(0.52)}$ | 58.90 $_{(1.04)}$ | 57.66 $_{(0.76)}$ | **59.31** $_{(0.69)}$ | 57.88 $_{(0.99)}$ |
| | BRCA PR | 75.91 $_{(0.58)}$ | **77.10** $_{(0.33)}$ | 75.91 $_{(0.58)}$ | 77.01 $_{(0.44)}$ | 70.99 $_{(0.57)}$ | 74.86 $_{(0.48)}$ |
| | GBMLGG C | 93.72 $_{(0.52)}$ | 93.47 $_{(0.38)}$ | 93.72 $_{(0.52)}$ | 93.10 $_{(0.39)}$ | 91.85 $_{(0.43)}$ | **94.28** $_{(0.43)}$ |
| | GBMLGG F | 49.57 $_{(1.25)}$ | **52.53** $_{(1.18)}$ | 49.57 $_{(1.25)}$ | 49.86 $_{(1.19)}$ | 50.71 $_{(1.25)}$ | 51.95 $_{(1.22)}$ |
| | LUNG EGFR | 62.00 $_{(1.77)}$ | 62.57 $_{(1.61)}$ | 62.00 $_{(1.77)}$ | 60.34 $_{(1.67)}$ | 58.43 $_{(1.64)}$ | **64.11** $_{(1.69)}$ |
| | LUNG KRAS | 60.10 $_{(0.96)}$ | **61.31** $_{(0.71)}$ | 60.10 $_{(0.96)}$ | 54.69 $_{(0.72)}$ | 55.95 $_{(0.89)}$ | 57.01 $_{(0.93)}$ |
| | LUNG STK11 | **69.10** $_{(1.16)}$ | 68.10 $_{(0.87)}$ | **69.10** $_{(1.16)}$ | 64.85 $_{(0.94)}$ | 68.56 $_{(0.97)}$ | 66.32 $_{(1.11)}$ |
| | LUNG TP53 | 69.89 $_{(0.94)}$ | 69.29 $_{(0.66)}$ | 69.89 $_{(0.94)}$ | 68.05 $_{(0.92)}$ | 68.11 $_{(0.77)}$ | **70.72** $_{(0.80)}$ |
| | **Average** | 71.47 | **72.60** | 71.47 | 71.12 | 71.64 | 71.95 |
| **CLAM** | BCNB ER | 91.22 $_{(0.42)}$ | **92.72** $_{(0.09)}$ | 90.99 $_{(0.09)}$ | 90.59 $_{(0.13)}$ | 89.56 $_{(0.13)}$ | 88.17 $_{(0.22)}$ |
| | BCNB HER2 | 73.91 $_{(0.38)}$ | **76.64** $_{(0.21)}$ | 73.76 $_{(0.24)}$ | 73.82 $_{(0.23)}$ | 72.97 $_{(0.28)}$ | 75.34 $_{(0.34)}$ |
| | BCNB PR | 84.16 $_{(0.48)}$ | **85.59** $_{(0.32)}$ | 83.69 $_{(0.28)}$ | 84.80 $_{(0.47)}$ | 83.76 $_{(0.30)}$ | 83.15 $_{(0.38)}$ |
| | BRCA ER | 86.46 $_{(0.35)}$ | **90.06** $_{(0.35)}$ | 87.15 $_{(0.35)}$ | 86.47 $_{(0.35)}$ | 85.32 $_{(0.35)}$ | 87.18 $_{(0.35)}$ |
| | BRCA HER2 | 64.38 $_{(1.31)}$ | 61.84 $_{(0.58)}$ | **66.94** $_{(0.60)}$ | 63.88 $_{(0.61)}$ | 63.15 $_{(0.58)}$ | 63.76 $_{(1.14)}$ |
| | BRCA PIK3CA | 59.15 $_{(1.26)}$ | 58.38 $_{(0.62)}$ | 59.11 $_{(0.50)}$ | 59.18 $_{(0.62)}$ | 57.96 $_{(0.68)}$ | **59.87** $_{(0.95)}$ |
| | BRCA PR | 77.73 $_{(0.46)}$ | **78.80** $_{(0.39)}$ | 76.50 $_{(0.46)}$ | 76.32 $_{(0.44)}$ | 75.27 $_{(0.39)}$ | 78.21 $_{(0.45)}$ |
| | GBMLGG C | 94.38 $_{(0.62)}$ | 94.53 $_{(0.53)}$ | 93.83 $_{(0.55)}$ | **95.17** $_{(0.58)}$ | 94.17 $_{(0.59)}$ | 92.68 $_{(0.59)}$ |
| | GBMLGG F | 49.78 $_{(1.57)}$ | 51.03 $_{(0.98)}$ | **52.50** $_{(1.15)}$ | 49.37 $_{(1.46)}$ | 46.42 $_{(1.20)}$ | 49.66 $_{(1.16)}$ |
| | LUNG EGFR | 65.85 $_{(1.60)}$ | 65.98 $_{(1.63)}$ | 61.41 $_{(1.34)}$ | 63.44 $_{(1.42)}$ | 64.33 $_{(1.38)}$ | 65.10 $_{(1.48)}$ |
| | LUNG KRAS | **60.81** $_{(0.97)}$ | 59.42 $_{(0.77)}$ | 59.18 $_{(0.66)}$ | 55.26 $_{(0.75)}$ | 56.33 $_{(0.68)}$ | 57.45 $_{(0.73)}$ |
| | LUNG STK11 | 65.75 $_{(1.33)}$ | 70.57 $_{(0.88)}$ | 70.47 $_{(0.99)}$ | 69.19 $_{(1.31)}$ | 69.85 $_{(1.27)}$ | **76.65** $_{(1.27)}$ |
| | LUNG TP53 | **73.04** $_{(0.67)}$ | 71.60 $_{(0.55)}$ | 68.22 $_{(0.46)}$ | 69.56 $_{(0.58)}$ | 70.44 $_{(0.52)}$ | 70.90 $_{(0.61)}$ |
| | **Average** | 72.82 | **73.63** | 72.60 | 72.08 | 72.50 | 72.93 |

Table A8: **Pre-Aggregation comparisons with molecular subtyping.** Performance comparison across MIL methods (ABMIL, maxMIL, meanMIL, DSMIL) for molecular subtyping. Best task-level performance is shown in **bold**, second best underlined.

| | Task | Base | +Ours | +PAMoE | +RRT | +MIL Dropout | +Querent |
|---|---|---|---|---|---|---|---|
| ABMIL | BCNB ER | $90.38_{(0.20)}$ | $\underline{92.25}_{(0.10)}$ | $\mathbf{93.04}_{(0.08)}$ | $91.71_{(0.08)}$ | $91.42_{(0.18)}$ | $90.07_{(0.09)}$ |
| | BCNB HER2 | $73.05_{(0.40)}$ | $\underline{74.70}_{(0.40)}$ | $72.28_{(0.42)}$ | $74.66_{(0.42)}$ | $73.91_{(0.41)}$ | $\mathbf{76.14}_{(0.40)}$ |
| | BCNB PR | $82.48_{(0.40)}$ | $\mathbf{85.84}_{(0.50)}$ | $83.81_{(0.45)}$ | $84.26_{(0.41)}$ | $84.08_{(0.46)}$ | $\underline{84.29}_{(0.49)}$ |
| | BRCA ER | $86.93_{(0.30)}$ | $\mathbf{87.94}_{(0.30)}$ | $87.71_{(0.25)}$ | $\underline{87.78}_{(0.28)}$ | $87.06_{(0.26)}$ | $86.80_{(0.29)}$ |
| | BRCA HER2 | $64.35_{(1.10)}$ | $\mathbf{68.35}_{(0.80)}$ | $61.11_{(0.69)}$ | $\underline{67.00}_{(0.80)}$ | $66.01_{(0.70)}$ | $64.33_{(0.90)}$ |
| | BRCA PIK3CA | $\underline{60.23}_{(0.70)}$ | $59.55_{(0.50)}$ | $59.55_{(0.58)}$ | $59.52_{(0.57)}$ | $59.63_{(0.67)}$ | $\mathbf{60.86}_{(0.68)}$ |
| | BRCA PR | $76.37_{(0.30)}$ | $\mathbf{78.77}_{(0.50)}$ | $76.53_{(0.44)}$ | $77.26_{(0.35)}$ | $\underline{78.31}_{(0.31)}$ | $77.66_{(0.47)}$ |
| | GBMLGG C | $91.82_{(0.40)}$ | $\mathbf{96.19}_{(0.40)}$ | $95.79_{(0.39)}$ | $\underline{95.81}_{(0.39)}$ | $95.09_{(0.40)}$ | $95.12_{(0.45)}$ |
| | GBMLGG F | $51.89_{(1.30)}$ | $\underline{52.22}_{(1.70)}$ | $\mathbf{54.15}_{(2.01)}$ | $47.98_{(1.76)}$ | $43.47_{(1.97)}$ | $51.94_{(1.68)}$ |
| | LUNG EGFR | $61.27_{(1.20)}$ | $63.68_{(1.20)}$ | $\underline{67.25}_{(1.18)}$ | $\mathbf{67.27}_{(1.19)}$ | $66.45_{(1.18)}$ | $63.90_{(1.19)}$ |
| | LUNG KRAS | $58.06_{(0.70)}$ | $59.40_{(1.50)}$ | $\mathbf{59.52}_{(1.54)}$ | $57.89_{(1.44)}$ | $58.88_{(1.46)}$ | $\underline{59.28}_{(0.81)}$ |
| | LUNG STK11 | $\mathbf{76.41}_{(1.10)}$ | $74.36_{(1.50)}$ | $75.41_{(1.57)}$ | $75.50_{(1.17)}$ | $75.54_{(1.49)}$ | $\underline{76.06}_{(1.38)}$ |
| | LUNG TP53 | $72.43_{(1.10)}$ | $\mathbf{76.20}_{(0.70)}$ | $70.22_{(0.69)}$ | $71.47_{(0.73)}$ | $70.87_{(0.73)}$ | $70.49_{(0.90)}$ |
| | **Average** | 72.74 | **74.57** | 73.57 | 73.70 | 73.13 | 73.61 |
| MaxMIL | BCNB ER | $90.07_{(0.60)}$ | $\mathbf{91.33}_{(0.12)}$ | $88.61_{(0.13)}$ | $\underline{90.74}_{(0.29)}$ | $90.38_{(0.41)}$ | $88.61_{(0.38)}$ |
| | BCNB HER2 | $74.33_{(0.50)}$ | $\underline{75.56}_{(0.23)}$ | $67.23_{(0.20)}$ | $75.33_{(0.34)}$ | $\mathbf{75.97}_{(0.28)}$ | $75.00_{(0.36)}$ |
| | BCNB PR | $\underline{84.34}_{(0.50)}$ | $\mathbf{85.12}_{(0.40)}$ | $82.89_{(0.34)}$ | $83.97_{(0.44)}$ | $82.65_{(0.46)}$ | $83.69_{(0.45)}$ |
| | BRCA ER | $\underline{86.84}_{(0.30)}$ | $\mathbf{87.65}_{(0.36)}$ | $81.16_{(0.41)}$ | $84.57_{(0.40)}$ | $84.18_{(0.37)}$ | $86.08_{(0.38)}$ |
| | BRCA HER2 | $63.58_{(2.60)}$ | $\underline{65.42}_{(0.40)}$ | $57.93_{(0.44)}$ | $63.57_{(1.29)}$ | $59.20_{(0.99)}$ | $\mathbf{65.91}_{(1.66)}$ |
| | BRCA PIK3CA | $\mathbf{61.67}_{(1.10)}$ | $\underline{60.22}_{(0.60)}$ | $54.07_{(0.57)}$ | $58.57_{(0.94)}$ | $56.76_{(0.61)}$ | $58.40_{(0.64)}$ |
| | BRCA PR | $77.79_{(0.20)}$ | $\underline{78.21}_{(0.20)}$ | $72.87_{(0.16)}$ | $\mathbf{79.34}_{(0.17)}$ | $78.06_{(0.17)}$ | $77.36_{(0.18)}$ |
| | GBMLGG C | $\underline{95.34}_{(1.00)}$ | $\mathbf{95.54}_{(0.50)}$ | $87.12_{(0.53)}$ | $93.49_{(0.55)}$ | $92.92_{(0.67)}$ | $91.60_{(0.90)}$ |
| | GBMLGG F | $50.31_{(0.80)}$ | $\underline{51.71}_{(0.60)}$ | $51.19_{(0.57)}$ | $\mathbf{55.32}_{(0.62)}$ | $49.49_{(0.69)}$ | $48.43_{(0.63)}$ |
| | LUNG EGFR | $65.43_{(4.10)}$ | $64.12_{(1.30)}$ | $\underline{65.59}_{(1.30)}$ | $\mathbf{66.44}_{(1.76)}$ | $60.62_{(2.89)}$ | $63.74_{(3.29)}$ |
| | LUNG KRAS | $56.93_{(0.60)}$ | $\mathbf{61.35}_{(0.71)}$ | $54.73_{(0.74)}$ | $56.23_{(0.70)}$ | $56.79_{(0.74)}$ | $\underline{60.51}_{(0.72)}$ |
| | LUNG STK11 | $70.06_{(2.60)}$ | $\mathbf{73.48}_{(0.30)}$ | $68.02_{(0.30)}$ | $69.47_{(0.73)}$ | $\underline{73.13}_{(1.79)}$ | $68.02_{(1.78)}$ |
| | LUNG TP53 | $69.57_{(0.90)}$ | $70.44_{(0.55)}$ | $70.09_{(0.63)}$ | $\underline{70.79}_{(0.61)}$ | $\mathbf{75.62}_{(0.79)}$ | $67.63_{(0.89)}$ |
| | **Average** | 72.79 | **73.86** | 69.35 | 72.91 | 72.98 | 71.92 |
| Mean | BCNB ER | $90.77_{(0.53)}$ | $\underline{92.19}_{(0.10)}$ | $\mathbf{92.61}_{(0.09)}$ | $90.63_{(0.20)}$ | $89.63_{(0.26)}$ | $89.53_{(0.18)}$ |
| | BCNB HER2 | $73.46_{(0.20)}$ | $\underline{76.35}_{(0.30)}$ | $70.98_{(0.27)}$ | $74.50_{(0.25)}$ | $73.50_{(0.23)}$ | $\mathbf{76.59}_{(0.24)}$ |
| | BCNB PR | $83.83_{(0.10)}$ | $\mathbf{84.84}_{(0.20)}$ | $83.62_{(0.20)}$ | $\underline{84.17}_{(0.10)}$ | $83.17_{(0.12)}$ | $84.00_{(0.18)}$ |
| | BRCA ER | $\underline{86.18}_{(0.30)}$ | $\mathbf{88.27}_{(0.30)}$ | $84.24_{(0.35)}$ | $84.78_{(0.32)}$ | $83.78_{(0.33)}$ | $85.07_{(0.30)}$ |
| | BRCA HER2 | $62.59_{(1.00)}$ | $\mathbf{67.59}_{(0.60)}$ | $63.50_{(0.62)}$ | $63.97_{(0.85)}$ | $62.97_{(0.73)}$ | $\underline{66.03}_{(0.84)}$ |
| | BRCA PIK3CA | $60.23_{(1.20)}$ | $58.99_{(0.80)}$ | $58.52_{(0.65)}$ | $\underline{60.53}_{(0.98)}$ | $59.53_{(0.76)}$ | $\mathbf{62.04}_{(1.08)}$ |
| | BRCA PR | $76.36_{(0.50)}$ | $\mathbf{79.54}_{(0.60)}$ | $75.05_{(0.51)}$ | $76.75_{(0.51)}$ | $75.75_{(0.59)}$ | $\underline{78.16}_{(0.54)}$ |
| | GBMLGG C | $\underline{94.34}_{(0.20)}$ | $\mathbf{95.34}_{(0.70)}$ | $93.59_{(0.71)}$ | $93.47_{(0.28)}$ | $92.47_{(0.41)}$ | $92.79_{(0.52)}$ |
| | GBMLGG F | $49.68_{(0.90)}$ | $\underline{51.63}_{(1.20)}$ | $\mathbf{53.02}_{(1.42)}$ | $48.44_{(0.95)}$ | $47.44_{(1.40)}$ | $50.32_{(1.19)}$ |
| | LUNG EGFR | $64.42_{(0.70)}$ | $\mathbf{66.17}_{(1.10)}$ | $62.45_{(1.22)}$ | $60.94_{(1.12)}$ | $61.94_{(1.09)}$ | $\underline{65.14}_{(0.99)}$ |
| | LUNG KRAS | $60.88_{(1.20)}$ | $\underline{61.22}_{(0.50)}$ | $\mathbf{62.58}_{(0.47)}$ | $58.33_{(0.82)}$ | $59.33_{(0.62)}$ | $61.03_{(1.04)}$ |
| | LUNG STK11 | $67.35_{(1.00)}$ | $\underline{74.31}_{(0.70)}$ | $67.83_{(0.57)}$ | $66.95_{(0.69)}$ | $67.95_{(0.79)}$ | $\mathbf{75.09}_{(0.72)}$ |
| | LUNG TP53 | $72.29_{(0.30)}$ | $\mathbf{75.00}_{(0.50)}$ | $\underline{72.46}_{(0.46)}$ | $69.81_{(0.39)}$ | $70.81_{(0.49)}$ | $69.55_{(0.49)}$ |
| | **Average** | 72.49 | **74.73** | 72.34 | 71.79 | 72.41 | 73.49 |
| DSMIL | BCNB ER | $88.84_{(0.80)}$ | $\underline{90.93}_{(0.11)}$ | $90.52_{(0.10)}$ | $\mathbf{91.91}_{(0.58)}$ | $90.91_{(0.78)}$ | $90.55_{(0.63)}$ |
| | BCNB HER2 | $72.09_{(0.50)}$ | $73.39_{(0.10)}$ | $70.21_{(0.11)}$ | $\mathbf{75.60}_{(0.42)}$ | $74.60_{(0.15)}$ | $\underline{75.03}_{(0.35)}$ |
| | BCNB PR | $82.96_{(0.40)}$ | $83.88_{(0.20)}$ | $83.13_{(0.23)}$ | $\underline{84.45}_{(0.27)}$ | $83.45_{(0.39)}$ | $\mathbf{84.68}_{(0.37)}$ |
| | BRCA ER | $\underline{87.46}_{(0.30)}$ | $86.75_{(0.50)}$ | $85.53_{(0.41)}$ | $\mathbf{88.05}_{(0.44)}$ | $87.05_{(0.47)}$ | $86.23_{(0.47)}$ |
| | BRCA HER2 | $60.90_{(0.60)}$ | $\mathbf{65.94}_{(0.70)}$ | $60.38_{(0.74)}$ | $64.36_{(0.71)}$ | $63.36_{(0.71)}$ | $\underline{64.76}_{(0.73)}$ |
| | BRCA PIK3CA | $\mathbf{61.30}_{(0.70)}$ | $\underline{60.96}_{(0.20)}$ | $53.64_{(0.21)}$ | $60.54_{(0.56)}$ | $59.54_{(0.56)}$ | $60.57_{(0.51)}$ |
| | BRCA PR | $78.00_{(0.80)}$ | $\mathbf{79.05}_{(0.20)}$ | $76.26_{(0.17)}$ | $\underline{78.33}_{(0.80)}$ | $77.33_{(0.63)}$ | $75.76_{(0.21)}$ |
| | GBMLGG C | $\mathbf{94.88}_{(0.40)}$ | $\underline{94.80}_{(0.47)}$ | $89.13_{(0.40)}$ | $92.76_{(0.41)}$ | $91.76_{(0.44)}$ | $92.82_{(0.45)}$ |
| | GBMLGG F | $49.53_{(2.40)}$ | $\underline{50.68}_{(0.70)}$ | $51.31_{(0.73)}$ | $\mathbf{53.84}_{(2.30)}$ | $52.84_{(1.31)}$ | $51.22_{(1.56)}$ |
| | LUNG EGFR | $63.93_{(1.60)}$ | $\underline{65.51}_{(1.30)}$ | $59.78_{(1.32)}$ | $63.39_{(1.47)}$ | $64.39_{(1.57)}$ | $\mathbf{65.39}_{(1.33)}$ |
| | LUNG KRAS | $\mathbf{59.21}_{(0.40)}$ | $58.10_{(0.70)}$ | $51.82_{(0.61)}$ | $55.59_{(0.61)}$ | $56.59_{(0.65)}$ | $\underline{59.14}_{(0.57)}$ |
| | LUNG STK11 | $65.65_{(1.40)}$ | $\underline{68.61}_{(0.40)}$ | $65.74_{(0.37)}$ | $71.66_{(0.55)}$ | $\mathbf{72.66}_{(1.29)}$ | $70.37_{(0.38)}$ |
| | LUNG TP53 | $\underline{71.31}_{(0.80)}$ | $\mathbf{71.88}_{(0.70)}$ | $70.70_{(0.74)}$ | $67.29_{(0.79)}$ | $68.29_{(0.77)}$ | $71.06_{(0.78)}$ |
| | **Average** | 72.00 | 73.11 | 69.86 | 72.91 | **73.52** | 72.89 |

Table A9: **Performance comparison between pathology-specific MoE methods**. M4 has a distinct MoE-based architecture, while MAMMOTH (+Ours) and PaMoE are added on top of ABMIL. Best performance is shown in **bold**, second best is underlined

| | | ABMIL | | | M4 |
| | | Base | + Ours | + PaMoE | Base |
|---|---|---|---|---|---|
| **Tissue** | BRACS C | 67.10 (1.20) | **72.70** (1.40) | 70.52 (1.42) | 62.27 (1.32) |
| | BRACS F | 42.84 (2.50) | 46.12 (2.40) | 43.29 (2.27) | 44.92 (2.71) |
| | EBRAINS C | 86.10 (1.10) | **89.98** (0.70) | 89.95 (0.62) | 86.09 (0.91) |
| | EBRAINS F | 67.20 (1.00) | **72.40** (1.20) | 66.68 (1.19) | 65.86 (1.34) |
| | NSCLC | 94.68 (0.10) | 94.68 (0.10) | 94.68 (0.10) | 94.54 (0.12) |
| | PANDA | 93.12 (0.20) | **94.28** (0.20) | 91.40 (0.22) | 91.48 (0.25) |
| | **Average** | 75.17 | **78.36** | 76.09 | 74.19 |
| **Molecular** | BCNB ER | 90.38 (0.20) | 92.25 (0.10) | **93.04** (0.08) | 90.23 (3.76) |
| | BCNB HER2 | 73.05 (0.40) | 74.70 (0.40) | 72.28 (0.47) | 74.48 (3.68) |
| | BCNB PR | 82.48 (0.40) | **85.84** (0.50) | 83.81 (0.54) | 83.97 (3.61) |
| | BRCA ER | 86.93 (0.30) | **87.94** (0.30) | 87.71 (0.35) | 87.04 (4.66) |
| | BRCA HER2 | 64.35 (1.10) | **68.35** (0.80) | 61.11 (0.88) | 66.32 (4.27) |
| | BRCA PIK3CA | 60.23 (0.70) | 59.55 (0.50) | 59.55 (0.44) | **61.38** (2.81) |
| | BRCA PR | 76.37 (0.30) | 78.77 (0.50) | 76.53 (0.57) | 78.31 (4.92) |
| | GBMLGG C | 91.82 (0.40) | **96.19** (0.40) | 95.79 (0.46) | 92.46 (0.51) |
| | GBMLGG F | 51.89 (1.30) | 52.22 (1.70) | **54.15** (1.59) | 53.01 (1.88) |
| | LUNG EGFR | 61.27 (1.20) | 63.68 (1.20) | 67.25 (1.37) | 65.03 (3.36) |
| | LUNG KRAS | 58.06 (0.70) | 59.40 (1.50) | 59.52 (1.60) | 59.24 (3.61) |
| | LUNG STK11 | 76.41 (1.10) | 74.36 (1.50) | 75.41 (1.65) | 74.13 (7.44) |
| | LUNG TP53 | 72.43 (1.10) | **76.20** (0.70) | 70.22 (0.61) | 73.26 (6.04) |
| | **Average** | 72.74 | **74.57** | 73.57 | 73.76 |

