# OpenReview forum: "Mixture of Mini Experts: Overcoming the Linear Layer Bottleneck in Multiple Instance Learning"
_ICLR.cc/2026/Conference — ICLR 2026 Poster_

### Official Review · Reviewer_DHSJ · 2025-10-30

**Soundness:** 3
**Presentation:** 3
**Contribution:** 3
**Rating:** 6
**Confidence:** 5

**Summary:**

This paper explores the performance bottleneck on general-purpose patch features re-embedding, where MAMMOTH, a MoE module, is designed to replace the specific linear layer in MIL to re-embed general-purpose patch features to a set of specialized features.

**Strengths:**

1. This work is well-motivated. It offers an insight about general-purpose patch features re-embedding, and based on this insight, a MoE module tailored to WSI problems is proposed.
2. Experiments are comprehensive, including thorough ablation studies and extensive comparisons on 19 datasets.
3. The motivation and effectiveness can be intuitively validated with the visualization, which provides good interpretability.

**Weaknesses:**

1. The exploration of why it works is lacking. Although the story is well-told, and the motivations align with the design, I'm still confused about why it achieves the intended purpose. For example, even though features are continuously partitioned into different heads, slots, and experts, why are they able to learn distinct things without explicit constraints?
2. The experimental comparison to R2T-MIL [1] is missing. The core insights are similar to what R2T-MIL claimed, which also re-embedded general-purpose patch features into the new ones. The difference is that feature merging of R2T-MIL is based on regional patches, while MAMMOTH relies on morphological similarities. Therefore, I think the authors should compare R2T-MIL on top of various MIL.

[1] Feature Re-Embedding: Towards Foundation Model-Level Performance in Computational Pathology, CVPR, 2024.

**Questions:**

See weakness.

---

> ### Author Response · Authors · 2025-11-23
> **Response 1/2**
>
> > W1: The exploration of why it works is lacking. Although the story is well-told, and the motivations align with the design, I'm still confused about why it achieves the intended purpose. For example, even though features are continuously partitioned into different heads, slots, and experts, why are they able to learn distinct things without explicit constraints?
>
> We thank the reviewer for highlighting this important detail. While we demonstrate that specialization occurs via pathologist-based assessment (**Figures 3, A3-A7**) and have added quantitative measures of specialization (**Figure A8**), we agree that uncovering insights into the mechanism behind this specialization would provide valuable intuition into why MAMMOTH is effective.
>
> To answer this, we traced the training trajectory of the top 10 slots associated with disease-specific pathology terms (tumor cells, alveoli, lymphocytes, red blood cells, etc). This slot association was identified using the vision-language model, MUSK, and described in our response to Reviewer 1BQN, as well in Lines 976 - 995. Analyzing how the routing scores for each histologic concept changed over the course of training (**Figure A9**), we make the following observations:
>
> **1. Specialization begins at initialization.**
> We observe that in nearly every case, the specialized slots exhibit higher than average relative routing importance for their target concepts at initialization. We repeated this experiment with 5 different random seeds, and this trend was ubiquitous across initialization weights and terms.
>
> **2. Specialization sharply increases during the first epoch and remains high for the rest of training.**
> This trend persisted across initialization schemes and histological terms. Similarly, the bottom 10 slots associated with a concept at the end of model training would sharply decrease in relevance score after the first epoch. The fact that this specialization emerges so rapidly is a strong signal that specialized processing is immediately advantageous for the model's training.
>
> We hypothesize that this rapid and consistent specialization stems from MAMMOTH's ability to circumvent competing gradients. In a standard shared layer, the model must reconcile updates from distinct morphologies—for example, texture features critical for identifying tumor cells might be irrelevant or distracting when processing stroma. By routing these concepts to different experts, MAMMOTH prevents these feature updates from interfering with one another. The rapid reinforcement we observe suggests the model leverages slight initialization asymmetries to separate these conflicting signals early on, leading to the consistent specialization we observe both qualitatively (**Figures 3, A3-A7**) and quantitatively (**Figures A8-A9**). We have added extensive discussion from these results to the **Interpretability** section (Lines 391-399).

---

> ### Author Response · Authors · 2025-11-23
> **Response 2/2**
>
> > W2. The experimental comparison to R2T-MIL [1] is missing. The core insights are similar to what R2T-MIL claimed, which also re-embedded general-purpose patch features into the new ones. The difference is that feature merging of R2T-MIL is based on regional patches, while MAMMOTH relies on morphological similarities. Therefore, I think the authors should compare R2T-MIL on top of various MIL.
>
> We have added extensive comparisons between MAMMOTH and R2T-MIL (i.e RRT) across all 8 MIL models and 19 evaluation tasks, with the results summarized in the public comment to all reviewers. The results highlight that MAMMOTH substantially outperforms RRT, with an average performance of 74.69 compared to 72.38 across all MIL methods and tasks.
>
> In addition to R2T, we have introduced 2 additional baselines (Querent and MIL Dropout) for pre-aggregation modules to evaluate the utility of MAMMOTH's re-embedding. Similar to R2T-MIL, Querent performs spatial aggregation to enable query-aware local self-attention. While MAMMOTH aims to combat overfitting through parameter efficiency, MIL Dropout does so through an attention-based regularizer for instance dropout.
>
> As shown in **Table 4** and **Tables A6-A8**, MAMMOTH consistently performs best across all MIL methods, suggesting that morphology-aware merging combined with expert-specific processing provides distinct benefits over spatial aggregation. We have updated the Related Works section (Lines 146-155) to discuss R2T-MIL and these distinctions in detail.

---

> > ### Comment · Reviewer_DHSJ · 2025-11-26
> >
> > Appreciate the impressive interpretation provided, which addressed most of my concerns. Regarding the comparison to Pre-aggregation approaches, I wonder which MIL method is applied in the experiments presented in the public comment? To ensure fairness, I believe it should be consistent with the one used in MAMMOTH.

---

> ### Author Response · Authors · 2025-11-26
>
> We are pleased to hear that our interpretation was found helpful, and we thank the reviewer for encouraging us to perform a detailed investigation into the mechanism behind MAMMOTH's specialization.
>
> > Regarding the comparison to Pre-aggregation approaches, I wonder which MIL method is applied in the experiments presented in the public comment? To ensure fairness, I believe it should be consistent with the one used in MAMMOTH.
>
> The performance presented in the public comment was **averaged across all 8 MIL methods** presented in the main manuscript. This was done to ensure the most robust and fair comparison possible with MAMMOTH, testing across a wide array of experimental conditions and aggregation strategies.
>
> For a detailed, method-by-method breakdown, the performance of each individual MIL method across all tasks and pre-aggregation methods is now displayed in the newly added **Table A6, Table A7**, and **Table A8** in the **Appendix**.
>
> We thank the reviewer for their prompt response and hope this confirms the consistency and rigor of our comparisons.

---

> > ### Comment · Reviewer_DHSJ · 2025-11-26
> >
> > Thanks to the authors' prompt response. All of my concerns have been addressed. I will maintain my positive rating.

---

> > > ### Author Response · Authors · 2025-12-03
> > > **Further Characterizing MAMMOTH Specialization**
> > >
> > > We thank the reviewer for their time and are encouraged to hear that all concerns have been addressed. Our initial results on training dynamics made us interested in the reason behind the observed **specialization at initialization** and the **rapid specialization within the first epoch**, and we are pleased to provide extensive additional results below.
> > >
> > > **Rapid specialization within the first epoch**: The rapid specialization seen in the first epoch implies that favorable training dynamics reinforce the specialization established at initialization. To study this further, we examined (1) the degree of gradient similarity between heterogeneous instances in standard single-expert settings (using either a standard linear layer or a single-expert MAMMOTH layer); and (2) the similarity between gradients when using MAMMOTH routing.
> > >
> > > For the first analysis, we performed K-means clustering with 8 clusters to identify heterogeneous instances and compared their gradients at the task-specific layer. In both settings, we find inter-cluster gradient updates had significantly lower cosine similarity than intra-cluster gradients, shown in **Table R1** below and **Figure A10 A-B** in the updated manuscript. This suggests that **the standard linear layer suffers from Instance Gradient Interference, where heterogeneous instances yield conflicting gradient updates**. These conflicting gradient updates can impede the quality of the final model [1][2], and highlights the potential for decoupling gradient updates between heterogeneous instances.
> > >
> > >
> > > **Table R1**. *Median cosine similarity between gradient updates for the task-specific layer for instances from the same cluster (Intra-cluster) and between different clusters (Inter-cluster)*
> > > | Layer | Task | Intra-cluster | Inter-cluster | Difference |
> > > | :--- | :--- | :--- | :--- | :--- |
> > > | **Linear Layer** | LUNG TP53 | 0.49 | 0.28 | -0.21 |
> > > | **Linear Layer** | EBRAINS | 0.35 | 0.17 | -0.18 |
> > > | **Linear Layer** | BRACS | 0.42 | 0.26 | -0.16 |
> > > | **Linear Layer** | PANDA | 0.32 | 0.18 | -0.14 |
> > > | **Single Expert** | LUNG TP53 | 0.55 | 0.37 | -0.18 |
> > > | **Single Expert** | EBRAINS | 0.65 | 0.45 | -0.20 |
> > > | **Single Expert** | BRACS | 0.53 | 0.41 | -0.12 |
> > > | **Single Expert** | PANDA | 0.49 | 0.41 | -0.08 |
> > >
> > >
> > > 2) For the second analysis, we investigate the gradient similarity within each expert when instances are routed to multiple experts (E=30) and report the average cosine similarity within each expert. To ensure comparability, we benchmarked this against MAMMOTH with a single expert. We find that routing instances to multiple experts significantly improves gradient similarity, leading to significant improvements in cosine similarity in all four tasks investigated. Boxplots of these results are shown in **Figure A10 C**, and the median values are reported below.
> > >
> > > **Table R2:** *Cosine similarity between gradient updates for MAMMOTH with a single expert versus 30 experts.*
> > > | Layer | Task | Single Expert | 30 experts | Difference |
> > > | :--- | :--- | :--- | :--- | :--- |
> > > | **MAMMOTH** | LUNG TP53 | 0.49 | 0.78 |+ 0.29 |
> > > | **MAMMOTH** | EBRAINS | 0.48 | 0.76 | +0.28 |
> > > | **MAMMOTH** | BRACS | 0.41 | 0.71 | +0.30 |
> > > | **MAMMOTH** | PANDA | 0.44 | 0.73 | +0.29 |
> > >
> > > **Specialization at initialization**
> > > We find that the K-means clusters encoded by the pathology foundation model are largely preserved after the randomly initialized linear projection $W$ responsible for routing patches to experts. For each of the six tasks, we selected 100 slides for analysis. For each slide, we performed K-means clustering with $K=8$ across all frozen feature instances in that slide. We then used these labels to examine the Adjusted Rand Index (ARI) after feeding the frozen features through a randomly initialized linear projection.
> > >
> > > Qualitative results according to t-SNE and quantitative results according to ARI are shown in **Figure A11** and **Figure A12**, respectively. These results indicate that the initial specialization observed in **Figure A9** arises due to MAMMOTH’s ability to preserve the semantic clusters learned by the pathology foundation model.
> > >
> > > **Table R3**. *Quantitative analysis of cluster preservation after linear projection.*
> > > |Task | Median Adjusted Rand Index|
> > > | :---: | :---: |
> > > | EBRAINS-C | $0.85$ |
> > > | LUNG TP53 | $0.77$ |
> > > | BRACS-C| $0.80$ |
> > > | BRCA-PR| $0.82$ |
> > > | GBMLGG-C| $0.85$ |
> > > | PANDA | $0.78$ |
> > >
> > > [1] Liu, B., Liu, X., Jin, X., Stone, P. & Liu, Q. Conflict-Averse Gradient Descent for Multi-task learning. in *Advances in Neural Information Processing Systems* vol. 34 18878–18890
> > >
> > > [2] Yu, T. et al. Gradient Surgery for Multi-Task Learning. in Advances in Neural Information Processing Systems vol. 33 5824–5836 (Curran Associates, Inc., 2020).

---

### Official Review · Reviewer_HDrx · 2025-10-30

**Soundness:** 3
**Presentation:** 4
**Contribution:** 3
**Rating:** 8
**Confidence:** 4

**Summary:**

The authors propose MAMMOTH, a multihead mixture of experts for classification of gigapixel pathology images, that can work with most MIL methods. The MoE mechanism differs from variants in the literature in that it uses slot-based pooling of the input patches, with each slot effectively capturing a morphological concept. Hence the bag size and its interpretation changes after this layer. A low-rank decomposition is used for the slot transformation in each expert to ensure parameter efficiency. The approach is demonstrated on several standard histology datasets across several classification tasks and on top of 8 widely used MIL methods. A thorough ablation study analyzes the impact of specific contributions (MoE method, number of heads, slot transformation, patch encoder, etc.).

**Strengths:**

1. Novelty: There is good methodological novelty in the context of WSI classification / MIL, as the specifics of the MoE mechanism are original.

2. Clarity: The method is clearly described and the paper well-positioned w.r.t. the literature

3. Significance: Slot-based pooling could be beneficial to interpretability, by reducing bag size and allowing each patch embedding in the initial bag to be linked to its most similar morphological concept.

4. Quality: There is a thorough ablation study + a study of data and inference efficiency + generalizability across diverse patch encoders

**Weaknesses:**

One concern with this paper is that the proposed MoE layer adds complexity to the MIL strategy, despite careful control of parameter efficiency.

This needs to be justified by strong and robust performance gains. For some datasets where no mention is made of cross-validation, e.g. BRACS and PANDA, the exact experimental setup is still unclear to me: is it a single train/test split, 1 run, 1000 bootstraps of the test set on this single run? As very large differences can occur across multiple runs in MIL for histopathology. If so, especially for these datasets, statistical significance tests on the performance gains would be welcome.

**Questions:**

How do the authors select the number of experts and slots robustly in a real-world scenario? Quite substantial performance degradation can be observed for some settings in Figure A2.

---

> ### Author Response · Authors · 2025-11-23
>
> We sincerely thank the reviewer for their thoughtful assessment. We are encouraged by the reviewer’s positive reception of the methodological novelty and clarity of our work. We have addressed raised comments below.
>
> > W1: For some datasets... is it a single train/test split, 1 run, 1000 bootstraps... statistical significance tests on the performance gains would be welcome.
>
> We confirm that for the BRACS and PANDA datasets, the results represent 1,000 bootstraps on the test set of a single split, as described on Line 311. We agree that statistical testing is valuable to validate these improvements given the variance often observed in MIL. We will be certain to incorporate statistical significance tests into the final manuscript to rigorously evaluate the robustness of MAMMOTH's performance gains over the baselines.
>
> ***
> > C1: How do the authors select the number of experts and slots robustly in a real-world scenario? quite substantial performance degradation can be observed for some settings in **Figure A2**.
>
> While the strictly optimal number of experts and slots varies between tasks, our ablation study in **Figure A2** demonstrates that there is a wide "safe zone" of configurations that yield consistent performance gains. Regarding the degradation observed in **Figure A2**: this occurs at extreme values (e.g., excessively high expert counts) which were explored specifically to identify the upper bounds of the architecture. In standard practice, pathology MoEs typically utilize a much smaller range (2-8 experts) [1][2].
> While recent work has begun to explore adaptive mechanisms for dynamically selecting slots [3], these approaches often introduce significant additional complexity. In this work, *our primary goal was to provide a robust, "plug-and-play" module*. We demonstrate that a fixed configuration of 30 experts, 300 slots, and 16 heads consistently outperforms the standard linear layer and other baselines across a diverse array of tasks.
>
> We believe there is substantial practical value in a configuration that performs reliably across varying distributions, even if it is not the theoretical optimum for every single dataset. This stability reduces the burden of extensive hyperparameter tuning for future users, while our ablation in Figure A2 identifies a reasonable search space if more specific tuning is required.
>
> [1] Li, J. et al. M4: Multi-Proxy Multi-Gate Mixture of Experts Network for Multiple Instance Learning in Histopathology Image Analysis. (2024).
> [2] Wu, J. et al. Learning Heterogeneous Tissues with Mixture of Experts for Gigapixel Whole Slide Images. in 5144–5153 (2025).
> [3] Fan, K. et al. Adaptive Slot Attention: Object Discovery with Dynamic Slot Number. CVPR (2024).

---

> > ### Comment · Reviewer_HDrx · 2025-11-26
> >
> > Thanks to the authors for their thorough responses to reviewers' comments. This confirms my assessment of the paper, I will maintain a score of 8: Accept.

---

> > > ### Author Response · Authors · 2025-11-26
> > >
> > > We thank the reviewer for their acknowledgment of our thorough response. We are pleased that it confirmed your positive assessment of our work.

---

### Official Review · Reviewer_FYXj · 2025-10-31

**Soundness:** 3
**Presentation:** 3
**Contribution:** 3
**Rating:** 6
**Confidence:** 5

**Summary:**

The paper introduces MAMMOTH, a parameter-efficient multihead mixture of experts (MoE) designed to replace the often-overlooked task-specific linear layer in multiple instance learning (MIL) for whole-slide image (WSI) classification. Traditional MIL pipelines rely on a single linear transformation for all patch embeddings, which limits the ability to capture diverse histomorphological patterns. MAMMOTH addresses this by partitioning embeddings into multiple heads, applying soft expert assignments for stable training, and using low-rank decomposition with weight sharing to maintain efficiency. This design reduces thousands of noisy patch embeddings into a compact, interpretable set of task-specific features. Experiments across 8 MIL methods and 19 tasks show consistent improvements, with MAMMOTH outperforming baselines in 130 of 152 configurations and achieving an average +3.8% performance gain. Interpretability analyses confirm that experts specialize in distinct morphological concepts, while ablations highlight the effectiveness of each design choice. MAMMOTH thus enhances accuracy, efficiency, and interpretability in computational pathology.

**Strengths:**

1. Originality: The paper highlights a neglected component in the MIL pipeline: the task-specific linear layer. By replacing it with MAMMOTH, a parameter-efficient multihead mixture-of-experts module, the authors introduce a novel perspective. This reframes the performance bottleneck in WSI classification, making the contribution both original and conceptually impactful.

2. Quality: The work is technically rigorous. The authors design MAMMOTH with innovations like soft expert assignment, low-rank decomposition, and multihead partitioning to address the instability and parameter inefficiency of traditional MoE. Extensive experiments across 8 MIL methods and 19 tasks (morphological and molecular) validate its robustness, with consistent improvements in 130 out of 152 configurations. Interpretability studies involving pathologists further strengthen the claims by showing clear expert specialization in distinct histomorphological features.

3. Clarity:The paper is generally well-structured, moving from motivation to method to results in a logical flow. Figures (e.g., embedding space visualizations, routing heatmaps) effectively illustrate key points. Ablation studies and efficiency comparisons are clearly presented, helping readers understand the contribution of each component

4. Significance: Mammoth is presented as a plug-and-play module that consistently boosts performance across a wide range of tasks and backbone architectures without increasing parameters. This makes it a highly practical and immediately useful contribution for researchers and practitioners.

**Weaknesses:**

1. Fixed hyperparameter configuration. MAMMOTH is evaluated with a fixed number of experts, heads, and slots across tasks. While effective, this may not reflect task-specific optimal configurations. Explore adaptive mechanisms to dynamically adjust expert/head/slot counts depending on task complexity or data size would be better.

2. Limited scope of tasks. Although MAMMOTH was validated on 19 tasks spanning morphology and biomarker prediction, the paper does not extend to other clinically critical tasks such as survival prediction or multimodal fusion with genomics and radiology. These are central to real-world deployment.

3. Marginal gains on simple tasks. On simple binary tasks with strong baselines (e.g., NSCLC subtyping), MAMMOTH shows little or no improvement.

**Questions:**

1. The author employed a fixed configuration of experts, heads, and slots across tasks. How sensitive is MAMMOTH to these choices?

2. The evaluation focuses on classification (morphological and biomarker tasks). Do the authors anticipate that MAMMOTH would extend naturally to other clinically relevant tasks such as survival prediction?

3. The expert specialization examples are compelling, but mostly qualitative. Is there a way to systematically quantify interpretability, for instance, by measuring overlap between expert clusters and pathologist-annotated regions of interest?

4. The reviewer is also interested in performance comparison to pathology-specific MoE methods such as [1][2] and existing plug-and-play module for the linear layer such as [3]

[1] Learning Heterogeneous Tissues with Mixture of Experts for Gigapixel Whole Slide Images
[2] M4: Multi-proxy multi-gate mixture of experts network for multiple instance learning in histopathology image analysis
[3] How Effective Can Dropout Be in Multiple Instance Learning ?

---

> ### Author Response · Authors · 2025-11-23
> **Response 1/3**
>
> We sincerely thank the reviewer for their positive assessment of our work, particularly for highlighting the originality of addressing the neglected task-specific linear layer and recognizing the technical rigor of our design. We appreciate the constructive feedback, which has prompted us to expand our evaluation to include survival analysis and quantitative interpretability.
>
> >W1. “The reviewer is also interested in performance comparison to pathology-specific MoE methods such as [1][2] and existing plug-and-play module for the linear layer such as [3]”
>
> We appreciate the references to related work. To address this, we have conducted benchmarks on all of the suggested methods: 1) Heterogeneous Tissues MoE (PaMoE), 2) M4, and 3) MIL Dropout, as well as two additional plug-and-play modules, 4) RRT-MIL and 5) Querent. We evaluated these methods across all 8 MIL aggregators and 19 classification tasks. We would like to note that (1) is the paper for PaMoE, which has been extensively benchmarked in our initial submission (**Table 4** and **Table A5**). The complete results (now in **Table 4** and **Tables A5-A7**) show that while other plug-and-play modules offer improvements over the baseline, MAMMOTH consistently achieves the highest aggregate performance across this diverse set of benchmarks.
>
> ***
> > W2.  “The author employed a fixed configuration of experts, heads, and slots across tasks. How sensitive is MAMMOTH to these choices?"
>
> Regarding sensitivity to experts, heads, and slots across tasks, we would like to highlight our extensive ablations included in the **Figures A1-A2** and described in the main text (Lines 510-529). We examined performance across 275 unique configurations (spanning 4–96 experts, 2–32 heads, and 1–500 slots). **These results indicate a wide range of hyperparameters for which MAMMOTH remains effective**, alongside general guidelines, such as pairing low expert counts (<=4 experts) with low total slots (50-100), while higher expert counts (>24 experts) pair well with 200-400 total slots.
>
> > W3. Explore adaptive mechanisms to dynamically adjust expert/head/slot counts...”
>
> While we agree that adaptive parameter selection is an interesting direction for future research, we emphasize that MAMMOTH’s fixed configuration is designed to serve as a robust, "plug-and-play" foundation. The fact that MAMMOTH provides consistent benefits across 22 total tasks (now including survival) and varying training data sizes (Figure 4), without requiring task-specific tuning, is a key strength illustrating its flexibility.
>
> This establishment of a strong fixed configuration is an important and standard first step for introducing new architectures (e.g., Slot Attention[1], Soft MoE[2], PANTHER[3]), while adaptive mechanisms typically warrant a dedicated follow-up study (e.g., Adaptive Slot Attention[4]). We have described adaptive selection as a promising avenue for future work in the text (Lines 538-539).
>
> [1] Locatello, F. et al. Object-Centric Learning with Slot Attention. in Advances in Neural Information Processing Systems vol. 33 11525–11538
> [2] Puigcerver, J., Ruiz, C. R., Mustafa, B. & Houlsby, N. From Sparse to Soft Mixtures of Experts. in The Twelfth International Conference on Learning Representations, ICLR 2024, Vienna, Austria, May 7-11, 2024
> [3] Song, A. H. et al. Morphological Prototyping for Unsupervised Slide Representation Learning in Computational Pathology. in Proceedings of the IEEE/CVF Conference on Computer Vision and Pattern Recognition (2024).
> [4] Fan, K. et al. Adaptive Slot Attention: Object Discovery with Dynamic Slot Number. in 2024 IEEE/CVF Conference on Computer Vision and Pattern Recognition (CVPR) 23062–23071 (2024).

---

> ### Author Response · Authors · 2025-11-23
> **Response 2/3**
>
> > W4. “Do the authors anticipate that MAMMOTH would extend naturally to other clinically relevant tasks such as survival prediction?”
>
> We thank the reviewer for highlighting the importance of survival prediction. While the module was originally validated on classification, we agree that MAMMOTH may also be suitable for survival tasks. We have extended our evaluation to include 4 survival tasks across 5 datasets: SurGen (colorectal cancer), TCGA-BRCA (breast cancer), and Lung Cancer survival (LUAD & LUSC across TCGA, CPTAC, and NLST cohorts).
>
> We evaluate each of these tasks using stratified 5-fold cross-validation. MAMMOTH demonstrates strong generalization to these survival tasks, and these results have been added to the main text (**Table 3**). Regarding multimodal fusion, MAMMOTH has demonstrated the ability to consistently improve the histologic representation across a wide range of architectures and tasks. Given that multimodal pipelines frequently use MIL for its WSI branch, we believe MAMMOTH is well-suited to improve this representation as well. However, the evaluation of these multimodal datasets is orthogonal to our work, which focuses entirely on histology. We consequently leave this evaluation for future work.
>
> ***
> > W5. “On simple binary tasks with strong baselines (e.g., NSCLC subtyping), MAMMOTH shows little or no improvement.”
>
> We acknowledge that on simpler binary tasks where baselines approach performance saturation (e.g., NSCLC subtyping), the margin for improvement is inherently limited, and specialized processing may not be necessary. Nonetheless, we find that MAMMOTH preserves performance on the simple task (+0.56 average AUC), while achieving substantial benefit on more complex tasks such as EBRAINS 30-class subtyping (+4.3% average balanced accuracy). These results provide valuable insight into MAMMOTH’s utility: it is neutral on tasks that do not require additional capacity, and highly beneficial when task complexity demands richer, task-specific representations.

---

> ### Author Response · Authors · 2025-11-23
> **Response 3/3**
>
> > W6. “The expert specialization examples are compelling, but mostly qualitative. Is there a way to systematically quantify interpretability...?”
>
> We thank the reviewer for the suggestion to obtain quantitative measures of interpretability. To move beyond qualitative examples, we have implemented a systematic, quantitative interpretability protocol. We utilized a pathology-specific Vision-Language Model, MUSK, to score patches based on their similarity to a predefined set of histological concepts, obtained by querying Gemini 2.5 Pro based on the tissue type and disease of interest. We then measured the correlation between specific expert-slot pairs and these concepts across the entire dataset.
>
> This analysis provides quantitative validation that MAMMOTH's slots specialize in distinct morphological patterns (e.g., tumor subtypes, stroma, necrosis) consistent with pathologist annotations. These results are now presented in **Figure A8**. Additionally, this quantitative assessment elucidates numerous interesting properties of expert specialization:
>
>  1) **Slots within an expert tend to have correlated, but not identical, levels of activation**. This is observable from the **Relative Slot Importance** component of **Figure A8**, in which experts with low activation tend to have low activation among all slots, suggesting that slots within an expert tend to specialize in similar or related concepts. We have included this discussion in the **Figure A8** caption.
> 2) **Experts which have specialized on concepts of high diagnostic importance assign near-zero weights to regions of low diagnostic importance**. For instance, in **Figure A8**, the top 7 experts attending to tumor cells are the 7 lowest experts for red blood cells.
>
> 3) **This approach highlights the key experts and slots correlated to a concept across all WSIs in the dataset.** Consequently, it allows us to deterministically and verifiably select the most representative slot for visualization. As requested by reviewer 1BQN, we have revisited the qualitative figures for **Figure 3** and **Figures A3-A6** using the key words already identified in the figure. In all supplemental figures, this pipeline yielded the same expert and slot that our pathologists had selected, with the exception of the **Lymphocytes** key term, which we have updated with a new slot and top patches.

---

### Official Review · Reviewer_1BQN · 2025-10-31

**Soundness:** 3
**Presentation:** 3
**Contribution:** 2
**Rating:** 6
**Confidence:** 4

**Summary:**

This paper targets a neglected but ubiquitous component in Multiple Instance Learning (MIL) pipelines for whole-slide image (WSI) classification: the task-specific linear layer that maps generic patch embeddings to task-optimized representations prior to aggregation. The authors hypothesize that this layer is a key performance bottleneck and propose MAMMOTH, a parameter-efficient, multi-head mixture-of-experts (MoE) module that replaces this linear transformation. MAMMOTH partitions the input embedding across heads, routes patches to slot-specific prototypes via soft attention, and applies low-rank expert-specific transformations before re-concatenation. Across 8 MIL methods and 19 tasks (both morphological classification and molecular biomarker prediction), MAMMOTH improves performance in 130 of 152 configurations, with an average percent change of +3.8% overall (larger on morphology), and even enables simple aggregations (mean/max pooling) to surpass more sophisticated MIL baselines when equipped with MAMMOTH. The authors also provide ablations for design choices, efficiency and data-efficiency analyses, and qualitative interpretability showing specialization of experts/slots to morphological concepts.

**Strengths:**

- Broad, careful empirical validation across 8 MIL methods and 19 tasks (morphology and biomarkers), with consistent improvements in 130/152 configurations and larger gains on morphological tasks.
- Ablations isolate the contributions of heads, slots, and low-rank experts; interpretability analyses show morphologically coherent routing/specialization; runtime/data-efficiency comparisons suggest favorable trade-offs versus sparse MoE variants.
- Solid training details (optimizer/schedule/regularization) and cross-validation protocols.

**Weaknesses:**

The paper has limitations in its positioning and comparative baselines, which affect the rigor and credibility of its conclusions. The main issues are as follows:

1. Lack of direct comparison with strong, relevant baselines: The proposed module, characterized as “feature re-embedding” or task-layer replacement prior to aggregation, does not include direct comparisons with recent, closely related works in WSI MIL, weakening the claims of its relative importance. Key baselines include:

- Feature Re-Embedding (CVPR 2024; Re-embedded Regional Transformer, RRT-MIL) [1].
- Query-aware dynamic long-context modeling (ICML 2025; “Context Matters,” Querent) [2].

Without comparisons under matched experimental settings, the conclusion that “the task layer matters more than aggregation” appears overly strong.

[1] Tang, Wenhao, et al. "Feature re-embedding: Towards foundation model-level performance in computational pathology." Proceedings of the IEEE/CVF conference on computer vision and pattern recognition. 2024.
[2] Guo, Zhengrui, et al. "Context Matters: Query-aware Dynamic Long Sequence Modeling of Gigapixel Images." Forty-second International Conference on Machine Learning.

2. Post-hoc selection bias: The interpretability analysis showcases specific expert × slot visualizations reviewed by pathologists, but the paper does not define a deterministic protocol for selecting these slots. This lack of predefined selection criteria makes it unclear which expert × slot should be inspected to observe the claimed morphological types, increasing the risk of cherry-picking and reducing the robustness of the interpretability findings.

**Questions:**

1. In light of the absence of matched-setting comparisons to strong re-embedding baselines, would you consider revising the main claim to: “Under our setup and tasks, task-layer re-embedding yields larger improvements compared to changing aggregators,” contingent on further comparisons and significance testing?

2. How were expert × slot pairs chosen for visualization? Implementing a pre-registered, deterministic selection protocol could enhance transparency. For instance, CLIP-style text prompts or pathology-specific language encoders could be used to rank slots by text-concept similarity (e.g., “necrosis,” “lymphocytic infiltration”) and compare these rankings to pathologist-provided labels.

---

> ### Author Response · Authors · 2025-11-23
> **Response 1/2**
>
> We thank the reviewer for the thorough assessment and constructive suggestions. We appreciate the recognition of our broad empirical validation, careful ablations, and solid training and evaluation protocols. Below we address the two primary concerns: lack of comparison with pre-aggregation modules, and nondeterministic heatmap selection.
> > W1. The proposed module, characterized as “feature re-embedding” or task-layer replacement prior to aggregation, does not include direct comparisons with recent, closely related works in WSI MIL, weakening the claims of its relative importance.
>
>  We agree that comparing against state-of-the-art pre-aggregation modules is essential to contextualize our contributions. We have updated the discussion of **Related Works** (Lines 146-155) to discuss pre-aggregation. We have also now incorporated all suggested pre-aggregation modules, RRT [1], Querent [2], and MIL Dropout [3] (per Reviewer DHSJ), into a unified evaluation across our full experimental suite of 19 tasks and 8 MIL methods.
>
> The full results, covering 456 additional configurations, are included in the response to all reviewers, as well as **Tables A6, A7, and A8**. We have also expanded **Table 4** to introduce a “pre-aggregation module” category for the model-design ablation, indicating how these alternative pre-aggregation procedures perform compared to our proposed module.
>
> Across these matched-setting comparisons, we observe that all three benchmark modules improve over baseline, confirming that the pre-aggregation component is an important intermediary step. Nevertheless, MAMMOTH yields the largest and most consistent improvement compared to these alternative re-embedding modules, substantiating the unique benefits of our specialized approach.
>
> [1] Tang, Wenhao, et al. "Feature re-embedding: Towards foundation model-level performance in computational pathology." Proceedings of the IEEE/CVF conference on computer vision and pattern recognition (2024).
> [2] Guo, Zhengrui, et al. "Context Matters: Query-aware Dynamic Long Sequence Modeling of Gigapixel Images." Forty-second International Conference on Machine Learning (2025).
> [3] Zhu, W. et al. How Effective Can Dropout Be in Multiple Instance Learning ? Forty-second International Conference on Machine Learning (2025).
> ***
> > W2: Without comparisons under matched experimental settings, the conclusion that “the task layer matters more than aggregation” appears overly strong.
>
> While we have performed the additional comparisons, we recognize the reviewer’s concern and have softened and clarified our main claim.
>
> We have updated the abstract to now state:
> “*Across 8 MIL methods and 19 different tasks, we find that this improvement to the task-specific transformation yields higher performance gains than changing to the most effective aggregator.*”
>
> We have similarly updated lines 355 to state:
> "*These results suggest that the linear layer is a bottleneck for performance, with the inclusion of MAMMOTH leading to larger improvements in performance compared to changing MIL architectures.*"

---

> ### Author Response · Authors · 2025-11-23
> **Response 2/2**
>
> > Q1. How were expert × slot pairs chosen for visualization? Implementing a pre-registered, deterministic selection protocol could enhance transparency.
>
> In the initial submission, expert-slot pairs were chosen by visualizing the 30 patches with the highest routing scores for each pair. Our pathologists then identified pairs with distinct morphological features for visualization. When visualizing the top patches for a WSI, we deterministically selected the patches with the highest routing score within that WSI to improve the objectivity of our visualizations and select the most representative patches.
>
> **Updated Selection Protocol**: With the reviewer’s suggestion, we have now implemented a fully deterministic protocol to select these key slots, described as follows:
> 1) We query Gemini 2.5 Pro for 30 histologic terms relating to a given organ’s histology.
> 2) We generate text embeddings with the vision-language pathology foundation model, MUSK.
> 3) We generate MUSK patch embeddings for all WSIs in the dataset of interest.
> 4) We score each patch’s “relatedness” to each histologic term’s text embedding via cosine similarity.
>  5) For each expert-slot pair, we take the weighted sum for the relatedness to a term, weighted by the routing scores to each patch.
> 6) For each term, we select the expert-slot pair with the highest sum for visualization, and select the top-K terms with the highest scores for visualization. In-house pathologists then reviewed these top pairs to ensure accuracy of the results. This process is described in greater detail in the **Appendix** (Lines 976 - 995)
>
> Importantly, this approach highlights the key experts and slots correlated to a concept across all WSIs in the dataset. Consequently, it allows us to deterministically and verifiably select the most representative slot for visualization. As requested by the reviewer, we have revisited the qualitative figures for **Figure 3** and **Figures A3-A6**. From this process, we found that the majority of top expert-slot pairs identified for **Figure 3** and **Figure A3-A6** were consistent with the top pairs identified by our pathologists, with the exception of the top pair for lymphocytes in **Figure 3**. We have updated **Figure 3** with updated patches based on **Expert 13, Slot 2** to reflect this change.
>
> **Quantitative visualization provides insights into specialization**:
> Next, we use these results to obtain measures of quantitative interpretability, shown in **Figure A7** and **Figure A8**. In **Figure A7**, we visualize the routing scores for each slot towards each histologic concept, with the relevance of each expert shown as the sum of slot routing scores. These findings are described in greater depth in our response to reviewer FYXj (See A6). This figure also indicates the most salient slot for each histologic term, providing additional details into the aforementioned deterministic slot selection process for each term.
>
> Additionally, we use these quantifications of term similarity to evaluate how MAMMOTH learns to specialize over the course of training in response to Reviewer DHSJ. These results are shown in **Figure A8**. We observe two insightful and consistent patterns across multiple initializations, terms, and slots: 1) The slots which strongly specialize in a histologic concept consistently begin with a higher initial routing score for that term at initialization. 2) Slot specialization rapidly increases during the first epoch of training and remains elevated across remaining epochs, providing intuition behind how semantically-meaningful specialization emerges and solidifies.
>
> We would like to once again thank the reviewer for their suggestion to perform vision-language-based selection of top slots. While performing deterministic selection of representative heatmaps is not typically done in computational pathology studies, we find that its inclusion greatly strengthens the reliability of our selected examples and flexibility of our quantitative interpretability analysis. We are hopeful that its inclusion will encourage future work to adopt a similar pipeline for enhanced transparency.

---

### Author Response · Authors · 2025-11-23
**Response to all Reviewers**

We sincerely thank all reviewers for their thoughtful and detailed feedback. We are encouraged that the reviewers found the work well-motivated, the design innovative, and the ablations thorough. We have uploaded a revised manuscript with all changes highlighted in blue.

**Summary of Major Revisions**
We address two primary themes raised by multiple reviewers:

1. **Additional Benchmarking:** Comparisons against alternative pre-aggregation schemes (**R1-1BQN, R2-FYXj, R4-DHSJ**) and existing pathology-based MoE methods (**R2-FYXj**).
2. **Explainability & Intuition:** Quantitative explainability schemes (**R1-1BQN, R2-FYXj**) and improved intuition regarding MAMMOTH’s specialization dynamics (**R4-DHSJ**).

Below, we detail the extensive new experimentation conducted to address these points.

**(1) Benchmarking Against Pre-aggregation Schemes & MoEs**
We have extended our results across the full suite of 19 tasks and 8 MIL methods to encompass 6 total pre-aggregation strategies (Linear, MAMMOTH, PaMoE, RRT, MIL dropout, and Querent).

- **Pre-Aggregation Comparison:** As shown in our revised model design ablation (**Table 4**) and detailed per-model tables (**Tables A7-A8**), MAMMOTH achieves the highest performance in **17 out of 19 tasks** averaged across MIL methods. MAMMOTH achieves the highest average performance (**74.69**) compared to all pre-aggregation baselines (Range: 71.76 – 72.97).
- **MoE Comparison:** We explicitly compare MAMMOTH with ABMIL against M4, an MoE using multiple ABMIL layers as distinct experts (**Tables 4** and **A9**). MAMMOTH substantially outperforms M4 (MAMMOTH: **75.77** vs. M4: 73.89).

**(2) Survival Analysis**
We have introduced 4 survival prediction tasks (Breast, Colon, Lung Adenocarcinoma, Lung Squamous Cell Carcinoma) across 6 datasets (TCGA-BRCA, SurGen, TCGA-LUAD, TCGA-LUSC, CPTAC, NLST). Performance with and without MAMMOTH for all 8 MIL methods is provided in **Table 3**.

**(3) Quantitative Explainability via Vision-Language Scoring**
To objectively validate expert specialization, we employed the **MUSK [1]** vision-language foundation model.

- **Method:** We used MUSK to score patches based on relationships with pathology concepts (e.g., tumor cells, smooth muscle, necrosis). We then calculated a weighted average using routing scores between slots and patches.
- **Results:** **Figure A8** displays the degree to which specific slots specialize in these concepts. This allows us to deterministically identify key slots for visualization and quantify the extent of specialization, improving the objectivity of our selected interpretability heatmaps.

**(4) Specialization Dynamics**
Using the slot-concept scores derived above, we analyzed how routing scores evolve during training (**Figure A9**).

- **Findings:** Concept-aligned slots (e.g., those specializing in alveoli or stroma) exhibit higher importance at initialization. This is followed by a sharp increase in specialization within the first epoch, which remains stable thereafter.
- **Conclusion:** This pattern, consistent across 5 trials and 8 terms, suggests MAMMOTH’s specialization is partly driven by differential routing at initialization, which is rapidly reinforced during early training.

[1] Xiang, J. *et al.* A vision–language foundation model for precision oncology. *Nature* **638**, 769–778 (2025).

---
*Performance for Pre-aggregation methods, averaged across 8 MIL methods*
| Task | Base | **MAMMOTH** | PAMoE | RRT | MIL Drop | Querent |
|------|------|-------|--------|------|--------------|----------|
| BRACS C | 63.63 | **71.23** | 63.45 | 60.99 | 62.81 | *64.97* |
| BRACS F | 34.95 | **41.44** | 35.21 | 39.02 | 39.68 | *39.93* |
| EBRAINS C | 86.22 | **89.42** | 86.35 | *87.88* | 86.43 | 85.27 |
| EBRAINS F | 66.87 | **71.20** | 63.11 | *69.16* | 67.39 | 65.28 |
| NSCLC | 93.58 | *94.13* | 93.58 | 93.95 | 94.04 | **94.74** |
| PANDA | 91.75 | **93.25** | 91.03 | *91.87* | 91.26 | 91.83 |
| BCNB ER | 90.31 | **91.90** | 90.71 | 90.56 | *90.92* | 89.72 |
| BCNB HER2 | 72.04 | *74.71* | 70.63 | 73.50 | 73.83 | **75.39** |
| BCNB PR | 82.93 | **84.81** | 82.82 | 83.34 | 83.37 | *83.69* |
| BRCA ER | 86.48 | **88.07** | 85.49 | 86.24 | *86.85* | 86.45 |
| BRCA HER2 | 63.02 | **65.26** | 62.35 | 63.33 | 63.13 | *64.14* |
| BRCA PIK3CA | *59.71* | **59.86** | 57.46 | 58.31 | 58.63 | 59.65 |
| BRCA PR | 77.16 | **78.75** | 75.72 | 76.89 | 76.12 | *77.17* |
| GBMLGG C | *94.04* | **95.16** | 92.66 | 93.91 | 94.01 | 92.59 |
| GBMLGG F | 50.44 | *51.65* | **52.03** | 50.49 | 50.26 | 50.84 |
| LUNG EGFR | 63.35 | **65.11** | 62.67 | 62.86 | 62.99 | *64.67* |
| LUNG KRAS | *59.31* | **60.18** | 58.32 | 55.88 | 57.81 | 58.79 |
| LUNG STK11 | 69.30 | 70.66 | 69.62 | 67.77 | **71.40** | *71.21* |
| LUNG TP53 | 70.85 | **72.34** | 70.16 | 69.20 | *70.85* | 70.14 |
| **Average** | 72.42 | **74.69** | 71.76 | 72.38 | 72.73 | *72.97* |

---

> ### Author Response · Authors · 2025-11-23
> **Survival Results**
>
> **C-index for 5-fold cross-validation of predicting overall survival with MAMMOTH**
> | Task | State | ABMIL | CLAM | TransMIL | Trans. | ILRA | Mean | Max | DSMIL |
> | --- | --- | --- | --- | --- | --- | --- | --- | --- | --- |
> |BRCA  | Base | 58.56 | 61.91 | 58.86 | 56.90 | 57.26 | 58.27 | 56.69 | 59.31 |
> | BRCA | +Ours | 63.98 | 64.36 | 65.02 | 65.23 | 63.94 | 63.18 | 62.70 | 60.43 |
> |**BRCA** | Δ | +5.42 | +2.45 | +6.16 | +8.33 | +6.68 | +4.91 | +6.01 | +1.12 |
> |SURGEN   | Base | 63.67 | 63.93 | 57.94 | 60.59 | 62.29 | 64.03 | 56.53 | 60.17 |
> | SURGEN | +Ours | 65.64 | 64.99 | 63.10 | 63.91 | 64.80 | 64.97 | 59.31 | 65.11 |
> |**SURGEN**  | Δ | +1.97 | +1.06 | +5.16 | +3.32 | +2.51 | +0.94 | +2.78 | +4.94 |
> |LUAD | Base | 58.70 | 58.97 | 56.76 | 58.31 | 57.24 | 59.11 | 55.95 | 58.14 |
> | LUAD | +Ours | 60.12 | 61.89 | 60.97 | 60.49 | 58.10 | 60.56 | 57.18 | 57.99 |
> | **LUAD**  | Δ | +1.42 | +2.92 | +4.21 | +2.18 | +0.86 | +1.46 | +1.23 | -0.15 |
> | LUSC | Base | 56.62 | 55.63 | 52.53 | 55.11 | 54.44 | 56.04 | 49.39 | 51.26 |
> | LUSC | +Ours | 59.30 | 58.91 | 53.48 | 54.67 | 54.48 | 58.30 | 50.12 | 55.69 |
> |**LUSC**   | Δ | +2.70 | +3.28 | +0.94 | -0.44 | +0.03 | +2.26 | +0.73 | +4.43 |

---

> ### Author Response · Authors · 2025-12-03
> **Further Characterizing MAMMOTH Specialization**
>
> We are pleased to provide extensive new mechanistic evidence answering **Reviewer DSHJ**’s question of how MAMMOTH learns to specialize, as well as **Reviewer FYXj**'s request for additional quantitative measures of specialization. Our initial investigation into the dynamics of MAMMOTH specialization revealed two valuable insights into MAMMOTH specialization, that 1) specialization begins at initialization, and 2) that specialization sharply increases during the first epoch of training, which **Reviewer DSHJ** “appreciate[d] the impressive interpretation provided”. We now provide further insights into *why* each of these features occur.
>
> 1. **Mitigation of IGI via Gradient Decoupling**: We identify Instance-Gradient Interference (IGI)—a limitation in standard MIL where shared task-layers produce conflicting per-instance gradients across heterogeneous tissue types. We demonstrate that while standard linear layers suffer from conflicting gradient signals between heterogeneous patches, MAMMOTH experts exhibit significantly smoother gradient flows (**Figure A10**). By routing diverse patches to separate experts, each expert is shielded from the noise of conflicting tissue types, promoting rapid specialization within the first epoch (**Figure A8**).
>
> Specifically, we investigate the gradients during the first epoch of training the task-specific linear layer across four tasks, finding that the cosine similarity between gradients from instances of different clusters is significantly lower (p < 0.001) than the gradients from instances from the same cluster (**Figure A10 A-B**). In contrast, by routing instances to different experts, each expert receives significantly smoother gradient updates (p<0.001, **Figure A10 C**).
>
> 2. **Preservation of Semantic Clusters at Initialization**: MAMMOTH’s routing mechanism effectively capitalizes on the pre-existing feature space of pathology foundation models. We show qualitatively(**Figure A11**) and quantitatively (**Figure A12**) that the K-means clusters generated from the foundation model’s features, which are well understood to encode distinct histologic concepts[1][2][3], are preserved after the initial MAMMOTH projection through $\mathbf{W} \in \mathbb{R}^{(PH \times D}$. This preservation of semantic clusters formed by the foundation model supplies the routing scheme with a strong prior and provides insight into *why* specialization can be observed at initialization (**Figure A8**).
> Taken together with our initial results investigating the dynamics of expert specialization during training, these results provide a comprehensive answer to how MAMMOTH learns to specialize: At initialization, experts specialize by preserving the semantic clusters learned by pathology foundation models. This initial specialization, paired with congruent gradient updates due to expert routing, yields rapid specialization within the first epoch of training. In our investigation, we follow the feedback provided by **Reviewers FYXj** and **1BQN** and ensure that each step in our analysis is deterministic and supported by quantifiable evidence. Additional details are provided in our response to **Reviewer DSHJ**.
> ---
> Finally, we note that the scope of our work has substantially expanded to include survival analysis and a deeper investigation into the linear layer bottleneck. Consequently, we would like to express our intention to rename the paper to better encapsulate our findings, pending approval from the AC and PCs.
>
> **Original**: "Multihead Mixture of Experts for Classification of Gigapixel Pathology Images"
> **Retitled**: "Mixture of Mini Experts: Overcoming the Linear Layer Bottleneck in Multiple Instance Learning"
>
> ---
> [1] Chen, R. J. et al. Towards a general-purpose foundation model for computational pathology. Nat Med 30, 850–862 (2024).
>
> [2] Xu, H. et al. A whole-slide foundation model for digital pathology from real-world data. Nature 630, 181–188 (2024).
>
> [3] Lu, M. Y. et al. A visual-language foundation model for computational pathology. Nat Med 30, 863–874 (2024).

---

### Meta-Review · Area_Chair_emy8 · 2025-12-14

**Summary:**

This work introduces MAMMOTH, a parameter-efficient multihead MoE to replace the task-specific linear layer in MIL for WSI classification. All the reviewers gave positive ratings, including one accept (rating 8) and three marginally above the acceptance threshold (rating 6).

The authors wrote a very thorough response during the rebuttal period, adding comprehensive discussions. Most of the reviewers were concerned about the comparison methods, while the authors have added extensive experiments to address these concerns. Related works, experimental settings, and inappropriate descriptions have been carefully revised. Reviewer DHSJ further asked about the motivation, and the authors have well clarified this part.

In summary, all of the concerns have been well addressed. I believe the reviewers will keep their positive ratings. I am very glad to recommend acceptance.

**Reviewer Concerns:**

All of the concerns have been well addressed.

**Reviewer Scores:**

I think the reviewers will keep their positive ratings.

---

### Decision · Program_Chairs · 2026-01-26

Accept (Poster)